# ZEROTH-ORDER FINE-TUNING OF LLMS WITH TRANSFERABLE STATIC SPARSITY

**Wentao Guo[1], Jikai Long[2], Yimeng Zeng[3], Zirui Liu[4], Xinyu Yang[5], Yide Ran[2],**
**Jacob Gardner[3], Osbert Bastani[3], Christopher De Sa[6], Xiaodong Yu[2],**
**Beidi Chen[5], Zhaozhuo Xu[2]**
[1]Princeton University, [3]University of Pennsylvania, [2]Stevens Institute of Technology,
[4]University of Minnesota, [5]Carnegie Mellon University, [6]Cornell University
`wg0420@princeton.edu, {jlong1,yran1,xyu38,zxu79}@stevens.edu`
`{yimengz,jacobrg,obastani}@seas.upenn.edu,zrliu@umn.edu`
`{xinyuya2,beidic}@andrew.cmu.edu,cdesa@cs.cornell.edu`

## ABSTRACT

Zeroth-order optimization (ZO) is a memory-efficient strategy for fine-tuning Large Language Models using only forward passes. However, applying ZO fine-tuning in memory-constrained settings such as mobile phones and laptops remains challenging since these settings often involve weight quantization, while ZO requires full-precision perturbation and update. In this study, we address this limitation by combining static sparse ZO fine-tuning with quantization. Our approach transfers a small, static subset (0.1%) of "sensitive" parameters from pre-training to downstream tasks, focusing fine-tuning on this sparse set of parameters. The remaining untuned parameters are quantized, reducing memory demands. Our proposed workflow enables efficient ZO fine-tuning of an Llama2-7B model on a GPU device with less than 8GB of memory while outperforming full model ZO fine-tuning performance and in-context learning. We provide an open-source implementation at `https://github.com/GarlGuo/SensZOQ`.

## 1 INTRODUCTION

Large language models (LLMs) have demonstrated superior performance in general-purpose language generation (Brown et al., 2020; Radford et al., 2019; Liu et al., 2019; Shypula et al., 2024; Hicke et al., 2023; Liu et al., 2024a; Achiam et al., 2023). Despite their success, fine-tuning LLMs for specific tasks remains necessary to achieve optimal results. However, the fine-tuning process often requires significantly more memory compared to inference. Specifically, there are four main components that occupy the memory during fine-tuning LLMs: **(1)** the weight parameter; **(2)** the optimizer state, which contains the information about the past gradient (Kingma & Ba, 2015); **(3)** the gradient used to update the parameters; **(4)** the activation cached to calculate the weight gradient (Liu et al., 2024e); Previous work, such as QLoRA (Dettmers et al., 2023), has successfully reduced memory usage for both **(1)** and **(2)** by combining weight quantization and low-rank adaptation (Hu et al., 2021), which enables fine-tuning huge LLMs under consumer level GPUs. However, on memory-constrained hardware, such as smartphones, the memory required for caching **(3)** gradient and **(4)** activations remains significant. Prior approaches to address this issue are often system-based, such as CPU offloading. The disparity between the memory demands of LLM fine-tuning and hardware capacity limits the adaptability of LLMs, especially when customizing them for edge devices.

**Exploring zeroth-order optimization in LLM fine-tuning.** Recently, there has been a resurgence of interest in Zeroth-Order (ZO) optimization methods for LLM fine-tuning (Malladi et al., 2023a; Liu et al., 2024c; Chen et al., 2024). ZO optimization method perturbs model parameters in random directions and utilizes the loss value difference to compute the gradient direction for parameter updates. A key advantage of ZO methods in LLM fine-tuning is that they do not require backpropagation procedures, significantly reducing computation and memory requirements. Being backpropagation-free, ZO methods do not need to cache **(3)** gradients and **(4)** activations during fine-tuning. In practice, ZO methods have demonstrated the potential to achieve performance comparable to first-order methods in LLM fine-tuning, which creates new venues for efficient LLM adaptation strategies.

**Efficient ZO LLM fine-tuning with sparsity.** Although ZO methods eliminate the need for backpropagation, a significant drawback of these methods is the slow convergence rate (Zhao et al., 2024b; Liu et al., 2024c). A recent approach addresses this by fine-tuning with a sparse mask (Liu et al., 2024c; Zhang et al., 2024b), achieving approximately $\sim 75\%$ *dynamic* sparsity (perturb & tune 25% parameters per step). Nonetheless, this sparsity level barely reduces computational overhead, as the latency during the forward pass with even $\sim 90\%$ sparsity is still comparable to that of dense matrix operations. This latency increase can greatly impact user experience on applications such as personal assistants, where even a twofold increase in latency is perceptible. In addition, dynamic sparsity leads to a reduction in training iterations but not necessarily wall-clock time – determine and apply the sparsity pattern for each training step could be expensive. Moreover, *dynamic sparsity inherently assumes the whole model must all be in dense weights*, and an attempt to combine dynamic sparse training with parameter-size reduction techniques such as quantization is not computationally tractable (otherwise it will involve frequent dequantization and quantization). This raises the questions:

*Is it possible to develop an extreme static sparsity method for ZO fine-tuning that is easy to combine with the quantization method? Would the memory efficiency of ZO be pushed even further?*

**Our proposal: ZO LLM fine-tuning with transferable static sparsity.** In this paper, we answer the raised research question by proposing a transferable static sparse ZO LLM fine-tuning strategy. We observe an extreme sparsity pattern in LLM parameters: a subset, determined by selecting the top $k$ magnitude entries from the diagonal of the empirical Fisher information matrix, is effective for ZO fine-tuning. Moreover, we find this sparsity pattern can be obtained through LLM's pre-training process and transferred to various downstream tasks without modification (as a static selection).

**Summary of contributions.** Building on these insights, our work proposes a comprehensive framework for ZO fine-tuning, making the following contributions:

- We identify that only an extremely small portion (**0.1%**) of LLM parameters should be updated during ZO LLM fine-tuning. Moreover, we observe that this sparsity pattern can be derived in LLM pre-training process and transferred across different downstream tasks while still maintaining good ZO performance without any modification.
- Based on this observation, we propose SensZOQ, an on-device LLM personalization workflow via integrating **Sens**itive **ZO** optimization with **Q**uantization to further improve the memory-efficiency of ZO fine-tuning (Figure 1).
- We conduct extensive experiments across various LLMs and demonstrate that our method achieves competitive performance across various downstream tasks.

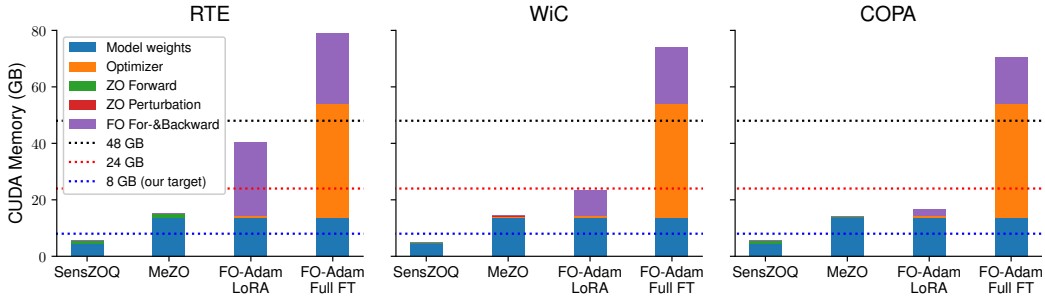

Figure 1: CUDA memory profiling of Llama2-7B on 3 fine-tuning tasks (all with batch size 8). "SensZOQ" is our method, and MeZO refers to the ZO full fine-tuning (Malladi et al., 2023a). "ZO" refers to backpropagation-free zeroth-order optimization method, and "FO" refers to the first-order optimization method. "LoRA" stands for Low-Rank adaptation (Hu et al., 2021). We find that SensZOQ can meet the 8 GB memory target *without any CPU offloading*.

## 2 SPARSE ZO FINE-TUNING WITH STATIC SENSITIVE PARAMETERS IN LLM

In this section, we first go through the background of ZO optimization in Section 2.1. We then inspect an extreme sparsity pattern in LLMs in Section 2.2 and its theoretical guarantees in Section 2.3.

### 2.1 ZEROTH-ORDER OPTIMIZATION

**ZO surrogate gradient estimator.** ZO optimizers have been studied widely in the machine learning community. Given a dataset $\mathcal{D} = \{(\mathbf{x}_1, y_1), \ldots, (\mathbf{x}_n, y_n)\}$ and a loss function $f$ with model parameters $\mathbf{w} \in \mathbb{R}^d$, ZO optimizer will estimate the gradient at $\mathbf{w}$ via ZO surrogate gradient estimator. Simultaneous Perturbation Stochastic Approximation (SPSA) (Spall, 1992) is such an estimator that would first sample a random vector $\mathbf{z} \in \mathbb{R}^d$ and uses the *loss value difference* to scale the update direction. $\mathbf{z}$ is usually sampled from an Gaussian distribution $\mathcal{N}(\mathbf{0}, \mathbf{I}_d)$.

**Definition 1** (**Simultaneous Perturbation Stochastic Approximation (SPSA)** (Spall, 1992)). *SPSA estimates the gradient w.r.t. $\mathbf{w}$ with a data example $(\mathbf{x}, y)$, a small constant $\epsilon \in \mathbb{R}$, and a sampled random vector $\mathbf{z} \in \mathbb{R}^d$ as follows:*

$$\hat{g}(\mathbf{w}, (\mathbf{x}, y), \mathbf{z}) = \frac{f(\mathbf{w} + \epsilon \mathbf{z}; (\mathbf{x}, y)) - f(\mathbf{w} - \epsilon \mathbf{z}; (\mathbf{x}, y))}{2\epsilon} \mathbf{z} \tag{1}$$

There are other ZO surrogate gradient estimators (Liu et al., 2020; Ohta et al., 2020), but in practice, SPSA achieves good performance in ZO fine-tuning. Some ZO algorithms such as DeepZero (Chen et al., 2024) would utilize the *parameter-wise* finite difference of loss values to derive *parameter-wise* update directions. This would yield $O(d)$ query costs per training step *even when combining with certain sparse masking methods* and is not practical for LLM fine-tuning scenarios. We therefore select SPSA with random Gaussian perturbation as our ZO gradient estimator.

**ZO-SGD algorithm.** ZO-SGD is an optimizer similar to SGD but replaces the FO gradient with ZO surrogate gradient estimate per training step, as defined below:

**Definition 2** (**ZO-SGD update rule**). *ZO-SGD is an optimizer that uses ZO surrogate gradient to update parameters $\mathbf{w}_t$ with learning rate $\eta_t$ and a data example $(\mathbf{x}_t, y_t)$ sampled at timestep $t$:*

$$\mathbf{w}_{t+1} = \mathbf{w}_t - \eta_t \hat{g}_{\mathbf{w}}(\mathbf{w}_t, (\mathbf{x}_t, y_t), \mathbf{z}_t) \tag{2}$$

MeZO (Malladi et al., 2023a) is a ZO-SGD algorithm that uses the "random seed trick" to save the need of caching ZO surrogate gradient. The choice of optimizer (SGD) is orthogonal to ZO optimization techniques, but in our preliminary experiments we find adaptive optimizers such as Adam (Kingma & Ba, 2015) would not necessarily accelerate ZO convergence in LLM fine-tuning scenarios. There are other ZO optimizers aware of the parameter-wise heterogeneity of loss curvatures to accelerate the optimization convergence (Zhao et al., 2024b), and we leave how to combine our method with theirs as future works.

### 2.2 SPARSE ZO OPTIMIZATION WITH STATIC SENSITIVE PARAMETERS.

Given model parameters $\mathbf{w}$, a loss function $f$, a data example $(\mathbf{x}, y)$, sensitive parameters are defined as *parameters whose corresponding FO coordinate-wise gradient square values are maximized*.

**Definition 3** (**Sensitive parameter mask**). *A sensitive sparse mask $\mathbf{m}_k \in \{0, 1\}^d$ with $k$ nonzero entries ($\sum_i \mathbf{m}(i) = k$) is defined as[1]*

$$\mathbf{m}_k = argmax_{\mathbf{m}} \|\mathbf{m} \odot \nabla f(\mathbf{w})\|_2^2. \tag{3}$$

In the context of ZO optimization, we will update sensitive parameters *only*. Denote that $\bar{\mathbf{z}} = \mathbf{z} \odot \mathbf{m}_k$. We will modify the SPSA gradient estimator from $\hat{g}(\mathbf{w}, (\mathbf{x}, y), \mathbf{z})$ to $\hat{g}(\mathbf{w}, (\mathbf{x}, y), \bar{\mathbf{z}})$, and accordingly:

**Definition 4** (**Sensitive sparse ZO-SGD update rule**).

$$\mathbf{w}_{t+1} = \mathbf{w}_t - \eta_t \hat{g}_{\mathbf{w}}(\mathbf{w}_t, (\mathbf{x}_t, y_t), \mathbf{z}_t \odot \mathbf{m}_{k,t}) \tag{4}$$

The theoretical support of sensitive parameters can be derived from the lens of SPSA gradient estimator and Fisher information matrix as follows:

---

[1]When the context is clear, we will abbreviate $f(\mathbf{w}; (\mathbf{x}, y))$ as $f(\mathbf{w})$ and $\nabla f(\mathbf{w}; (\mathbf{x}, y))$ as $\nabla f(\mathbf{w})$. Notice that for full batched gradient we will use $\nabla \mathcal{F}(\mathbf{w})$.

• **Maximum zeroth-order loss value changes, from the lens of ZO SPSA estimator.**

The square (account for negativity) of loss value difference for $\hat{g}_{\mathbf{w}}(\mathbf{w}, (\mathbf{x}, y), \bar{\mathbf{z}})$ is as follows:

$$\mathbb{E}_{\bar{\mathbf{z}}}\{f(\mathbf{w} + \epsilon\bar{\mathbf{z}}; (\mathbf{x}, y)) - f(\mathbf{w} - \epsilon\bar{\mathbf{z}}; (\mathbf{x}, y))\}^2 \approx \mathbb{E}_{\bar{\mathbf{z}}}\{2\epsilon\bar{\mathbf{z}}^\top \nabla_{\mathbf{w}} f(\mathbf{w})\}^2 = 4\epsilon^2 \|\mathbf{m}_k \odot \nabla_{\mathbf{w}} f(\mathbf{w})\|_2^2$$

Since by Definition 3 our sensitive mask would maximize $\|\mathbf{m}_k \odot \nabla_{\mathbf{w}} f(\mathbf{w})\|^2$ for a given sparsity ratio, we would expect our sensitive mask to *maximize* the magnitude of the loss value difference *for any given sparsity ratio*. This property is important for ZO as ZO directly leverages the loss-value difference as a probe of loss landscape to determine the descent direction.

• **Maximum coverage of Hessian diagonal, from the lens of Fisher matrix.**

LLMs are often pre-trained on large text corpus[2] to reach low perplexity before entering the fine-tuning stage. In this case, we would assume $p_{\text{LLM}}(y|\mathbf{x}) \sim p_{\mathcal{D}}(y|\mathbf{x})$, which implies the empirical Fisher $\hat{\mathbf{F}}$ should be close to the (true) Fisher matrix $\mathbf{F}$ as follows:

$$\mathbf{F} = \mathbb{E}_{\mathbf{x}\sim p_{\mathcal{D}}, \hat{y}\sim p_{\text{LLM}}(\cdot|\mathbf{x})} \nabla_{\mathbf{w}} \log p_{\text{LLM}}(\hat{y}|\mathbf{x})(\nabla_{\mathbf{w}} \log p_{\text{LLM}}(\hat{y}|\mathbf{x}))^\top$$

$$\approx \hat{\mathbf{F}} = \mathbb{E}_{(\mathbf{x},y)\sim p_{\mathcal{D}}} \nabla_{\mathbf{w}} \log p_{\text{LLM}}(y|\mathbf{x})(\nabla_{\mathbf{w}} \log p_{\text{LLM}}(y|\mathbf{x}))^\top$$

As we assume the empirical Fisher matrix approximates Fisher, which also approximates the Hessian, and empirical Fisher's diagonal is *equal* to the coordinate-wise gradient square vector when computing with downstream task-specific loss, our sensitive parameters would cover a large fraction of the largest Hessian diagonal entries.

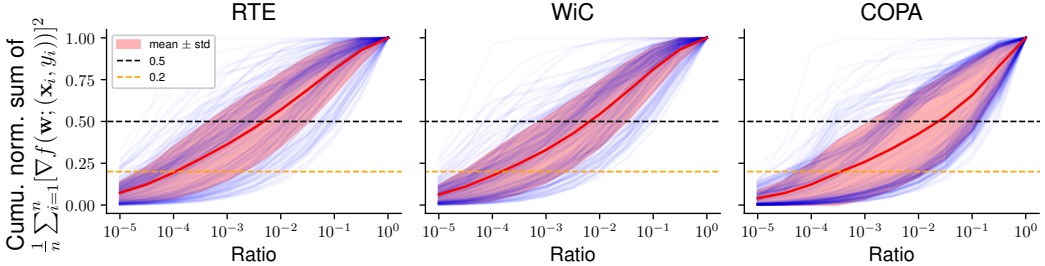

Figure 2: Cumulative normalized sum of coordinate-wise $\frac{1}{n}\sum_{i=1}^{n}[\nabla f(\mathbf{w}; (\mathbf{x}_i, y_i))]^2$ of linear layers during Llama2-7B (Touvron et al., 2023) FO-SGD full fine-tuning. For each linear layer, we first sort parameters by the decreasing order of their gradient square value $\frac{1}{n}\sum_{i=1}^{n}[\nabla f(\mathbf{w}; (\mathbf{x}_i, y_i))]^2$, and we take the cumulative sum and normalize it to draw a blue curve, and the red-shaded region is the mean $\pm$ std of all blue curves. Similar figures for Mistral-7B and OPT-6.7B are in Figure 7. **We observe that roughly 0.1%-1% parameters contribute about 50% gradient norm square.**

This idea of sensitive parameters has been studied in the quantization community (Kim et al., 2024; Guo et al., 2023) and FO optimization (Sung et al., 2021). However, *we are the first one to leverage the extremely sparse sensitive parameters in LLM fine-tuning to accelerate ZO fine-tuning with LLMs.* When we have perturbation and updating in the scale of billion parameters, finding which parameters to fine-tune would be important for improving ZO performance. Notice that here we use sensitive masks $\mathbf{m}_k$ for understanding purposes. In Section 3.2, we will discuss how to transform Definition 4 to a parameter-efficient optimization pipeline *via transferable static sparsity*.

## 2.3 THEORETICAL CONVERGENCE RATE

We would investigate the theoretical convergence of sensitive sparse ZO-SGD on sensitive parameters under the non-convex optimization settings. Our assumptions are included in Appendix C.1.

**Theorem 1 (Convergence rate of sensitive sparse ZO-SGD (Definition 4)).** *If we pick $\eta_t = 1/(L(k+2))$, under Assumptions 1 (bounded gradient error), 2 (Lipschitz smoothness), and 4 (sparse sensitive parameters), we have*

$$\frac{1}{T}\sum_{t=0}^{T-1} \mathbb{E}_{\bar{\mathbf{z}},(\mathbf{x},y)} \|\nabla_{\mathbf{w}} \mathcal{F}(\mathbf{w}_t)\|^2 \leq O\left(\frac{k}{c} \cdot \frac{L}{T}\right)(\mathcal{F}(\mathbf{w}_0) - \mathcal{F}^*) + \sigma^2. \tag{5}$$

---

[2]Here we assume data examples $(\mathbf{x}, y) \sim p_{\mathcal{D}}$ in fine-tuning datasets after verbalization would also appear in the large text corpus during pre-training.

*If we pick $\eta_t = \min\{1/(L(k+2)), 1/(c\mu)\}$ with an extra P.L. condition (Assumption 3), we have*

$$\mathbb{E}_{\bar{\mathbf{z}},(\mathbf{x},y)}\{\mathcal{F}(\mathbf{w}_T) - \mathcal{F}^*\} \leq \left(1 - O\left(\frac{\mu}{L} \cdot \frac{c}{k}\right)\right)^T (\mathcal{F}(\mathbf{w}_0) - \mathcal{F}^*) + \frac{\sigma^2}{2\mu}. \tag{6}$$

The proof for Equation 5 is in Appendix C.2 and the proof for Equation 6 is in Appendix C.3. If we choose $k = d$ and $c = 1$, both convergence rates trivially reduce to the standard zeroth-order convergence rates as $O(d/T) + O(\text{constant})$ and $(1 - O(1/d))^T + O(\text{constant})$. As we assume $c \gg k/d$, we know $d \gg k/c$ and therefore both $O((k/c)(1/T))$ and $(1 - O(c/k))^T$ are much lower than $O(d/T)$ and $(1 - O(1/d))^T$ that zeroth-order method will yield.

We want to emphasize that our contributions are more on empirical LLM fine-tuning instead of general machine learning tasks, and in Section 4.2 we extensively compare our sparse ZO methods with other sparse ZO methods and we demonstrate its superiority during LLM fine-tuning. We do not use the strict "local $r$-effective rank" assumption that Malladi et al. (2023a) uses, and our Assumption 4 can be easily observed empirically in Figure 7. Liu et al. (2024c) and Ohta et al. (2020) also provide analysis on the convergence of sparse ZO optimization but they do not include our sensitive sparse masks in their studies.

## 3 SENSZOQ: A SPARSE ON-DEVICE FINE-TUNING RECIPE

In this section, we describe the transferable static sparsity pattern we observed in LLMs and how we utilize it for developing an on-device fine-tuning pipeline of LLMs as SensZOQ.

### 3.1 TRANSFERABILITY OF STATIC SENSITIVE PARAMETER

**Transferable sensitive gradient features: from dynamic to static sparsity.** Our Theorem 1 focuses on *dynamic* sparse fine-tuning. However, Panigrahi et al. (2023) notices that in real LLM fine-tuning scenarios, the fine-tuning performance could be attributed to a sparse subset of weights ($\sim 0.01\%$). Malladi et al. (2023b) also finds certain fine-tuning tasks would demonstrate kernel behaviors, which include "fixed (gradient) features": $\nabla_{\mathbf{w}} f(\mathbf{w}_{\text{after FT}}; (\mathbf{x}, y)) \sim \nabla_{\mathbf{w}} f(\mathbf{w}_{\text{before FT}}; (\mathbf{x}, y))$.

The similarity of gradient features during fine-tuning would imply that we *do not* need to re-select our sensitive parameters during fine-tuning i.e. select once *before fine-tuning* should be sufficient. This hypothesis can be validated by Figure 3 and Figure 6b. In Figure 3, the fact that "task grad, static" does *not* vanish and still has a large ratio over "task grad, dyn." at the end of training demonstrates that we can select parameters *before fine-tuning*.

**Surrogate static sensitive parameter mask.** Another observation from Figure 3 is that the sensitive parameters derived from pre-training datasets (C4) would still cover a large fraction of model sensitivity. Specifically, the parameters overlap between top C4 gradient entries and task gradient entries are much ($>20\times$) higher than all weight magnitude baselines. Therefore, we could use it as a *surrogate* sensitive sparse mask when gradients on downstream tasks are unavailable, particularly in scenarios of *on-device personalization*. [3]

C4 covers a diverse set of text corpus across different domains and we believe it will produce a *generally good* transferable static mask. We also note that if we have better knowledge of the exact downstream domain or task our method will be applied to, we can extract sparse masks with better specific task performance. So we include an ablation study on other surrogate sensitive parameter masks in Appendix D.3 and we analyze the overlap ratio of top gradient features in Appendix E.3.

### 3.2 SENSZOQ: AN OPPORTUNITY FOR ON-DEVICE LLM PERSONALIZATION

**Transferable static sparse fine-tuning as a parameter-efficient optimization method.** The sparse optimization on *fixed* parameters can be implemented as a parameter-efficient optimization workflow, which will reduce the perturbation and updating time during ZO optimization. Suppose we have

---

[3]Obtaining gradients of LLMs on edge devices is expensive, and we usually cannot transfer data from edge devices to the cloud to compute the gradient on downstream tasks on the cloud. In this case, we would need some surrogate gradient information to derive sensitive sparse masks onthe cloud. We will discuss this in Section 3.2.

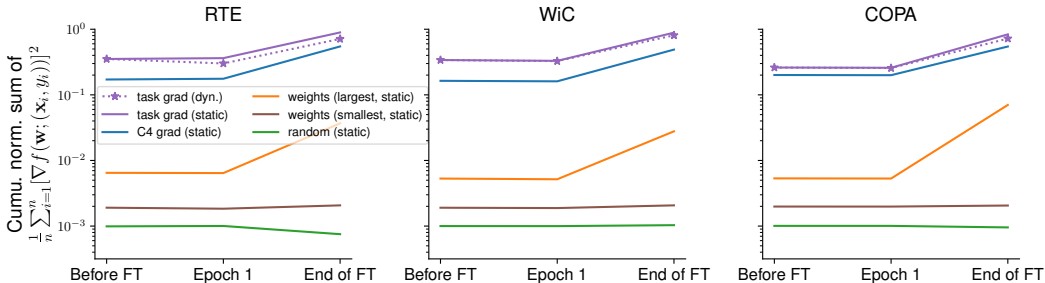

Figure 3: Cumulative normalized gradient square values (y-axis) of different sparse masks on Llama2-7B checkpoints during FO-SGD full FT. "task grad (dyn.)" refers to the sensitive parameters selected at the given timestep (x-axis) on each downstream dataset, and "task grad (static)" refers to the sensitive parameters selected before fine-tuning. "C4 grad (static)" refers to the sensitive parameters selected once with gradients taken from causal language modeling on C4 dataset (Raffel et al., 2019). "weights (largest/smallest, static)" mean we pick the weights with the largest/smallest magnitude in the pre-trained model, respectively. We provide a longer discussion in Appendix E.2.

derived a sensitive sparse mask $\mathbf{m}_k$, and we know it is fixed during fine-tuning. Instead of applying $\mathbf{m}_k$ to $\mathbf{z}$, we would apply it directly to $\mathbf{w}$ and extract the nonzero parts as below:

$$\mathbf{w}_{\text{sparse}} = \mathbf{w} \odot \mathbf{m}_k, \quad \mathbf{w}_{\text{dense}} = \mathbf{w} \odot (\mathbf{1}_d - \mathbf{m}_k) \tag{7}$$

Denote $\mathbf{z}_{k,t} \sim \mathcal{N}(\mathbf{0}_k, \mathbf{I}_k)$ as the Gaussian perturbation sampled in timestep $t$. We will determine $\mathbf{w}_{\text{sparse}}$ before fine-tuning and optimize on $\mathbf{w}_{\text{sparse}}$ *only* and leave $\mathbf{w}_{\text{dense}}$ frozen during fine-tuning. In this case, our sensitive sparse ZO-SGD update rule will become:

$$\mathbf{w}_{\text{sparse},t+1} = \mathbf{w}_{\text{sparse},t} - \eta_t \hat{g}(\mathbf{w}_{\text{sparse},t}, (\mathbf{x}_t, y_t), \mathbf{z}_{k,t}) \tag{8}$$

**Equation 8's formulation allows an opportunity for an on-device personalization workflow.**

**SensZOQ: integrating sensitive sparse ZO fine-tuning with quantization.** As LLMs are often pre-trained with user-agnostic public datasets, personalizing LLMs with individual users' preferences and meeting users' specific needs before real-world deployment are vital (Tan et al., 2024a; Mairittha et al., 2020). However, transferring the user-specific data to the upstream cloud before fine-tuning LLMs would raise privacy concerns (Xu et al., 2018). On the other hand, personal devices usually have less computational budget and are more memory-constrained than the cloud (Zhu et al., 2023), and performing full fine-tuning would easily exceed the device memory budget.

In response, we propose an on-device personalization workflow **SensZOQ** illustrated in Figure 4. The high-level overview is that we use surrogate gradient information from pre-training datasets $\nabla_{\mathbf{w}} p_{\text{LLM}}(y|\mathbf{x})$ to extract sensitive parameters $\mathbf{w}_{\text{sparse}}$ and keep $\mathbf{w}_{\text{sparse}}$ in 16 bits, while we quantize the remaining dense weights $\mathbf{w}_{\text{dense}}$ (Step 1-4). We send $\mathbf{w}_{\text{sparse}}$ and $Q(\mathbf{w}_{\text{dense}})$ to personal devices (Step 5), and **we perform on-device ZO fine-tuning only on $\mathbf{w}_{\text{sparse}}$** (Step 6).

We highlight that SensZOQ's memory consumption is *nearly minimal*: we can fine-tune a Llama2-7B model under 8 GB GPU memory *without any offloading* as illustrated in Figure 1. This would satisfy the memory constraint by a wide range of edge or mobile devices as illustrated in Table 13.

In addition, our method does *not* put strict constraints on specific choices of quantization algorithms since any algorithm that aims to minimize the least-square quantization error term $Q(\mathbf{w}) = \text{argmin}_{\hat{\mathbf{w}}} \mathbb{E}_{\mathbf{x}} \|(\mathbf{w} - \hat{\mathbf{w}})\mathbf{x}\|_2^2$ or its variant would suffice (Chee et al., 2024; Nagel et al., 2020; Frantar et al., 2022; Lin et al., 2023; Kim et al., 2024).

**Efficient implementation of sensitive sparse linear layers.** In Appendix F, we discuss how to efficiently implement our sensitive sparse ZO fine-tuning in forward passes of linear layers with training and inference workflow: use Equation 19 when we have access to efficient uniform integer matmul to compute $Q(\mathbf{w}_{\text{dense}})Q(\mathbf{x})$ and in other cases, we would usually use Equation 21 for token generation and Equation 23 for ZO training.

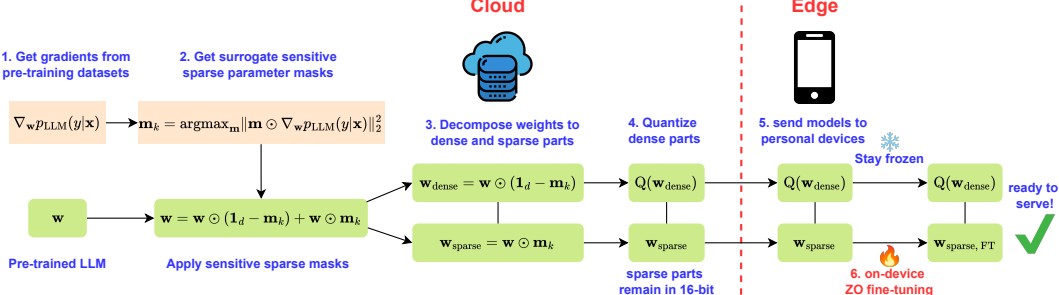

Figure 4: **SensZOQ**: an on-device LLM personalization workflow via integrating **Sens**itive **ZO** optimization with **Q**uantization. The high-level idea is that we first decompose $\mathbf{w}$ to $\mathbf{w}_{\text{sparse}}$ and $\mathbf{w}_{\text{dense}}$, and then quantize $\mathbf{w}_{\text{dense}}$ (on the cloud). On the edge devices, we will fine-tune $\mathbf{w}_{\text{sparse}}$ only.

## 4 EXPERIMENTS

In this section, we aim to validate the effectiveness of our SensZOQ, shown in Figure 4, as a memory-efficient LLM fine-tuning solution. This naturally leads to comparison with other ZO methods, which we evaluate in Section 4.1. Additionally, we assess the effectiveness of our sensitive parameter mask derived from pre-training texts (C4) against other heuristic sparsity methods in Section 4.2. Specifically, we aim to address the following research questions:

- **RQ1:** What is the performance of our SensZOQ compared with other ZO methods?
- **RQ2:** Is optimizing C4-gradient-derived sensitive parameters more effective than optimizing other subset of parameters during ZO fine-tuning?

We focus on 7B-level LLM models, including Llama2-7B (Touvron et al., 2023), Mistral-7B (Jiang et al., 2023), and OPT-6.7B (Zhang et al., 2022) as they would fit with common on-device memory constraints (8 GB) listed on Table 13 after applying quantization. We use SST-2 (Socher et al., 2013), RTE (Wang et al., 2018), CB (De Marneffe et al., 2019), BoolQ (Clark et al., 2019), WSC (Levesque et al., 2012), WiC (Pilehvar & Camacho-Collados, 2019), COPA (Roemmele et al., 2011), and WinoGrande (WinoG) (Sakaguchi et al., 2020) datasets. We follow standard ZO fine-tuning settings and use the same codebase as in Malladi et al. (2023a). More details of our experiments (hyperparameters, task-specific prompts, etc.) are described in Appendix G.

We include additional results for Llama2-13B and OPT-13B, and harder tasks such as commonsense reasoning, math reasoning, and MMLU in Appendix D. **To the best of our knowledge, there are no ZO-LLM research yet evaluated on commonsense reasoning or math tasks. We take a pioneering step in this direction and establish these ZO baselines.**

### 4.1 ON-DEVICE PERSONALIZATION

We evaluate the performance of our SensZOQ method in Table 1. We follow the exact recipe as described Figure 4, where we only optimize 0.1% sensitive parameters derived from a small batch of C4 texts on top of a 4-bit quantized model. SensZOQ's results are shown as $1^{\text{st}}$ row for each subtable.

**Comparison with ICL & ZO Full FT.** The results of in-context learning (ICL) and ZO full fine-tuning (ZO Full FT) on FP16 models are shown as the $3^{\text{rd}}$ and $5^{\text{th}}$ row for each substable in Table 1. 7B models are usually not large enough such that ICL would outperform with FT (Liu et al., 2022; Mosbach et al., 2023). In addition, ICL induces additional KV-cache memory burden if the number of demonstration examples is too many. Our method SensZOQ *does not* induce significant inference latency and memory burden as it only needs to use 0.1% parameters. SensZOQ also outperform ICL and ZO Full FT. This is impressive given that quantization would degrade performance of the base model, but SensZOQ still manages to match the fine-tuning performance.

**Comparison with ZO LoRA.** The primary purpose of quantization is to represent parameters in less bits (therefore reducing model sizes) and improve system-level metrics such as weight loading time and inference latency (Dettmers et al., 2022; Chee et al., 2024). In order to retain such benefits

Table 1: Fine-tuning performance of different methods. In the first column, "Q" means the full model is quantized with 4-bit quantization method (SqueezeLLM (Kim et al., 2024)), and "ZO-SGD" means the model is fine-tuned with ZO-SGD optimizer. For each cell, we report the mean and standard deviation of validation set accuracy (↑) of 3 random trials in the format of mean$_{std}$. We finally report the average accuracy in the last 2 columns. Notice that we add the results for FO-SGD & FO-Adam for reference and we do not use it for performance rank computation.

(a) **Llama2-7B**

| | Methods | SST-2 | RTE | CB | BoolQ | WSC | WiC | COPA | WinoG | Acc↑ |
|---|---|---|---|---|---|---|---|---|---|---|
| Q, ZO-SGD | **SensZOQ** | $94.4_{0.5}$ | $\mathbf{76.9}_{1.0}$ | $70.8_{3.0}$ | $\mathbf{83.9}_{0.4}$ | $61.9_{0.9}$ | $\mathbf{63.4}_{0.6}$ | $\mathbf{85.7}_{1.9}$ | $65.1_{0.5}$ | **75.3** |
| | LoRA | $92.9_{0.6}$ | $58.7_{0.9}$ | $67.9_{1.5}$ | $77.7_{1.4}$ | $\mathbf{62.8}_{0.5}$ | $61.3_{0.4}$ | $85.0_{0.0}$ | $62.9_{1.0}$ | 71.2 |
| ZO-SGD | Full FT | $94.6_{0.5}$ | $73.3_{5.1}$ | $66.7_{0.8}$ | $81.9_{0.8}$ | $60.6_{3.4}$ | $61.9_{0.2}$ | $82.7_{1.7}$ | $63.1_{0.4}$ | 73.1 |
| | Zero-shot | $89.0_{0.0}$ | $57.8_{0.0}$ | $32.1_{0.0}$ | $69.9_{0.2}$ | $50.2_{0.0}$ | $36.5_{0.0}$ | $79.0_{0.0}$ | $64.8_{0.6}$ | 59.9 |
| | ICL | $\mathbf{94.8}_{0.2}$ | $71.5_{4.3}$ | $\mathbf{72.6}_{15.2}$ | $77.5_{4.6}$ | $53.2_{1.1}$ | $61.1_{4.3}$ | $87.0_{2.2}$ | $\mathbf{67.5}_{1.3}$ | 73.2 |
| FO-SGD | Full FT | $95.4_{0.3}$ | $84.1_{0.9}$ | $73.2_{0.0}$ | $85.1_{1.2}$ | $62.8_{0.5}$ | $72.0_{1.8}$ | $85.3_{1.2}$ | $71.1_{1.7}$ | 78.6 |
| FO-Adam | Full FT | $95.5_{0.1}$ | $85.7_{0.7}$ | $92.9_{1.5}$ | $85.3_{0.9}$ | $64.1_{1.2}$ | $71.3_{0.4}$ | $85.3_{0.5}$ | $78.3_{1.2}$ | 82.3 |

(b) **Mistral-7B**

| | Methods | SST-2 | RTE | CB | BoolQ | WSC | WiC | COPA | WinoG | Acc↑ |
|---|---|---|---|---|---|---|---|---|---|---|
| Q, ZO-SGD | **SensZOQ** | $94.2_{0.4}$ | $\mathbf{76.8}_{3.0}$ | $67.3_{4.5}$ | $76.3_{1.0}$ | $\mathbf{59.9}_{1.2}$ | $62.2_{0.4}$ | $87.7_{0.5}$ | $73.6_{0.7}$ | **74.8** |
| | LoRA | $91.6_{1.4}$ | $69.3_{2.2}$ | $64.9_{0.8}$ | $65.2_{2.0}$ | $57.7_{1.4}$ | $62.1_{1.1}$ | $88.3_{0.9}$ | $71.6_{0.8}$ | 71.3 |
| ZO-SGD | Full FT | $\mathbf{94.6}_{0.1}$ | $74.6_{2.1}$ | $\mathbf{68.8}_{6.2}$ | $76.6_{0.2}$ | $54.8_{6.2}$ | $\mathbf{62.6}_{0.5}$ | $88.3_{0.5}$ | $72.2_{0.5}$ | 74.1 |
| | Zero-shot | $54.8_{0.0}$ | $50.5_{0.0}$ | $37.5_{0.0}$ | $43.4_{1.8}$ | $50.8_{0.0}$ | $39.4_{0.0}$ | $78.0_{0.0}$ | $66.2_{0.1}$ | 52.6 |
| | ICL | $60.7_{16.7}$ | $55.2_{4.7}$ | $33.3_{13.1}$ | $46.8_{6.5}$ | $50.4_{0.6}$ | $63.8_{0.9}$ | $\mathbf{88.7}_{0.5}$ | $\mathbf{74.0}_{0.8}$ | 59.1 |
| FO-SGD | Full FT | $94.9_{0.6}$ | $87.6_{1.2}$ | $85.7_{3.9}$ | $86.1_{0.7}$ | $62.5_{0.0}$ | $70.8_{0.6}$ | $88.3_{1.7}$ | $82.1_{1.1}$ | 82.3 |
| FO-Adam | Full FT | $95.1_{0.2}$ | $86.4_{0.7}$ | $88.1_{3.4}$ | $83.1_{1.5}$ | $64.7_{7.3}$ | $72.7_{2.9}$ | $82.7_{1.7}$ | $85.9_{0.3}$ | 82.3 |

(c) **OPT-6.7B**

| | Methods | SST-2 | RTE | CB | BoolQ | WSC | WiC | COPA | WinoG | Acc↑ |
|---|---|---|---|---|---|---|---|---|---|---|
| Q, ZO-SGD | **SensZOQ** | $\mathbf{94.6}_{0.7}$ | $\mathbf{74.1}_{1.1}$ | $\mathbf{81.0}_{3.0}$ | $70.4_{1.0}$ | $57.7_{2.4}$ | $\mathbf{62.2}_{0.6}$ | $81.7_{0.9}$ | $\mathbf{64.8}_{1.6}$ | **73.3** |
| | LoRA | $93.8_{0.9}$ | $71.0_{2.4}$ | $69.0_{1.7}$ | $70.2_{1.7}$ | $\mathbf{60.6}_{1.6}$ | $57.5_{0.5}$ | $\mathbf{84.3}_{2.5}$ | $63.0_{0.9}$ | 71.3 |
| ZO-SGD | Full FT | $94.4_{0.3}$ | $72.7_{1.2}$ | $79.8_{3.0}$ | $\mathbf{72.1}_{1.2}$ | $57.4_{4.6}$ | $60.2_{0.9}$ | $82.3_{2.6}$ | $64.6_{0.3}$ | 72.9 |
| | Zero-shot | $61.0_{0.0}$ | $60.7_{0.0}$ | $46.4_{0.0}$ | $55.7_{1.0}$ | $55.5_{0.0}$ | $36.5_{0.0}$ | $77.0_{0.0}$ | $61.1_{0.3}$ | 56.7 |
| | ICL | $74.0_{14.6}$ | $65.8_{11.2}$ | $54.8_{5.9}$ | $67.9_{2.1}$ | $53.2_{1.7}$ | $41.0_{4.5}$ | $80.7_{2.9}$ | $61.5_{0.8}$ | 62.4 |
| FO-SGD | Full FT | $95.2_{0.3}$ | $81.8_{0.9}$ | $92.3_{3.0}$ | $79.2_{1.3}$ | $59.0_{7.7}$ | $66.5_{2.3}$ | $85.7_{0.9}$ | $68.8_{0.6}$ | 78.6 |
| FO-Adam | Full FT | $95.7_{0.2}$ | $81.1_{2.6}$ | $83.9_{3.9}$ | $81.1_{0.7}$ | $56.1_{7.9}$ | $66.5_{0.5}$ | $81.3_{1.2}$ | $66.4_{0.8}$ | 76.5 |

during fine-tuning stage, our fine-tuning methods should be parameter-efficient. A natural baseline becomes fine-tuning other PEFT methods such as LoRA (Hu et al., 2021) on top of the same quantized LLM weights as SensZOQ. These results are shown as the 2$^{nd}$ row for each substable in Table 1. SensZOQ still outperforms LoRA when applied to the same 4-bit quantized base model.

**Empirical convergence speedup over ZO full FT.** To investigate the empirical convergence rate of SensZOQ, we plot their training losses w.r.t. 20k optimization steps for OPT-13B fine-tuning tasks, as shown in Figure 5. In this figure, "Sensitive ZO (C4 mask, 4 bits)" is our SensZOQ method, and the Sensitive ZO (C4 mask, FP16) will keep the dense weights in FP16.

In Figure 5a, we find that a direct learning rate transfer from ZO Full FT to SensZOQ already results in a similar convergence. **In Figure 5b, we identify that SensZOQ achieves faster convergence under the same hyperparameter searching efforts.** In Figure 5b, the best learning rates for ZO Full FT are still 2e-7 while we use 5e-7 for Sensitive ZO and produce much faster convergence results.

**Comparison with Adam Full FT of smaller size model.** The memory-efficiency of ZO is achieved via cheap queries in the loss landscape with random perturbation. The absence of accurate descent direction naturally leads an inferior performance than first-order (FO) full FT on the same model size, as also observed by Malladi et al. (2023a). However, we argue that such performance degradation is still acceptable in terms that *ZO FT on larger models outperforms FO FT on smaller models*. In Table 2, we pick 2 popular LLM sizes (1B and 7B scales) and we find that applying SensZOQ on 6.7B OPT model generally outperforms FO-Adam full FT on 1.3B OPT model, and the latter already surpasses the 8 GB memory budget (e.g., 12.5 GB on RTE compared to SensZOQ's 5.6 GB).

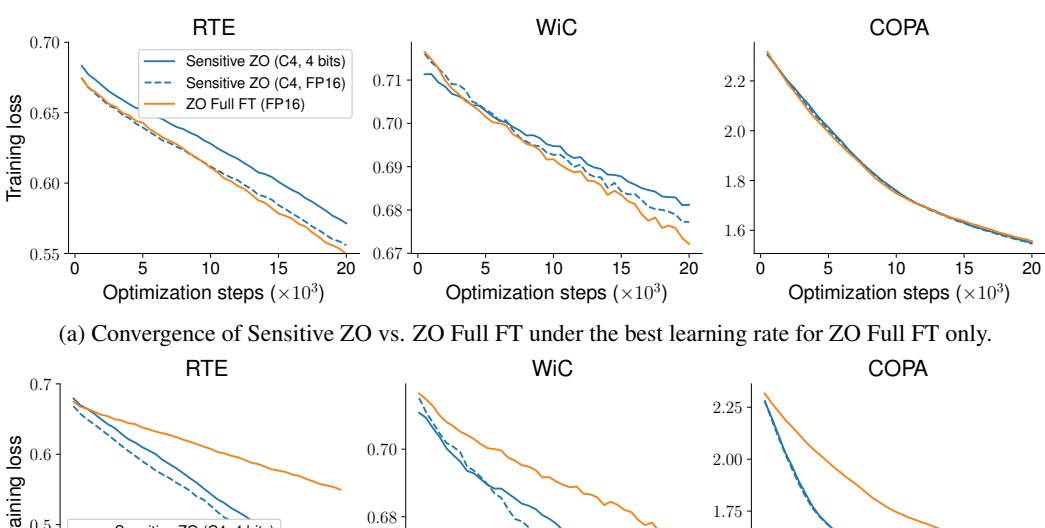

(a) Convergence of Sensitive ZO vs. ZO Full FT under the best learning rate for ZO Full FT only.

(b) Convergence of Sensitive ZO vs. ZO Full FT under the best learning rate for each method.

Figure 5: Convergence of SensZOQ for OPT-13B. SensZOQ refers to "Sensitive ZO (C4 mask, 4 bit)", and we also provide the convergence results of fine-tuning FP16 sensitive weights. In Figure 5a, We first search the best learning rate for ZO Full FT that reaches the lowest training loss in [1e-7, 2e-7, 5e-7, 1e-6] grid, and we use such learning rate for sensitive ZO. In Figure 5b, we search for the best learning rate for ZO Full FT and sensitive ZO separately on the same grid. The other hyperparameters are kept the same for both subfigures. We plot the mean over 3 random trials for all lines.

Table 2: Fine-tuning performance of SensZOQ versus Adam FT of smaller model in the OPT family. We follow the same experiment procedure as in Table 1.

| | # Params | Methods | SST-2 | RTE | CB | BoolQ | WSC | WiC | COPA | WinoG | Acc↑ |
|---|---|---|---|---|---|---|---|---|---|---|---|
| 4 bits ZO-SGD | 6.7B | **SensZOQ** | **94.6**$_{0.7}$ | **74.1**$_{1.1}$ | **81.0**$_{3.0}$ | 70.4$_{1.0}$ | 57.7$_{2.4}$ | 62.2$_{0.6}$ | 81.7$_{0.9}$ | **64.8**$_{1.6}$ | **73.3** |
| FP16 ZO-SGD | 6.7B | Full FT | 94.4$_{0.3}$ | 72.7$_{1.2}$ | 79.8$_{3.0}$ | 72.1$_{1.2}$ | 57.4$_{4.6}$ | 60.2$_{0.9}$ | **82.3**$_{2.6}$ | 64.6$_{0.3}$ | 72.9 |
| BF16 FO-Adam | 1.3B | Full FT | 93.6$_{0.5}$ | 73.9$_{2.4}$ | 75.0$_{3.9}$ | **73.8**$_{1.0}$ | **61.9**$_{1.2}$ | 62.4$_{1.5}$ | 76.7$_{1.2}$ | 60.8$_{0.6}$ | 72.3 |

## 4.2 EFFECTIVENESS OF SPARSE ZO FINE-TUNING ON SENSITIVE PARAMETERS

**Comparison with other static sparsity masks.** We investigate the performance of optimizing our sensitive parameters versus other subsets of parameters in a static sparsity regime with the FP16 model. We consider standard weight-magnitude baselines as weights with largest or smallest magnitude (SparseMeZO's sparsity patterns (Liu et al., 2024c)), and a random subset of weights baseline. There are some other weight importance metrics in the pruning community, such as GraSP (Wang et al., 2020), and we also evaluate their performance in the static transfer setting. *We note that these pruning metrics were originally proposed for deciding which parameters to retain instead of being removed during pruning, and it has no direct implications for ZO FT.* Given a threshold vector $\tau$, the definitions of all methods are listed below:

- sensitive parameters with C4 gradients: $\mathbf{m} = |\nabla f(\mathbf{w}_{\text{before FT}}; (\mathbf{x}_{\text{C4}}, y_{\text{C4}}))| \geq \tau$
- random subsets: $\mathbf{m} = \text{random\_dim\_d\_vector\_with\_k\_nnz}(d, k)$
- weights with largest magnitude: $\mathbf{m} = |\mathbf{w}_{\text{before FT}}| \geq \tau$
- weights with smallest magnitude: $\mathbf{m} = |\mathbf{w}_{\text{before FT}}| \leq \tau$
- smallest GraSP scores: $\mathbf{m} = -\mathbf{w}_{\text{before FT}} \odot \mathbb{E}_{(\mathbf{x}, y) \sim \mathcal{D}} H(\mathbf{w}_{\text{before FT}}) \nabla_{\mathbf{w}} f(\mathbf{w}_{\text{before FT}}) \leq \tau$

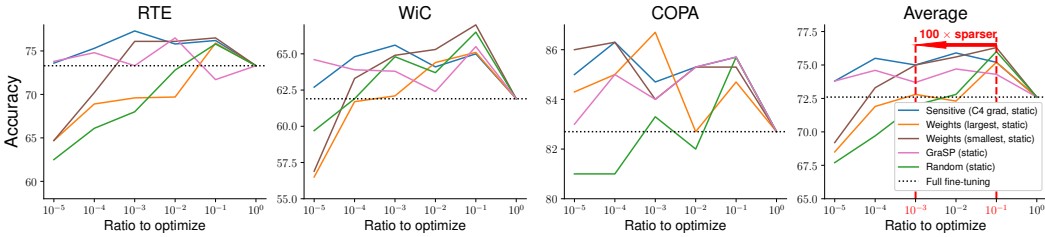

(a) Static transferability performance of different sparsity masks. "static" means that we will determine the trainable parameters (sparsity mask) before fine-tuning and other parameters are kept unchanged.

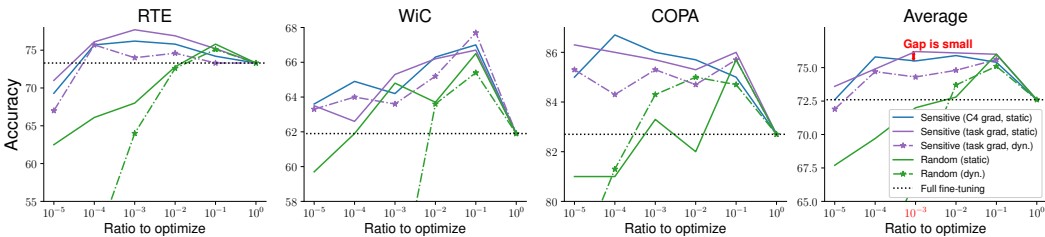

(b) The performance of sensitive parameters derived from causal LM loss in C4 datasets and gradients from each fine-tuning task. "Dyn." means the sparse masks are updated every 100 training steps (dynamic sparsity).

Figure 6: Performance of different sparsity methods in Llama2-7B ZO fine-tuning.

As illustrated in Figure 6a, we can find that ZO FT would benefit from sparse optimization, as all methods would achieve higher than ZO full FT when optimizing 10% parameters. However, only sensitive parameters would maintain its performance as we move to the extreme sparsity region (<1%). *In fact, the performance curve of sensitive parameters w.r.t. different sparsity levels is near a flat curve*, which indicates the performance loss by moving from 10% to 0.1% is minimal. We also find that optimizing weights with smallest magnitude is more effectively than optimizing weights with largest magnitude, which aligns with Liu et al. (2024c)'s findings. However, sensitive parameters is still more effective for optimizing weights with smallest magnitude. This suggests that sensitive parameters are more effective to serve as the sparse parameters in our SensZOQ instead of SparseMeZO's weights with smallest magnitude.

**Transferability of SensZOQ's sparsity masks.** SensZOQ uses C4 gradients to produce a transferable static mask used in downstream tasks. In Figure 6b, we compare the performance of optimizing sensitive parameters with gradients on C4 dataset with its theoretical upper bound: static sensitive parameters derived from gradients on each fine-tuning task as the solid line and its dynamic version as the dash-dotted line. We also include the static and dynamic random subset parameters as a baseline. We can find that the gap of sensitive parameters between deriving from gradients on the C4 dataset and gradients on each fine-tuning task at a ratio 1e-3 is *small*. Together with Figure 9 that we evaluate the top gradient entries' similarity between C4 and downstream tasks, we believe SensZOQ's sensitive masks from C4 gradients would yield satisfactory performance in general.

## 5 CONCLUSION

In this work, we identify that only a small portion of LLM parameters needs to be updated during ZO fine-tuning, and these static and sparse subset parameters can be derived during the pre-training phase and transferred across various downstream tasks without requiring any modifications, preserving efficient ZO performance. We propose SensZOQ, a workflow that integrates sparse ZO optimization with 4-bit quantization to further enhance the memory efficiency of on-device fine-tuning. SensZOQ leverages static sparse fine-tuning to enable the personalization of 7B LLMs on-device, reducing memory consumption to less than 8 GB of CUDA memory. In addition to this efficiency, SensZOQ achieves better performance than both in-context learning and ZO full fine-tuning. Therefore, SensZOQ creates a new venue to facilitate on-device fine-tuning.

## 6 ACKNOWLEDGMENT

This research used resources of the Argonne Leadership Computing Facility, a U.S. Department of Energy (DOE) Office of Science user facility at Argonne National Laboratory and is based on research supported by the U.S. DOE Office of Science-Advanced Scientific Computing Research Program, under Contract No. DE-AC02-06CH11357. We also appreciate the support from Amazon, Intel, Li Auto, and Moffet AI. We also thank Lin Xiao and numerous anonymous reviewers for providing valuable feedback on this paper.

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

APPENDIX

In Section A, we discuss some related works of our paper. In Section B we describe all notations used in this paper. In Section C, we include the assumptions and exact proof on the convergence rate (Theorem 1). In Section D, we include experiment results on OPT-13B and Llama2-13B. We also take a pioneering move to investigate the effectiveness of SensZOQ and ZO methods in commonsense reasoning, Math, and MMLU tasks. In Section E, we inspect the appearance of sensitive parameters across models and tasks. We also investigate the effects of different data sources that would produce different sensitive masks. In Section F, we provide a high-level recommendation on how to efficiently implement our SensZOQ's framework in linear layers with existing quantization methods or training/inference workflow. In Section G, we describe miscellaneous details (hyperparameters, task templates, hardware config, etc.) in our experiments.

## A  RELATED WORKS

**Zeroth-order fine-tuning of LLMs.**  Since MeZO (Malladi et al., 2023a) first demonstrates the effectiveness of ZO for LLM fine-tuning, ZO has attracted great research interest from the LLM community in different aspects. For example, ZO is effective for edge device fine-tuning in communication-constrained and federated settings (Ling et al., 2024; Zhang et al., 2024a; Tang et al., 2024; Liu et al., 2024d). Notably, Zelikman et al. (2023) illustrates the possibility of exchanging single-byte projected gradients in distributed zeroth-order workloads, yielding both communication and privacy benefits. Numerous research has also focused on enhancing the optimizer aspect of zeroth-order optimization (Jiang et al., 2024; Pang & Zhou, 2024; Gautam et al., 2024). Other researchers are also interested in improving ZO's efficiency or convergence rate from cleverer optimizer designs. Li et al. (2024b) explores the middle ground between small-batched FO-SGD and large-batched ZO-SGD to balance the convergence speed and memory footprints. Liu et al. (2024c) and Zhang et al. (2024b) suggest that sparsity would potentially accelerate ZO optimization convergence. Zhao et al. (2024b) preconditions ZO perturbation with knowledge from parameter-wise loss curvature heterogeneity to gain convergence speedup. To the best of our knowledge, Liu et al. (2024c)'s work is the only ZO fine-tuning with sparsity (smallest weight magnitude mask) work at this moment, and we have ablated on its static extreme sparsity performance in Figure 6a.

**Sparsity in LLM.**  Sparsity-driven techniques are widely adopted in improving ML model's efficiency (Frankle & Carbin, 2019; Xia et al., 2023; Liu et al., 2023; Peng et al., 2013; Xu et al., 2024; Tan et al., 2024b) and robustness (Zhong et al., 2024; 2021). Frankle & Carbin (2019) shows that within large feed-forward networks, there exists a subnetwork that, when trained in isolation, can achieve test accuracy comparable to that of the original network. In the foundation models era, Liu et al. (2023) demonstrates that transformer-based models, such as OPT (Zhang et al., 2022), exhibit great sparsity ($\geq 95\%$) in activations. Moreover, Panigrahi et al. (2023) discovers that for RoBERTa (Liu et al., 2019), fine-tuning a very small subset of parameters ($\sim 0.01\%$) can yield performance exceeding $95\%$ of that achieved by full fine-tuning.

**First-order gradient optimization methods.**  Researchers in the optimization community also propose a variety of first-order optimization methods to promote faster convergence or reduce the memory footprints. For faster convergence, Liu et al. (2024b) designs a second-order optimizer that approximates the diagonal Hessian and enables more stable and faster LLM training. You et al. (2020) proposes a variant of Adam that enables the use of large batch sizes to finish BERT (Devlin et al., 2019) pre-training within 76 minutes. Cooper et al. (2023) and Lu et al. (2022) also attempt to effectively reorder the training dataset online to accelerate convergence. To reduce memory footprints, Zhao et al. (2024a) projects the gradient to a low-rank subspace that will simultaneously reduce gradients and Adam optimizer states. Both Shazeer & Stern (2018) and Lv et al. (2024) factor the second-moment of adaptive optimizer to reduce the Adam optimizer state memory costs.

# B NOTATIONS

We present the notations used in this work as follows.

Table 3: Notations used in this paper

| Term/Symbol | Explanation |
|---|---|
| $f$ | loss function |
| $t$ | optimization timestep $t$ |
| $n$ | number of training data examples (training dataset size) |
| $d$ | number of model parameters |
| $d_{\text{layer}}$ | number of parameters in one linear layer. This means the number of input channels times the number of output channels in each linear layer. |
| $(\mathbf{x}_t, y_t)$ | a data example sampled at timestep $t$ as a pair of input vector and training target |
| $\mathbf{w}_t \in \mathbb{R}^d$ | weight/parameter vector at optimization timestep $t$ |
| $f(\mathbf{w}; (\mathbf{x}, y))$ | training loss of $\mathbf{w}$ evaluated at a single data example $(\mathbf{x}, y)$ |
| $\mathcal{F}(\mathbf{w})$ | full-batched training loss of $\mathbf{w}$ |
| $H(\mathbf{w}; (\mathbf{x}, y))$ | Hessian matrix of $\mathbf{w}$ evaluated at $(\mathbf{x}, y)$ |
| $\epsilon$ | a small perturbation scaling constant (close to 0) |
| $\mathbf{z}_t \in \mathbb{R}^d$ | random Gaussian perturbation vector sampled at timestep $t$ |
| $\hat{g}(\mathbf{w}, (\mathbf{x}, y), \mathbf{z})$ | estimated ZO surrogate gradient for $\mathbf{w}$ with a data example $(\mathbf{x}, y)$ and a sampled Gaussian perturbation vector $\mathbf{z}$ (Definition 1) |
| $\eta_t$ | learning rate for ZO-SGD optimizer (Definition 2) at timestep $t$ |
| $\mathbf{m}_k \in \{0,1\}^d$ | a sensitive sparse mask with $k$ nonzero entries (Definition 3) |
| $\mathbf{m}_{k,t} \in \{0,1\}^d$ | a sensitive sparse mask with $k$ nonzero entries derived at optimization timestep $t$ |
| $\mathbf{I}_d$ | Identity matrix with shape $\mathbb{R}^{d \times d}$ |
| $\tilde{\mathbf{I}}_{d,\mathbf{m}_k}$ | Identity matrix $\mathbf{I}_d$ with the main diagonal masked by $\mathbf{m}_k$ |
| $\mathbf{1}_d$ | a vector of size $d$ with all entries equal to 1 |
| Tr | trace operation |
| $Q(\mathbf{w})$ | parameter vector $\mathbf{w}$ that is quantized by quantizer $Q$ |
| $\mathbf{F}$ | (true) Fisher information matrix |
| $\hat{\mathbf{F}}$ | empirical Fisher information matrix |
| $p_{\text{LLM}}$ | LLM as a probabilistic model |
| $p_{\mathcal{D}}$ | data distribution |
| $L$ | Lipschitz constant in Assumption 2 |
| $\mu$ | P.L. condition number in Assumption 3 |
| $\sigma^2$ | stochastic gradient error term in Assumption 1 |
| $W_Q$ | the query weight matrix in self attention layers |
| $W_K$ | the key weight matrix in self attention layers |
| $W_V$ | the value weight matrix in self attention layers |
| $W_O$ | the output weight matrix in self attention layers |
| $W_{\text{Gate}}$ | the weight matrix for the gated linear unit in SwiGLU layers |
| $W_{\text{Up}}$ | the weight matrix for the up projection layer in SwiGLU/MLP layers |
| $W_{\text{Down}}$ | the weight matrix for the down projection layer in SwiGLU/MLP layers |

## C THEORETICAL CONVERGENCE RATE

### C.1 ASSUMPTIONS

We start with listing standard assumptions in the nonconvex optimization literature:

**Assumption 1** (**Bounded stochastic gradient errors**). *For any data example* $(\mathbf{x}, y) \in \mathcal{D}$ *and for any* $\mathbf{w} \in \mathbb{R}^d$, *denote the full-batched loss function* $\mathcal{F}(\mathbf{w}) = \mathbb{E}_{(\mathbf{x},y)\in\mathcal{D}} f(\mathbf{w}; (\mathbf{x}, y))$, *we have*

$$\|\nabla_{\mathbf{w}} f(\mathbf{w}; (\mathbf{x}, y)) - \nabla_{\mathbf{w}}\mathcal{F}(\mathbf{w})\|^2 \leq \sigma^2. \tag{9}$$

**Assumption 2** (**Lipschitz smoothness**). *We assume that* $f(\mathbf{w}, \mathbf{x})$ *is L-Lipschitz smooth* ($L > 0$): *for any* $\mathbf{w}, \mathbf{w}' \in \mathbb{R}^d$,

$$\|\nabla_{\mathbf{w}} f(\mathbf{w}; (\mathbf{x}, y)) - \nabla_{\mathbf{w}} f(\mathbf{w}'; (\mathbf{x}, y))\| \leq L\|\mathbf{w} - \mathbf{w}'\|. \tag{10}$$

**Assumption 3** (**P.L. inequality**). *We assume that* $\mathcal{F}(\mathbf{w})$ *fulfills the Polyak-Lojasiewicz (P.L.) condition: there exists some* $\mu > 0$, *for any* $\mathbf{w} \in \mathbb{R}^d$

$$\frac{1}{2}\|\nabla_{\mathbf{w}}\mathcal{F}(\mathbf{w})\|^2 \geq \mu(\mathcal{F}(\mathbf{w}) - \mathcal{F}^*), \quad \mathcal{F}^* \text{ is the minimum value } \mathcal{F}^* = \inf_{\mathbf{w}} \mathcal{F}(\mathbf{w}). \tag{11}$$

Inspired by Figure 7, we assume the sensitive parameters of $\mathbf{w}$ are sparse.

**Assumption 4** (**Sensitive parameters are sparse**). *We assume at timestep* $t$ $\exists \mathbf{m}_t \in \{0, 1\}^d$ *with the number of nonzero entries as* $k$, $\exists c \in [0, 1]$ *such that*

$$\|\mathbf{m}_t \odot \nabla_{\mathbf{w}} f(\mathbf{w}_t; (\mathbf{x}_t, y_t))\|^2 = c\|\nabla_{\mathbf{w}} f(\mathbf{w}_t; (\mathbf{x}_t, y_t))\|^2.$$

*Here we assume* $c \gg k/d$. [4]

### C.2 PROOF FOR EQUATION 5, THEOREM 1

We will start with formulating the expectation of sensitive sparse ZO surrogate gradient norm square in terms of its corresponding stochastic gradient norm square.

**Lemma 1** (**Sensitive sparse ZO surrogate gradient norm square**).

$$\mathbb{E}_{\bar{\mathbf{z}}}[\|\hat{g}(\mathbf{w}_t, (\mathbf{x}_t, y_t), \bar{\mathbf{z}}_t)\|^2] = (2 + k)c\|\nabla_{\mathbf{w}} f(\mathbf{w}, (\mathbf{x}_t, y_t))\|^2$$

***Proof for Lemma 1.*** We know that our $\bar{\mathbf{z}}$ can be considered as being sampled from $\mathcal{N}(\mathbf{0}, \tilde{\mathbf{I}}_{d, \mathbf{m}_k})$ where $\tilde{\mathbf{I}}_{d, \mathbf{m}_k}$ is the identity matrix $\mathbf{I}_d$ with the main diagonal masked by $\mathbf{m}_k$.

We expand the sensitive sparse ZO surrogate gradient covariance matrix as follows:

$$\mathbb{E}_{\bar{\mathbf{z}}}\hat{g}(\mathbf{w}, (\mathbf{x}, y), \bar{\mathbf{z}})\hat{g}(\mathbf{w}, (\mathbf{x}, y), \bar{\mathbf{z}})^\top$$
$$= \mathbb{E}_{\bar{\mathbf{z}}_i}[\bar{\mathbf{z}}_i\bar{\mathbf{z}}_i^\top \left((\mathbf{m}_k \odot \nabla_{\mathbf{w}} f(\mathbf{w}; (\mathbf{x}, y)))(\mathbf{m}_k \odot \nabla_{\mathbf{w}} f(\mathbf{w}; (\mathbf{x}, y)))^\top\right) \bar{\mathbf{z}}_i\bar{\mathbf{z}}_i^\top]$$
$$= 2\left((\mathbf{m}_k \odot \nabla_{\mathbf{w}} f(\mathbf{w}; (\mathbf{x}, y)))(\mathbf{m}_k \odot \nabla_{\mathbf{w}} f(\mathbf{w}; (\mathbf{x}, y)))^\top\right) + \|\mathbf{m}_k \odot \nabla_{\mathbf{w}} f(\mathbf{w}; (\mathbf{x}, y))\|^2 \tilde{\mathbf{I}}_{d, \mathbf{m}_k}$$

Then the sensitive sparse ZO surrogate gradient norm square is the square of the *diagonal* of its corresponding covariance matrix:

$$\mathbb{E}_{\bar{\mathbf{z}}}[\|\hat{g}(\mathbf{w}_t, \mathbf{x}_t, \bar{\mathbf{z}}_t)\|^2] = \text{diag}\left(\mathbb{E}_{\bar{\mathbf{z}}}\hat{g}(\mathbf{w}, (\mathbf{x}, y), \bar{\mathbf{z}})\hat{g}(\mathbf{w}, (\mathbf{x}, y), y), \bar{\mathbf{z}})^\top\right)^2$$
$$= 2c\|\nabla_{\mathbf{w}} f(\mathbf{w}, (\mathbf{x}_t, y_t))\|^2 + kc\|\nabla_{\mathbf{w}} f(\mathbf{w}, (\mathbf{x}_t, y_t))\|^2$$
$$= (2 + k)c\|\nabla_{\mathbf{w}} f(\mathbf{w}, (\mathbf{x}_t, y_t))\|^2$$

$\square$

Then we are in good shape of deriving the convergence rate under the Lipschitz smoothness condition:

---

[4] From Figure 7, we know that for $c \sim 0.5$, we only need $k/d \sim 0.001$ to $0.01$. In this case $k/c \sim 0.002d$ to $0.02d$.

***Proof for Equation 5, Theorem 1.***

$$f(\mathbf{w}_{t+1}, \mathbf{x}_t) \leq f(\mathbf{w}_t; (\mathbf{x}_t, y_t)) + \langle \nabla f(\mathbf{w}_t; (\mathbf{x}_t, y_t)), \mathbf{w}_{t+1} - \mathbf{w}_t \rangle + \frac{L}{2} \|\mathbf{w}_{t+1} - \mathbf{w}_t\|^2$$

$$\leq f(\mathbf{w}_t; (\mathbf{x}_t, y_t)) - \eta_t \langle \nabla f(\mathbf{w}_t; (\mathbf{x}_t, y_t)), \hat{g}(\mathbf{w}_t, \mathbf{x}_t, \bar{\mathbf{z}}_t) \rangle + \frac{L\eta_t^2}{2} \|\hat{g}(\mathbf{w}_t, \mathbf{x}_t, \bar{\mathbf{z}}_t)\|^2$$

$$\mathbb{E}_{\bar{\mathbf{z}}} f(\mathbf{w}_{t+1}, \mathbf{x}_t) \leq \mathbb{E}_{\bar{\mathbf{z}}} f(\mathbf{w}_t; (\mathbf{x}_t, y_t)) - \eta_t \mathbb{E}_{\bar{\mathbf{z}}} \|\mathbf{m}_{k,t} \odot \nabla f(\mathbf{w}_t; (\mathbf{x}_t, y_t))\|^2 + \frac{L\eta_t^2}{2} \mathbb{E}_{\bar{\mathbf{z}}} \|\hat{g}(\mathbf{w}_t, \mathbf{x}_t, \bar{\mathbf{z}})\|^2$$

$$\mathbb{E}_{\bar{\mathbf{z}}} f(\mathbf{w}_{t+1}, \mathbf{x}_t) \leq \mathbb{E}_{\bar{\mathbf{z}}} f(\mathbf{w}_t; (\mathbf{x}_t, y_t)) - c\eta_t \mathbb{E}_{\bar{\mathbf{z}}} \|\nabla f(\mathbf{w}_t; (\mathbf{x}_t, y_t))\|^2 + \frac{L\eta_t^2}{2} c(k+2) \mathbb{E}_{\bar{\mathbf{z}}} \|\nabla_{\mathbf{w}} f(\mathbf{w}_t; (\mathbf{x}_t, y_t))\|^2$$

$$\mathbb{E}_{\bar{\mathbf{z}}, (\mathbf{x}, y)} \mathcal{F}(\mathbf{w}_{t+1}) \leq \mathbb{E}_{\bar{\mathbf{z}}, (\mathbf{x}, y)} \{ \mathcal{F}(\mathbf{w}_t) - c\eta_t \|\nabla_{\mathbf{w}} \mathcal{F}(\mathbf{w}_t)\|^2 + \frac{L\eta_t^2}{2} c(k+2) \|\nabla_{\mathbf{w}} \mathcal{F}(\mathbf{w}_t)\|^2 + \frac{L\eta_t^2}{2} c(k+2)\sigma^2 \}$$

$$\mathbb{E}_{\bar{\mathbf{z}}, (\mathbf{x}, y)} \mathcal{F}(\mathbf{w}_{t+1}) \leq \mathbb{E}_{\bar{\mathbf{z}}, (\mathbf{x}, y)} \{ \mathcal{F}(\mathbf{w}_t) - \left( c\eta_t - \frac{L\eta_t^2}{2} c(k+2) \right) \|\nabla_{\mathbf{w}} \mathcal{F}(\mathbf{w}_t)\|^2 + \frac{L\eta_t^2}{2} c(k+2)\sigma^2 \}$$

Denote $\alpha = Lc(k+2)$, we will have

$$\mathbb{E}_{\bar{\mathbf{z}}, (\mathbf{x}, y)} \mathcal{F}(\mathbf{w}_{t+1}) \leq \mathbb{E}_{\bar{\mathbf{z}}, (\mathbf{x}, y)} \left\{ \mathcal{F}(\mathbf{w}_t) - \eta_t \left( c - \frac{\alpha}{2} \eta_t \right) \|\nabla_{\mathbf{w}} \mathcal{F}(\mathbf{w}_t)\|^2 \right\} + \frac{\alpha}{2} \sigma^2 \eta_t^2$$

Suppose we use a constant learning rate $\eta_t = \eta = \dfrac{c}{\alpha} = \dfrac{1}{L(k+2)}$[5], we have

$$\mathbb{E}_{\bar{\mathbf{z}}, (\mathbf{x}, y)} \mathcal{F}(\mathbf{w}_{t+1}) \leq \mathbb{E}_{\bar{\mathbf{z}}, (\mathbf{x}, y)} \left\{ \mathcal{F}(\mathbf{w}_t) - \frac{c\eta}{2} \|\nabla \mathcal{F}(\mathbf{w}_t)\|^2 \right\} + \frac{\alpha}{2} \sigma^2 \eta^2 \qquad (12)$$

If we apply our descent rule (Equation 12) recursively for $T$ steps,

$$\frac{1}{T} \sum_{t=0}^{T-1} \mathbb{E}_{\bar{\mathbf{z}}, (\mathbf{x}, y)} \|\nabla_{\mathbf{w}} \mathcal{F}(\mathbf{w}_t)\|^2 \leq \frac{2}{c\eta T} (\mathcal{F}(\mathbf{w}_0) - \mathcal{F}^*) + \frac{1}{T} \sum_{t=0}^{T-1} \frac{\left( \frac{\alpha}{2} \sigma^2 \eta^2 \right)}{\frac{c\eta}{2}}$$

$$\leq \frac{2L(k+2)}{cT} (\mathcal{F}(\mathbf{w}_0) - \mathcal{F}^*) + \sigma^2$$

$$\leq \frac{2L(k+2)}{c} \frac{1}{T} (\mathcal{F}(\mathbf{w}_0) - \mathcal{F}^*) + \sigma^2$$

$$= O\left( \frac{k}{c} \cdot \frac{L}{T} \right) (\mathcal{F}(\mathbf{w}_0) - \mathcal{F}^*) + O(\text{constant})$$

$$\square$$

### C.3 PROOF FOR EQUATION 6, THEOREM 1

We can derive a convergence rate of sensitive sparse ZO-SGD optimization method under P.L. inequality and Lipschitz-smoothness as follows (this proof resumes from our prior proof with the Lipschitz-smoothness condition alone):

***Proof for Equation 6, Theorem 1.*** Let's resume from Equation 12:

---

[5]If we use a diminishing learning rate we can remove the constant noise term, but this is not our main focus in this paper.

$$\mathbb{E}_{\bar{\mathbf{z}},(\mathbf{x},y)}\mathcal{F}(\mathbf{w}_{t+1}) \leq \mathbb{E}_{\bar{\mathbf{z}},(\mathbf{x},y)}\left\{\mathcal{F}(\mathbf{w}_t) - \frac{c\eta}{2}\|\nabla\mathcal{F}(\mathbf{w}_t)\|^2\right\} + \frac{\alpha}{2}\sigma^2\eta^2$$

$$\leq \mathbb{E}_{\bar{\mathbf{z}},(\mathbf{x},y)}\left\{\mathcal{F}(\mathbf{w}_t) - c\mu\eta(\mathcal{F}(\mathbf{w}_t) - \mathcal{F}^*)\right\} + \frac{\alpha}{2}\sigma^2\eta^2$$

$$\mathbb{E}_{\bar{\mathbf{z}},(\mathbf{x},y)}\{\mathcal{F}(\mathbf{w}_{t+1}) - \mathcal{F}^*\} \leq \mathbb{E}_{\bar{\mathbf{z}},(\mathbf{x},y)}\left\{(\mathcal{F}(\mathbf{w}_t) - \mathcal{F}^*) - c\mu\eta(\mathcal{F}(\mathbf{w}_t) - \mathcal{F}^*)\right\} + \frac{\alpha}{2}\sigma^2\eta^2$$

$$\mathbb{E}_{\bar{\mathbf{z}},(\mathbf{x},y)}\{\mathcal{F}(\mathbf{w}_{t+1}) - \mathcal{F}^*\} \leq (1 - c\mu\eta)\,\mathbb{E}_{\bar{\mathbf{z}},(\mathbf{x},y)}\{\mathcal{F}(\mathbf{w}_t) - \mathcal{F}^*\} + \frac{\alpha}{2}\sigma^2\eta^2$$

Plugging in a constant learning rate $\eta = \min\{\dfrac{1}{L(k+2)}, \dfrac{1}{c\mu}\}$ and sum up recursively for $T$ iterations,

$$\mathbb{E}_{\bar{\mathbf{z}},(\mathbf{x},y)}\{\mathcal{F}(\mathbf{w}_T) - \mathcal{F}^*\} \leq (1 - \frac{c\mu}{L(k+2)})^T(\mathcal{F}(\mathbf{w}_0) - \mathcal{F}^*) + \frac{\alpha}{2}\sigma^2 \cdot \eta^2 \cdot \sum_{j=0}^{T-1}(1 - c\mu\eta)^j$$

$$\leq (1 - \frac{c\mu}{L(k+2)})^T(\mathcal{F}(\mathbf{w}_0) - \mathcal{F}^*) + \frac{\alpha}{2}\sigma^2 \cdot \eta^2 \cdot \frac{1}{c\mu\eta}$$

$$\leq (1 - \frac{c\mu}{L(k+2)})^T(\mathcal{F}(\mathbf{w}_0) - \mathcal{F}^*) + \frac{\alpha}{2}\sigma^2 \cdot \frac{c}{\alpha} \cdot \frac{1}{c\mu}$$

$$\leq (1 - \frac{c\mu}{L(k+2)})^T(\mathcal{F}(\mathbf{w}_0) - \mathcal{F}^*) + \frac{\sigma^2}{2\mu}$$

$$= \left(1 - O\left(\frac{\mu}{L} \cdot \frac{c}{k}\right)\right)^T (\mathcal{F}(\mathbf{w}_0) - \mathcal{F}^*) + O(\text{constant})$$

$\square$

# D  MORE EXPERIMENT RESULTS

## D.1  13B MODEL RESULTS.

In Table 4, we compare SensZOQ vs. ZO full FT & ICL in 2 popular 13B models: Llama2-13B and OPT-13B. SensZOQ still maintains its superior performance over both ICL and ZO Full FT. In the last row of OPT-13B (Table 4b), we ablate on the effect of 4-bit SqueezeLLM quantization (K-means (MacQueen, 1967; Zhang & Amini, 2021) based quantization) and as expected, the average test accuracy is roughly the same after we remove the quantization on the dense weight parts.

Table 4: Fine-tuning performance of different methods. In the first column, "Q" means the full model is quantized with 4-bit quantization method (SqueezeLLM (Kim et al., 2024)), and "ZO" means the model is fine-tuned with ZO-SGD optimizer. For each cell, we report the mean and standard deviation of test set accuracy ($\uparrow$) of 3 random trials in the format of $\text{mean}_{\text{std}}$. **We finally report the average accuracy across tasks in the last column.**

(a) **Llama2-13B**

| | Methods | SST-2 | RTE | CB | BoolQ | WSC | WiC | COPA | WinoG | Avg |
|---|---|---|---|---|---|---|---|---|---|---|
| Q, ZO | **SensZOQ** | $\mathbf{95.8}_{0.3}$ | $\mathbf{78.8}_{1.0}$ | $81.5_{4.5}$ | $\mathbf{80.6}_{0.3}$ | $\mathbf{63.5}_{0.8}$ | $59.8_{0.9}$ | $89.7_{0.5}$ | $\mathbf{74.6}_{0.6}$ | **78.0** |
| ZO | Full FT | $95.0_{0.2}$ | $74.4_{1.0}$ | $\mathbf{85.7}_{1.5}$ | $80.5_{0.4}$ | $54.8_{4.7}$ | $\mathbf{60.9}_{1.4}$ | $\mathbf{90.7}_{0.5}$ | $70.5_{0.2}$ | 76.6 |
| | Zero-shot | $75.6_{0.0}$ | $54.5_{0.0}$ | $48.2_{0.0}$ | $76.1_{0.2}$ | $39.4_{0.0}$ | $50.6_{0.0}$ | $83.0_{0.0}$ | $65.4_{0.4}$ | 61.6 |
| | ICL | $95.4_{0.4}$ | $78.6_{2.5}$ | $79.8_{5.9}$ | $75.7_{4.2}$ | $57.4_{1.6}$ | $57.6_{3.0}$ | $89.3_{0.9}$ | $71.3_{0.5}$ | 75.6 |

(b) **OPT-13B**

| | Methods | SST-2 | RTE | CB | BoolQ | WSC | WiC | COPA | WinoG | Avg |
|---|---|---|---|---|---|---|---|---|---|---|
| Q, ZO | **SensZOQ** | $93.6_{0.5}$ | $\mathbf{75.2}_{2.1}$ | $\mathbf{69.0}_{1.7}$ | $\mathbf{73.0}_{1.4}$ | $61.5_{0.8}$ | $\mathbf{61.0}_{1.6}$ | $\mathbf{87.0}_{0.5}$ | $\mathbf{65.1}_{0.2}$ | **73.2** |
| ZO | Full FT | $\mathbf{93.9}_{0.5}$ | $74.0_{1.0}$ | $67.9_{2.5}$ | $72.4_{0.3}$ | $\mathbf{61.5}_{2.4}$ | $58.6_{2.3}$ | $\mathbf{87.0}_{1.4}$ | $63.3_{1.3}$ | 72.3 |
| | Zero-shot | $61.0_{0.0}$ | $58.5_{0.0}$ | $48.2_{0.0}$ | $59.8_{0.1}$ | $36.5_{0.0}$ | $52.0_{0.0}$ | $80.0_{0.0}$ | $60.7_{0.2}$ | 57.1 |
| | ICL | $83.0_{8.5}$ | $59.8_{4.2}$ | $72.0_{1.7}$ | $71.6_{2.4}$ | $38.1_{2.3}$ | $53.6_{2.2}$ | $84.0_{2.9}$ | $63.2_{0.8}$ | 65.6 |
| ZO | Sens. (C4, static) | $94.1_{0.6}$ | $75.1_{2.1}$ | $69.0_{2.2}$ | $72.0_{1.0}$ | $59.0_{2.0}$ | $60.9_{0.7}$ | $88.7_{0.5}$ | $65.4_{0.7}$ | 73.0 |

## D.2  COMMONSENSE REASONING, MATH, MMLU DATASET RESULTS.

In Table 5, we still compare SensZOQ vs. ZO full FT & ICL in standard commonsense benchmarks (Hu et al., 2023; Yang et al., 2024), 1 math algebraic word problem task AQuA (Ling et al., 2017), and MMLU (Hendrycks et al., 2021). **To the best of our knowledge, there are no ZO-LLM research yet evaluated on harder commonsense reasoning or math tasks. We take a pioneering step in this direction and establish the ZO baselines.**

If we compare SensZOQ with ZO full FT on pairs, SensZOQ wins 6/8 for commonsense reasoning, and the math task. However, SensZOQ loses in MMLU task, and we speculate it might be due to a data distribution mismatch between C4 and education domain in MMLU. In Table 6 we find that switching the source of extracting sensitive parameters from C4 to OpenWebMath will significantly close this gap. *Even so, C4 is still a generally good choice.*

## D.3  ALTERNATIVE DATA SOURCES FOR EXTRACTING SENSITIVE PARAMETERS.

C4 (Raffel et al., 2019) is extracted and cleaned from the CommonCrawl and covers a wide range of domains, and this drives us to adopt it as the default choice for extracting sensitive parameters. Here, we evaluate the SensZOQ's performance when fine-tuning sensitive parameters extracted from some alternative text choices that have certain domain specialties. As a case study, we pick 3 alternative choices that are more domain-specific and also commonly selected for pre-training dataset mixtures (e.g. Dolma (Soldaini et al., 2024)) as follows:

Table 5: Fine-tuning performance of different methods for **Mistral-7B** on 8 commonsense reasoning tasks (*cs*), 1 math task (*math*), and MMLU task. For each cell, we report the mean and standard deviation of validation set accuracy ($\uparrow$) of 3 random trials in the format of mean$_{std}$. **We finally report the average accuracy across tasks in the last column.**

| Methods | Arc-E (*cs*) | Arc-C (*cs*) | HS (*cs*) | OBQA (*cs*) | PIQA (*cs*) | SIQA (*cs*) | BoolQ (*cs*) | WinoG (*cs*) | AQuA (*math*) | MMLU | Avg |
|---|---|---|---|---|---|---|---|---|---|---|---|
| **SensZOQ** | $88.9_{0.2}$ | $79.8_{0.9}$ | $\mathbf{83.6}_{0.5}$ | $77.5_{0.7}$ | $\mathbf{84.9}_{0.9}$ | $69.4_{1.0}$ | $76.3_{1.0}$ | $\mathbf{73.6}_{0.7}$ | $26.1_{0.7}$ | $58.4_{0.0}$ | **71.9** |
| ZO Full FT | $89.6_{0.4}$ | $78.3_{1.5}$ | $82.3_{0.6}$ | $76.8_{0.3}$ | $84.3_{0.3}$ | $68.5_{0.4}$ | $\mathbf{76.6}_{0.2}$ | $72.2_{0.5}$ | $24.0_{1.4}$ | $\mathbf{59.2}_{0.1}$ | 71.2 |
| Zero-shot | $86.8_{0.0}$ | $75.9_{0.0}$ | $77.9_{0.8}$ | $71.0_{0.0}$ | $82.1_{0.3}$ | $59.9_{0.5}$ | $43.4_{1.8}$ | $66.2_{0.1}$ | $23.5_{1.9}$ | $57.5_{0.0}$ | 64.4 |
| ICL | $\mathbf{90.5}_{0.2}$ | $\mathbf{80.0}_{2.0}$ | $80.3_{1.4}$ | $\mathbf{79.8}_{0.7}$ | $84.5_{0.9}$ | $\mathbf{69.9}_{1.0}$ | $46.8_{6.5}$ | $74.0_{0.8}$ | $26.61.1$ | $\mathbf{59.2}_{0.2}$ | 69.2 |

- ArXiv (Cohan et al., 2018), a pile of scientific papers. We use the ArXiv articles subset from this dataset. `https://huggingface.co/datasets/armanc/scientific_papers`
- OpenWebMath (Paster et al., 2024), a pile of Internet mathematical proofs. `https://huggingface.co/datasets/open-web-math/open-web-math`
- Wiki103 (Merity et al., 2016), a pile of selected Wikipedia articles. `https://huggingface.co/datasets/Salesforce/wikitext`

Table 6: Fine-tuning performance of SensZOQ with **0.1% sensitive parameters extracted from C4, OpenWebMath, ArXiv, and Wiki103** for **Mistral-7B** on 8 commonsense reasoning tasks, 1 math task, MMLU task, and 3 SuperGLUE tasks (task category follows this order and is separated by vertical bars). For each cell, we report the mean accuracy ($\uparrow$) over 3 random trials. In the last row, we give ZO Full FT & ICL baselines as reference. **We finally report the average accuracy across tasks in the last column.**

| Source | Arc-E | Arc-C | HS | OBQA | PIQA | SIQA | BoolQ | WinoG | AQuA | MMLU | RTE | WiC | COPA | Avg |
|---|---|---|---|---|---|---|---|---|---|---|---|---|---|---|
| **C4** | 88.9 | 79.8 | **83.6** | 77.5 | **84.9** | 69.4 | 76.3 | **73.6** | 26.1 | 58.4 | 76.8 | 62.2 | 87.7 | **72.7** |
| ArXiv | 89.1 | 76.0 | 82.4 | 76.3 | 84.3 | 69.1 | 76.0 | **73.6** | **26.9** | 58.4 | 70.0 | 61.6 | **88.7** | 71.7 |
| OpenWebMath | 89.5 | 79.9 | 82.7 | 75.7 | 83.1 | 67.4 | 73.8 | 73.2 | **26.9** | 58.8 | **77.9** | 61.1 | 88.0 | 72.2 |
| Wiki103 | 88.5 | 77.6 | 83.0 | 77.6 | 84.6 | 67.8 | 76.4 | 73.2 | 26.5 | 58.6 | 75.8 | 61.8 | 88.3 | 72.3 |
| ZO Full FT | 89.6 | 78.3 | 82.3 | 76.8 | 84.3 | 68.5 | **76.6** | 72.2 | 24.0 | **59.2** | 74.6 | 62.6 | 88.3 | 72.1 |
| ICL | **90.5** | **80.0** | 80.3 | **79.8** | 84.5 | **69.9** | 46.8 | 74.0 | 26.6 | **59.2** | 55.2 | **63.8** | 74.0 | 68.0 |

In Table 6, we can find that when fine-tuning 0.1% sensitive parameters with Mistral-7B, **C4 achieves the highest average accuracy**, with notable performance on commonsense QA tasks like OBQA, PIQA and SIQA. If we want better performance on an expert-level knowledge task such as MMLU, OpenWebMath is a better choice than C4.

We believe that C4 is *not the only* choice but rather a *generally good* choice for downstream tasks, and the latter is more desired as *we are using the same set of sparse parameters for different tasks* and we want it to produce satisfactory performance for as many tasks as possible rather than focusing on excellent performance on a single task. Otherwise, we would have to create a separate sparse mask and quantized models for each task, and this will make our method impractical.

We also believe that a high-quality and diverse pre-training dataset mixture such as Pile (Gao et al., 2020) and Dolma (Soldaini et al., 2024) should also perform well, and we leave this study to future research.

# E  SENSITIVE PARAMETERS IN LLMS

## E.1  GRADIENT SPARSITY DURING LLM FINE-TUNING

**SuperGLUE tasks.**   In Figure 2, we explore the gradient sparsity of Llama2-7B during FO-SGD fine-tuning (at epoch 1 or 10% training steps). Here we plot the FO-SGD gradient sparsity for Llama2-7B, Mistral-7B, and OPT-6.7B before fine-tuning, epoch 1 (10% steps), and end of fine-tuning.

We observe that the gradient sparsity is exhibited throughout the fine-tuning and increases toward the end. OPT-6.7B, which uses ReLU as the activation function, demonstrates greater sparsity across tasks compared to Llama2-7B and Mistral-7B which both use SiLU. Overall, the gradient sparsity pattern is generally consistent across architectures, tasks, and fine-tuning time.

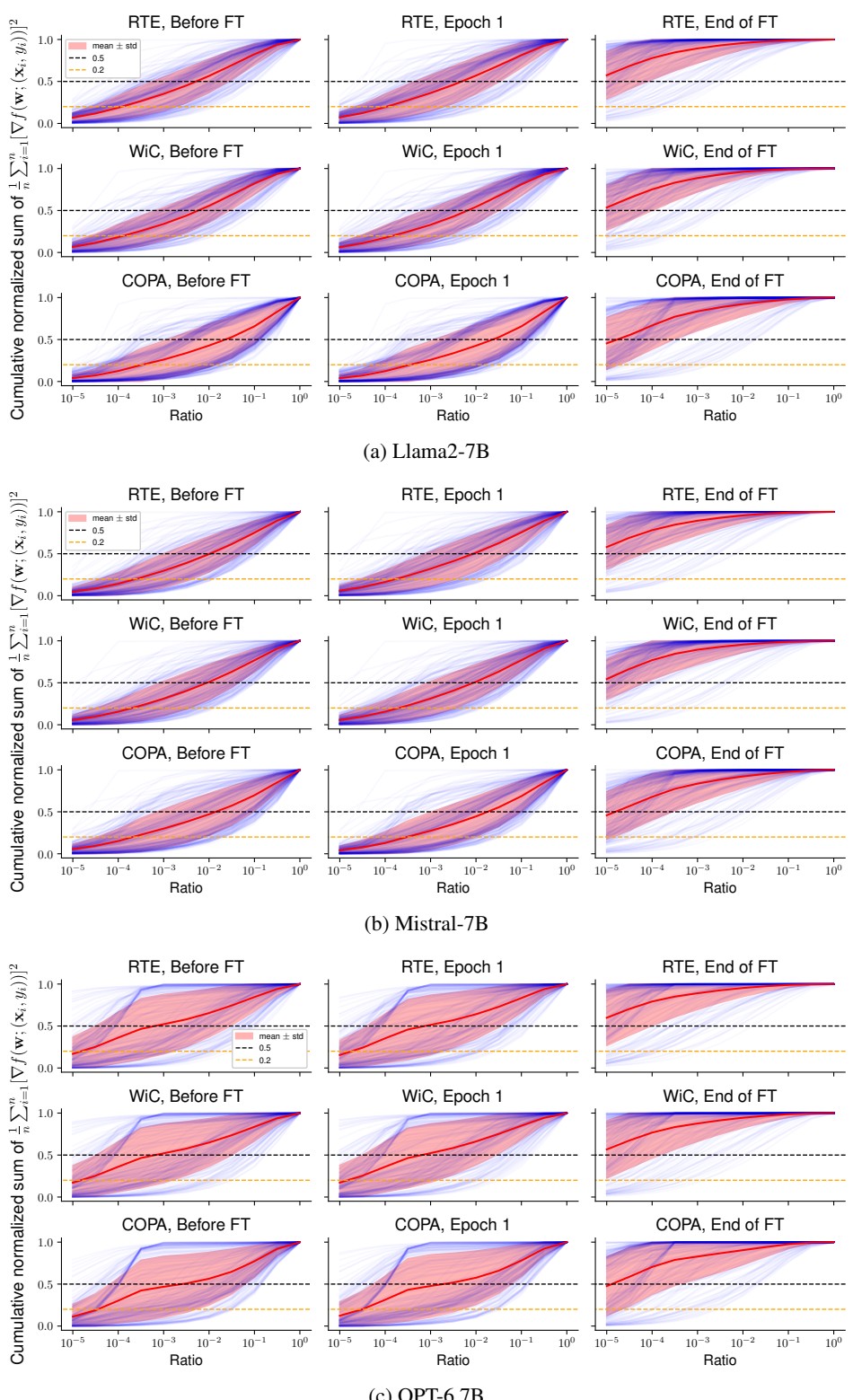

(a) Llama2-7B

(b) Mistral-7B

(c) OPT-6.7B

Figure 7: Cumulative normalized sum of coordinate-wise gradient square $\frac{1}{n} \sum_{i=1}^{n} [\nabla f(\mathbf{w}; (\mathbf{x}_i, y_i))]^2$ of linear layers for Llama2-7B (subfigure 7a), Mistral-7B (subfigure 7b), and OPT-6.7B (subfigure 7c) across RTE, WiC, and COPA tasks during FO-SGD full FT for 10 epochs. For each linear layer, we first sort parameters by the decreasing order of their gradient square value $\frac{1}{n} \sum_{i=1}^{n} [\nabla f(\mathbf{w}; (\mathbf{x}_i, y_i))]^2, i \in [d_{\text{layer}}]$, and we take the cumulative sum and normalize it to draw a blue curve, and the red-shaded region is the mean $\pm$ std of all blue curves.

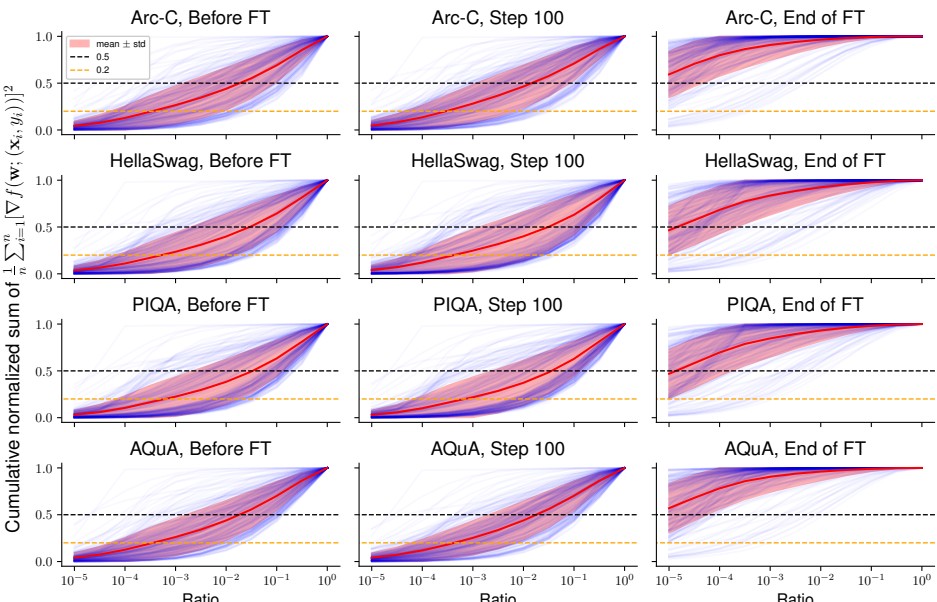

Figure 8: Cumulative normalized sum of coordinate-wise gradient square $\frac{1}{n}\sum_{i=1}^{n}[\nabla f(\mathbf{w};(\mathbf{x}_i,y_i))]^2$ of linear layers for **Mistral-7B** across Arc-C, HS, PIQA, and AQuA tasks during FO-Adam full FT for 1000 steps.

**Reasoning tasks.** We further analyze the sparsity of sensitive parameters for hard reasoning tasks such as Arc-C, HellaSwag, PIQA, and math reasoning task AQuA. We follow the same method that produced Figure 7b to produce Figure 8. We can see that sensitive parameters are still fairly sparse: we still need 0.1%-1% sensitive parameters for a sufficient coverage of the gradient norm.

## E.2 TRANSFERABILITY OF GRADIENT FEATURES FROM PRE-TRAINING DATASETS TO DOWNSTREAM TASKS

**SuperGLUE tasks.** In Figure 3, we explore the transferability of gradient features from pre-training datasets (C4) to downstream tasks, and here we will also validate this phenomenon across models, as shown in Figure 9. As there are *no* solid lines (top-(1e-2,1e-3,1e-4)) parameters with C4 gradient entries prior to fine-tuning) vanish to 0, we know the transferability of gradient features from C4 datasets to downstream datasets hold across models and tasks. We also include the results of weights with largest magnitude, weights with smallest magnitude, and random subsets of weights in Figure 10 at 1e-3 nnz level (same as 1e-3 in Figure 9) for comparison. It is clear that "C4 grad (static)" has much higher similarity with "task grad (dyn.)" than all of these 3 baselines. *We also note that weights with largest magnitude (weight outliers) are usually NOT gradient outliers as observed in Figure 10.*

In this case, sensitive parameters determined from C4 gradients would still be similar to sensitive parameters determined from downstream task-specific gradients across models.

**Reasoning tasks.** The transferability results for reasoning tasks are shown in Figure 11.

As there are still no solid lines (for all top-(1e-2,1e-3,1e-4) parameters with C4 mask) vanish to 0, C4 gradient mask still demonstrates great transferability to these 3 commonsense and 1 math task. At step 100, the lowest covered gradient norm for top-1e-3 C4 gradient mask is ∼0.2, while the maximum possible (*task grad, dyn.*) is ∼0.3 across tasks.

We still note that for AQuA (math algebraic word problem task), C4's transferability is weaker than the other 3 commonsense reasoning tasks. We speculate that this is due to the need of learning more math knowledge during FT as the covered task gradient squares by task gradient mask before FT mask (*task grad, static*) also declines more during FT.

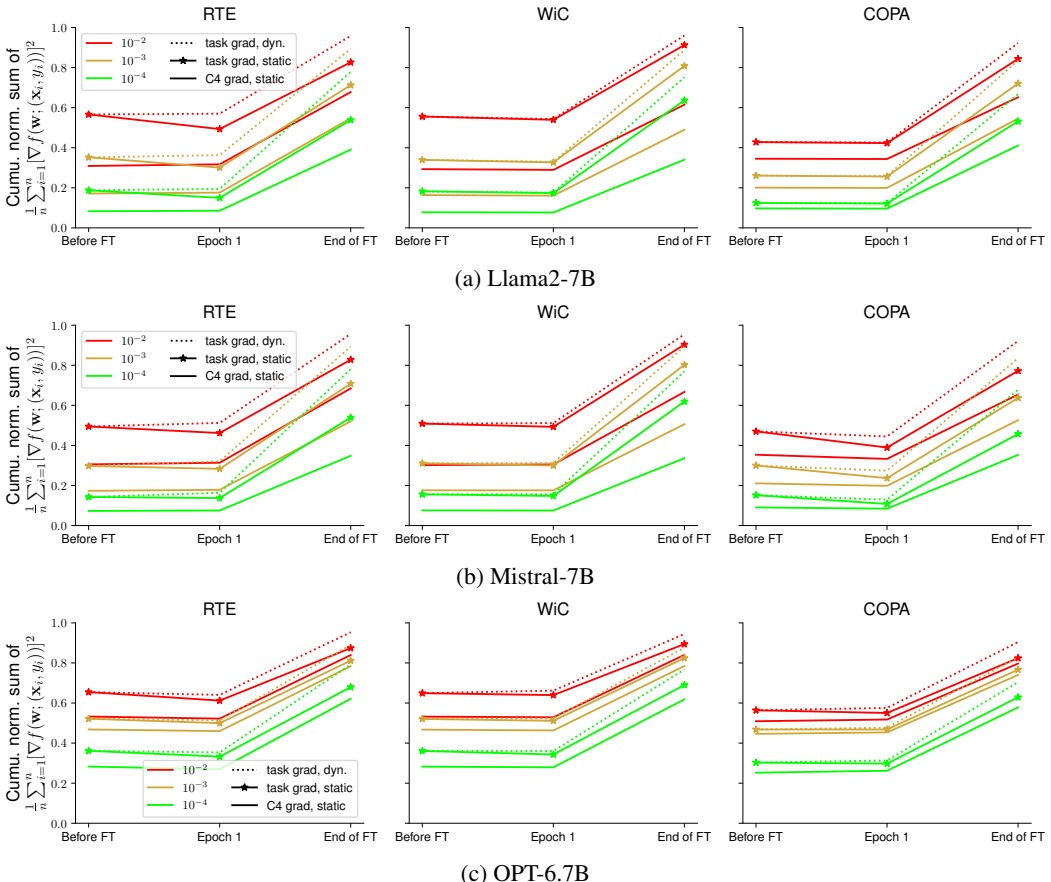

Figure 9: Cumulative normalized sum of coordinate-wise $\frac{1}{n}\sum_{i=1}^{n}[\nabla f(\mathbf{w};(\mathbf{x}_i, y_i))]^2$ of Llama2-7B (subfigure 9a), Mistral-7B (subfigure 9b), and OPT-6.7B (subfigure 9c)'s linear layers **after applying sparsity masks of each method** during FO-SGD full fine-tuning for 10 epochs. For a given model and training checkpoint, we report the average value across all linear layers as a line in each subfigure. For each line, the colors represent the fraction of parameters (1e-2,1e-3,1e-4) and the line style represents the category. "task grad, dyn." refers to the sensitive parameters selected at the given timestep (x-axis), and "task grad, static" refers to the sensitive parameters selected before fine-tuning. "C4 grad, static" refers to the sensitive parameters selected with gradients taken from causal language modeling on C4 datasets, and we keep it unchanged during fine-tuning.

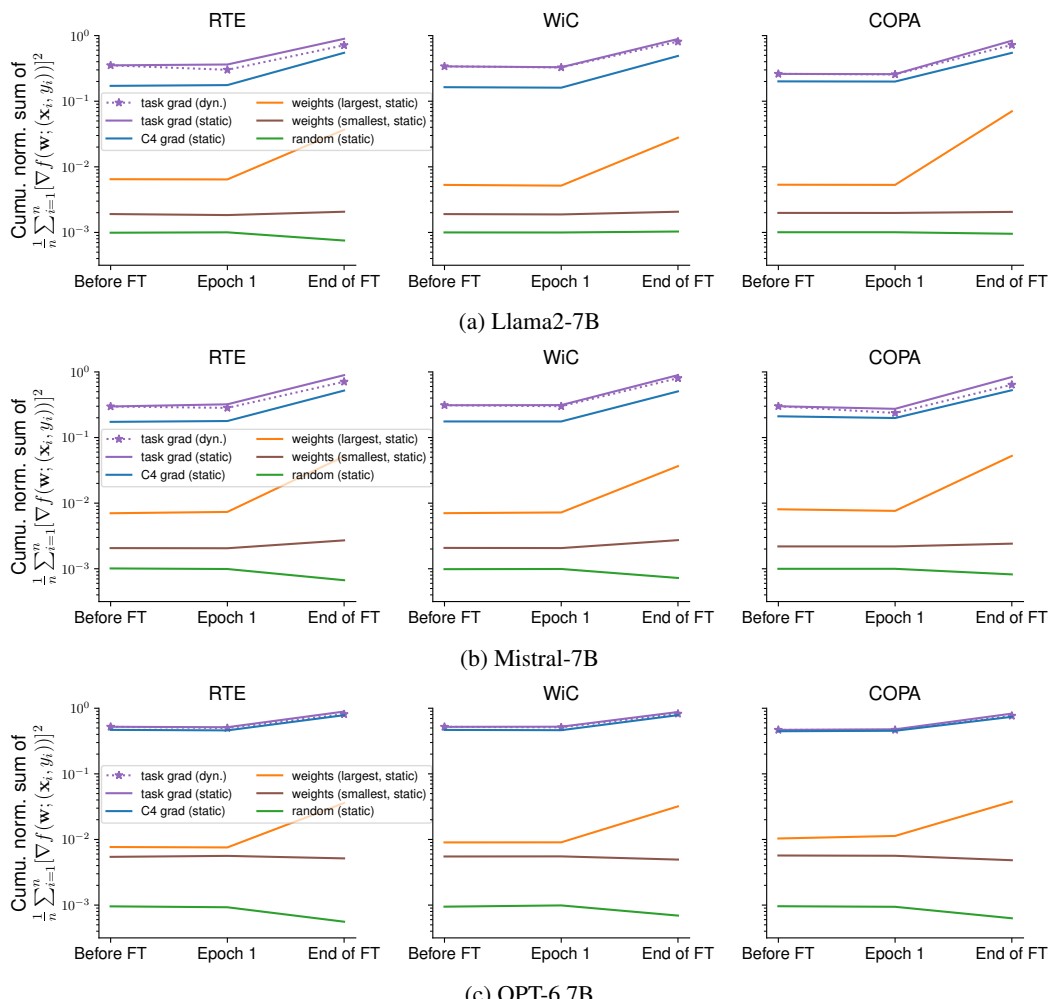

Figure 10: Cumulative normalized sum of coordinate-wise $\frac{1}{n}\sum_{i=1}^{n}[\nabla f(\mathbf{w};(\mathbf{x}_i,y_i))]^2$ of Llama2-7B (subfigure 10a), Mistral-7B (subfigure 10b), and OPT-6.7B (subfigure 10c)'s linear layers **after applying 99.9% sparsity masks of each method** during FO-SGD full fine-tuning. The results of "C4 grad, static", "task grad, dyn.", and "task grad, static" are the same as their 1e-3 results in Figure 9.

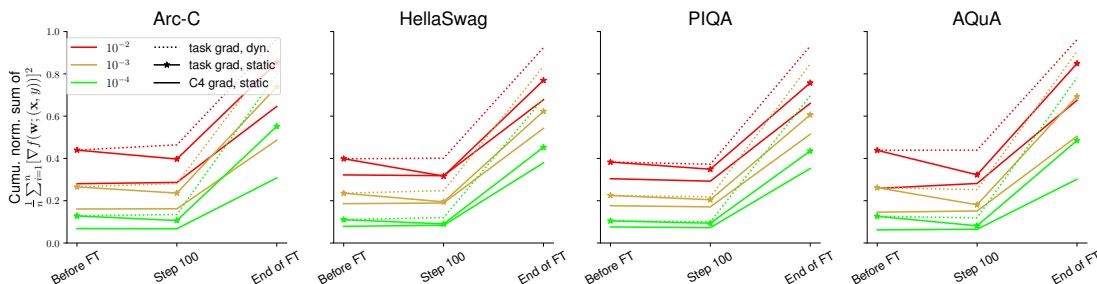

Figure 11: Cumulative normalized sum of coordinate-wise $\frac{1}{n}\sum_{i=1}^{n}[\nabla f(\mathbf{w};(\mathbf{x}_i,y_i))]^2$ of Mistral-7B after applying sparsity masks of each method during FO-Adam full FT for 1000 steps.

### E.3 OVERLAP RATIO OF TOP GRADIENT FEATURES

In addition to Table 6, we dive into the overlap ratio of the top gradient square entries from different data sources, as shown in Figure 12.

Notice that the order here is mapped to the column/row rank for each subfigure in Figure 12.

1. **C4**. SensZOQ's default choice.

2. OpenWebMath.
   We abbreviate it as "Math".

3. ArXiv

4. Wiki103
   We abbreviate it as "Wiki".

5. Task gradient before FT: $\nabla \frac{1}{n} \sum_{i=1}^{n} [\nabla f(\mathbf{w}_{\text{before FT}}; (\mathbf{x}_i, y_i))]^2$.
   We abbreviate it as $\nabla f(\mathbf{w}_{\text{st.}})$.

6. Task gradient after 10% fine-tuning steps: $\nabla \frac{1}{n} \sum_{i=1}^{n} [\nabla f(\mathbf{w}_{\text{mid FT}}; (\mathbf{x}_i, y_i))]^2$.
   We abbreviate it as $\nabla f(\mathbf{w}_{\text{mid}})$.

7. Task gradient at the end of FT: $\nabla \frac{1}{n} \sum_{i=1}^{n} [\nabla f(\mathbf{w}_{\text{end of FT}}; (\mathbf{x}_i, y_i))]^2$.
   We abbreviate it as $\nabla f(\mathbf{w}_{\text{end}})$.

We consider 7 tasks (3 commonsense reasoning tasks Arc-C, HellaSwag, PIQA, 1 math task AQuA, 3 SuperGLUE tasks RTE, WiC, COPA).

For a quick overview, we include Figure 13 as the average overlap ratio across tasks for the 4 pre-training text. The second row in Figure 13 provides an empirical evidence for the "fixed gradient feature" during FT as the top entries in $\nabla f(\mathbf{w}_{\text{st.}})$ (task grad before FT) resemble $\nabla f(\mathbf{w}_{\text{mid}})$ (task grad during FT) and $\nabla f(\mathbf{w}_{\text{end}})$ (task grad at the end of FT)

**Empirical findings**

- The top gradient entries from all 4 pre-training text corpus overlap considerably with the task gradient.
- C4 generally covers more top gradient entries than OpenWebMath, ArXiv, and Wiki103.

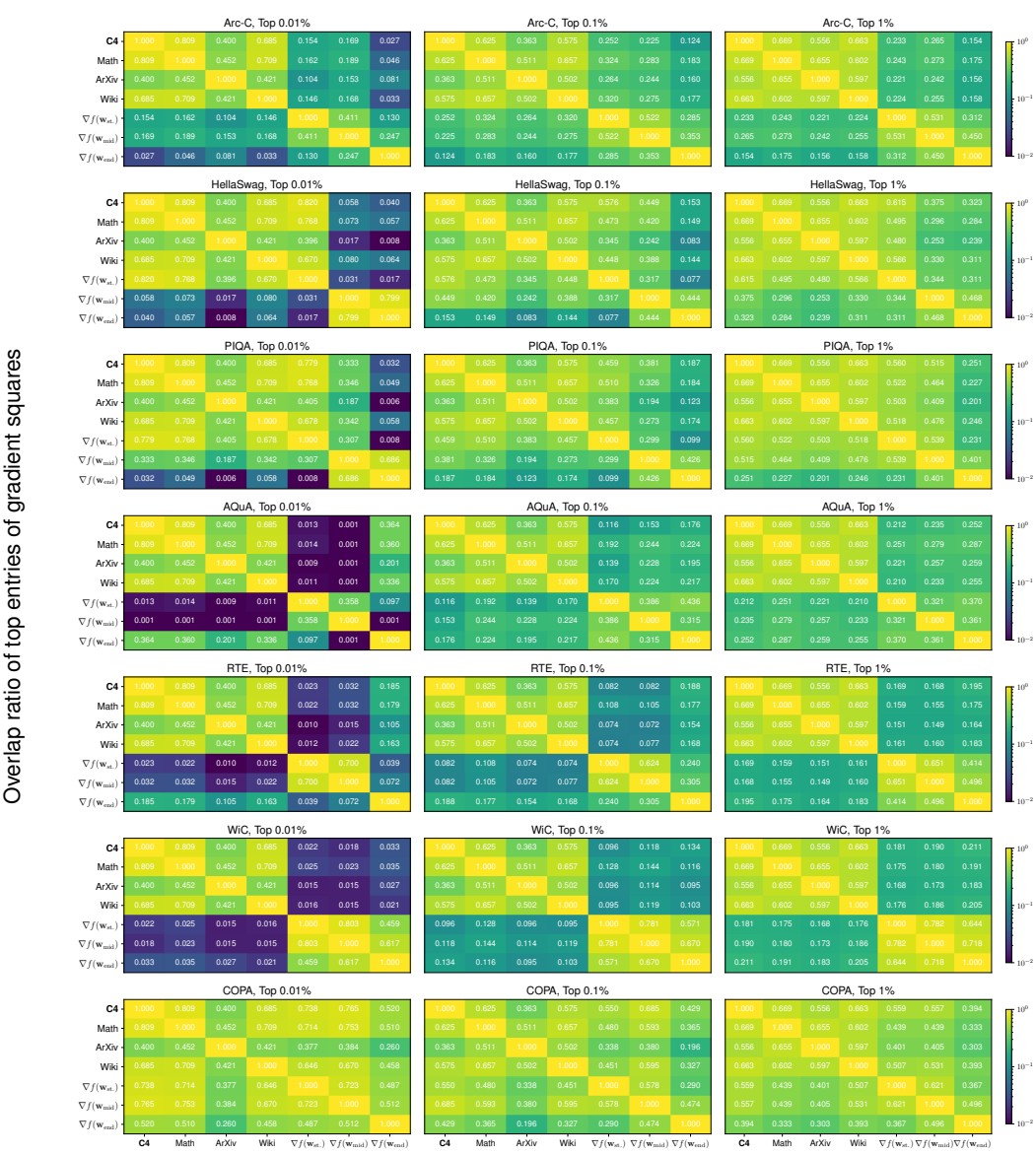

Figure 12: Overlap ratio in top entries of gradient squares of **Mistral-7B** for C4, OpenWebMath, ArXiv, Wiki103, and $\nabla f(\mathbf{w}_{\text{st.}})$, $\nabla f(\mathbf{w}_{\text{mid}})$, and $\nabla f(\mathbf{w}_{\text{end}})$ across 3 commonsense reasoning, 1 math, and 3 SuperGLUE tasks.

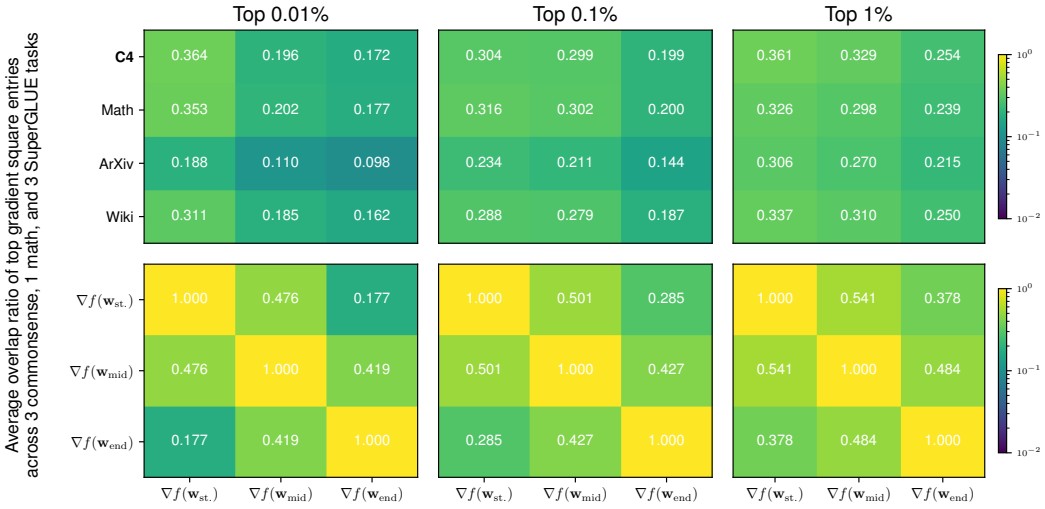

Figure 13: Average overlap ratio of top gradient square entries across 7 tasks in Figure 12.

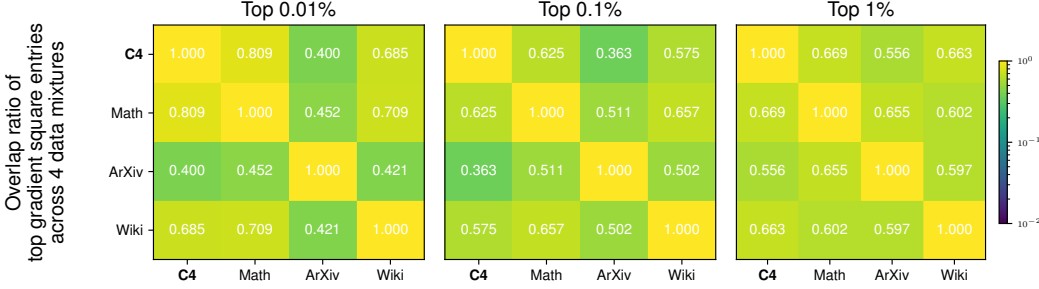

Figure 14: Overlap ratio of top gradient square entries among C4, OpenWebMath, ArXiv, and Wiki103 in Figure 12.

## F    IMPLEMENTATION OF SPARSE OPERATIONS IN LINEAR LAYERS

Linear layers in LLMs often contribute most parameters (Kaplan et al., 2020). Since from Equation 7 we know

$$\mathbf{w}_{\text{sparse}} = \mathbf{w} \odot \mathbf{m}_k, \quad \mathbf{w}_{\text{dense}} = \mathbf{w} \odot (\mathbf{1}_d - \mathbf{m}_k), \quad \mathbf{w} = \mathbf{w}_{\text{sparse}} + \mathbf{w}_{\text{dense}} \tag{13}$$

and since $\mathbf{w}_{\text{dense}}$ would have the same shape (and the same computational intensities) as $\mathbf{w}$, we need to improve *wall-clock time* efficiency of $\mathbf{w}_{\text{sparse}}\mathbf{x}$ to improve the computational efficiency of linear layers after extracting the sparse parameters. In this case, we would have two different methods to implement the forward pass of linear layers (with induced sparse operation colored in red):

$$
\begin{align}
\mathbf{wx} &= \mathbf{w}_{\text{dense}}\mathbf{x} + \mathbf{w}_{\text{sparse}}\mathbf{x} \tag{14} \\
&= \text{SparseAddMM}(\text{DenseMM}(\mathbf{w}_{\text{dense}}, \mathbf{x}), \mathbf{w}_{\text{sparse}}, \mathbf{x}) \quad \text{faster with token generation} \tag{15} \\
&= (\mathbf{w}_{\text{dense}} + \mathbf{w}_{\text{sparse}})\mathbf{x} \tag{16} \\
&= \text{DenseMM}(\text{SparseAdd}(\mathbf{w}_{\text{sparse}}, \mathbf{w}_{\text{dense}}), \mathbf{x}) \quad \text{faster with ZO training} \tag{17}
\end{align}
$$

```python
# PyTorch's F.linear uses Y = X W^T + b.
# Here we use Y = W X + b for simplicity.
class SensitiveZOLinear(Linear):
    w_sparse  # in FP16, CSR format
    w_dense   # in FP16, dense tensor
    bias      # in FP16, dense tensor

    def forward_small_batched_decoding(self, X):
        # dense matmul (linear forward)
        dense_result = dense_linear(self.w_dense, X, self.bias)
        # sparse addmm
        return sparse_addmm(dense_result, self.w_sparse, X) + self.bias

    def forward_large_batched_ZO_training(self, X):
        # add sparse weights to the dense weights
        w         = self.w_dense + self.w_sparse
        # dense matmul (linear forward)
        return dense_linear(w, X, self.bias)
```

Listing 1: Example PyTorch-like code snippet that implements the forward calls with FP16 sparse and FP16 dense parameters.

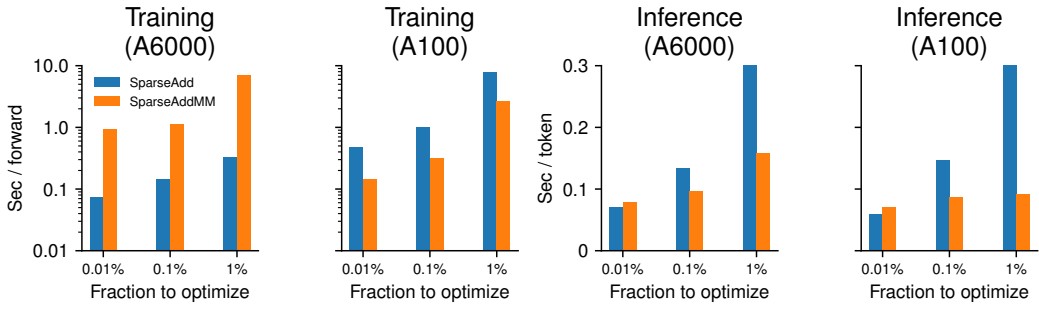

Figure 15: Time of SparseAdd (Equation 17) versus SparseAddMM (Equation 15) in Llama2-7B ZO training forward & inference. In subfigure 1 and 3, we use Nvidia RTX A6000 and Intel Xeno Gold 6342 CPUs, with PyTorch version 2.2, HuggingFace version 4.36, and CUDA 12.2. In subfigure 2 and 4, we use Nvidia A100-SXM4 (40 GB) and AMD EPYC 7543P 32-Core CPU with PyTorch version 2.1, HuggingFace version 4.38.2, and CUDA 12.2. We use Flash Attention 2 (Dao, 2023) for all 4 subfigures.

The specific choice of employing Equation 15 or Equation 17 needs benchmarking, and we also provide a general guideline based on the size of input vectors $\mathbf{x}$ and the resulting arithmetic intensity.

**Size of input vectors $\mathbf{x}$ and arithmetic intensity.** $\mathbf{w}_{\text{sparse}}\mathbf{x}$ in Equation 15 would have a computational dependency over $\mathbf{x}$. During large-batched ZO training, $\mathbf{x}$ would be large enough such that Equation 15 would induce large computational overhead, as shown in subfigure 1 of Figure 15. In contrast, the computational costs of Equation 17 is *independent* of $\mathbf{x}$ and when $\mathbf{x}$ is large, we would expect Equation 17 is *much* faster than Equation 15. As an example, we use sequence length of 512 and batch size 16 sampled from WikiText-2 dataset (Merity et al., 2016) as a representative forward costs for ZO training in subfigures 1 and 2 in Figure 15.

However, during autoregressive token generation, on each step we would only append *a single token* to the previously cached embeddings, and in this case $\mathbf{x}$ is small and computing $\mathbf{w}_{\text{dense}} + \mathbf{w}_{\text{sparse}}$ is generally not worthwhile, especially given that $\mathbf{w}_{\text{sparse}}$ is already sparse. This is also illustrated in subfigure 3 and 4 in Figure 15. However, we note that the specific implementation choice is hardware and task dependent and requires thorough benchmarking and we will leave it as a future work.

**We recommend using Equation 17 during large-batched ZO training and Equation 15 during small-batched autoregressive token generation.**

**Integration with weight quantization method.** Weight integer quantization algorithms can be categorized into 2 categories: uniform integer quantization method and non-uniform integer quantization method. For uniform integer quantization method, we use integer matrix multiplication with fused scaling to compute $Q(\mathbf{w}_{\text{dense}})Q(\mathbf{x})$ efficiently *without* first dequantizing $Q(\mathbf{w}_{\text{dense}})$ to FP16 (Xi et al., 2023; Park et al., 2024). However, this is not compatible with our "SparseAdd" approach as we will *violate the constraint of uniformly-spaced int quantization bins* by computing $\text{SparseAdd}(Q(\mathbf{w}_{\text{dense}}) + \mathbf{w}_{\text{sparse}})$. In this case, we also have 3 different implementations:

$$Q(\mathbf{w})\mathbf{x} \sim Q(\mathbf{w}_{\text{dense}})Q(\mathbf{x}) + \mathbf{w}_{\text{sparse}}\mathbf{x} \tag{18}$$

$$= \text{SparseAddMM}\Big(\text{ScaledIntMM}(Q(\mathbf{w}_{\text{dense}}), Q(\mathbf{x})), \mathbf{w}_{\text{sparse}}, \mathbf{x}\Big) \quad \text{uniform int Q} \tag{19}$$

$$\sim Q(\mathbf{w}_{\text{dense}})x + \mathbf{w}_{\text{sparse}}\mathbf{x} \tag{20}$$

$$= \text{SparseAddMM}\Big(\text{Dequantize}(Q(\mathbf{w}_{\text{dense}})), \mathbf{x}, \mathbf{w}_{\text{sparse}}\Big) \quad \text{similar to Equation 15} \tag{21}$$

$$\sim \big(Q(\mathbf{w}_{\text{dense}}) + \mathbf{w}_{\text{sparse}}\big)\mathbf{x} \tag{22}$$

$$= \text{DenseMM}\big(\text{SparseAdd}\left(\mathbf{w}_{\text{sparse}}, \text{Dequantize}(Q(\mathbf{w}_{\text{dense}}))\right), \mathbf{x}\big) \quad \text{similar to Equation 17} \tag{23}$$

Equation 19 would compute $\text{ScaledIntMM}\big(Q(\mathbf{w}_{\text{dense}}), Q(\mathbf{x})\big)$ without dequantizing $Q(\mathbf{w}_{\text{dense}})$ to FP16. Notice that both Equation 21 and Equation 23 would still dequantize $Q(\mathbf{w}_{\text{dense}})$ first and the choice of implementation would follow into our discussion of input vector size $\mathbf{x}$ in last paragraph. We leave a practical implementation and thorough benchmarking into a future work.

```python
class SensZOQ(Linear):
    w_sparse            # in FP16, CSR format
    w_dense_quantized   # in Int4, dense tensor
    bias                # in FP16, dense tensor

    def dequantize(self, w_quantized):
        ...
        return w_16_bit

    def forward_scaled_int_matmul(self, X):
        # dense int4 matmul with fused scaling
        dense_result  = scaled_int_matmul(self.w_dense_quantized, X)
        # sparse addmm
        return sparse_addmm(dense_result, self.w_sparse, X) + self.bias

    def forward_small_batched_decoding(self, X):
```

```
17          # dequantize the dense weights to FP16
18          w_dense_16_bit = self.dequantize(self.w_dense_quantized)
19          # dense FP16 linear forward
20          dense_result   = dense_linear(w_dense_16_bit, X, self.bias)
21          # sparse addmm
22          return sparse_addmm(dense_result, self.w_sparse, X) + self.bias
23
24      def forward_large_batched_ZO_training(self, X):
25          # dequantize the dense weights to FP16
26          w_dense_16_bit = self.dequantize(self.w_dense_quantized)
27          # add sparse weights to the dense weights
28          w               = w_dense_16_bit + self.w_sparse
29          # dense FP16 linear forward
30          return dense_linear(w, X, self.bias)
```

Listing 2: Example PyTorch-like code snippet that implements the forward calls with 16-bit sparse and quantized dense parameters.

**We recommend using Equation 19 when we use integer matmul to compute $Q(\mathbf{w_{dense}})Q(\mathbf{x})$ and in other cases, using Equation 21 or Equation 23 follows our previous recommendation.**

# G  SUPPLEMENTARY EXPERIMENT DETAILS

## G.1  HYPERPARAMETERS IN EXPERIMENTS

For all ZO experiments, we use 20,000 training steps with ZO-SGD optimizer (Definition 2). We evaluate on the validation or test set at the end of the training. Usually, the training/validation set will be sampled from the original training dataset with size 1000/500 respectively and the evaluation set is of size $\min(1000, |\text{original validation or test set}|)$. However, for CB and COPA, we use 100 for the validation set size. We use 2051/200 for Arc-E, 919/200 for Arc-C, 39705/200 for HellaSwag, 15913/200 for PIQA, 4757/200 for OBQA, 33210/200 for SIQA, 20000/200 for MMLU (training is on an auxiliary training set), and 97267/200 for AQuA.

For all ZO experiments in Table 7, 8, and 9, we use a batch size of 16 except for the Mistral-7B on MMLU experiment in Table 8 we use a batch size of 8 for all methods.

Table 7: The chosen hyperparameters (ZO perturbation constant, learning rates, and # ICL examples) for experiments in Table 1 and 4. We repeat each learning rate for 3 random trials and report the average and standard deviation in Table 1 and 4.

(a) **Llama2-7B**

|  | Methods | SST-2 | RTE | CB | BoolQ | WSC | WiC | COPA | WinoG |
|---|---|---|---|---|---|---|---|---|---|
| Q, ZO, SGD | **SensZOQ** ($\epsilon$ =1e-3) | 5e-7 | 1e-6 | 1e-6 | 1e-6 | 5e-7 | 1e-6 | 1e-6 | 2e-6 |
|  | LoRA ($\epsilon$ =1e-3) | 1e-5 | 2e-5 | 1e-5 | 2e-5 | 1e-5 | 2e-5 | 1e-5 | 5e-5 |
| ZO, SGD | Full FT ($\epsilon$ =1e-3) | 5e-7 | 5e-7 | 5e-7 | 5e-7 | 2e-7 | 5e-7 | 5e-7 | 1e-7 |
|  | ICL (#examples) | 16 | 16 | 16 | 8 | 16 | 8 | 8 | 16 |
| FO, SGD | Full FT | 1e-4 | 1e-4 | 2e-5 | 5e-5 | 5e-5 | 5e-5 | 5e-5 | 1e-4 |
| FO, Adam | Full FT | 1e-5 | 5e-6 | 2e-5 | 1e-5 | 5e-6 | 5e-6 | 2e-6 | 2e-5 |

(b) **Mistral-7B**

|  | Methods | SST-2 | RTE | CB | BoolQ | WSC | WiC | COPA | WinoG |
|---|---|---|---|---|---|---|---|---|---|
| Q, ZO, SGD | **SensZOQ** ($\epsilon$ =1e-4) | 2e-8 | 5e-8 | 2e-8 | 2e-8 | 1e-8 | 2e-8 | 2e-8 | 1e-7 |
|  | LoRA ($\epsilon$ =1e-4) | 2e-6 | 5e-6 | 2e-6 | 2e-6 | 2e-6 | 2e-6 | 2e-6 | 1e-5 |
| ZO, SGD | Full FT ($\epsilon$ =1e-4) | 2e-8 | 2e-8 | 1e-8 | 1e-8 | 1e-8 | 1e-8 | 2e-8 | 5e-8 |
|  | ICL (#examples) | 4 | 8 | 4 | 16 | 4 | 4 | 8 | 8 |
| FO, SGD | Full FT | 2e-6 | 5e-6 | 1e-5 | 2e-6 | 1e-5 | 2e-6 | 1e-5 | 5e-6 |
| FO, Adam | Full FT | 2e-6 | 2e-6 | 2e-6 | 1e-6 | 2e-6 | 2e-6 | 2e-6 | 2e-6 |

(c) **OPT-6.7B**

|  | Methods | SST-2 | RTE | CB | BoolQ | WSC | WiC | COPA | WinoG |
|---|---|---|---|---|---|---|---|---|---|
| Q, ZO | **SensZOQ** ($\epsilon$ =1e-3) | 2e-7 | 5e-7 | 5e-7 | 5e-7 | 2e-7 | 5e-7 | 2e-7 | 1e-6 |
|  | LoRA ($\epsilon$ =1e-3) | 1e-5 | 2e-5 | 1e-5 | 2e-5 | 1e-5 | 2e-5 | 2e-5 | 5e-5 |
| ZO, SGD | Full FT ($\epsilon$ =1e-3) | 2e-7 | 2e-7 | 2e-7 | 2e-7 | 2e-7 | 2e-7 | 5e-7 | 5e-7 |
|  | ICL (#examples) | 16 | 4 | 16 | 16 | 16 | 8 | 16 | 16 |
| FO, SGD | Full FT | 5e-5 | 2e-5 | 1e-4 | 2e-5 | 1e-5 | 2e-5 | 1e-4 | 5e-5 |
| FO, Adam | Full FT | 5e-6 | 5e-6 | 2e-5 | 1e-5 | 5e-6 | 1e-5 | 2e-6 | 2e-6 |

(d) **Llama2-13B**

|  | Methods | SST-2 | RTE | CB | BoolQ | WSC | WiC | COPA | WinoG |
|---|---|---|---|---|---|---|---|---|---|
| Q, ZO, SGD | **SensZOQ** ($\epsilon$ =1e-3) | 1e-6 | 1e-6 | 1e-6 | 1e-6 | 1e-7 | 1e-6 | 2e-6 | 5e-6 |
| ZO, SGD | Full FT ($\epsilon$ =1e-3) | 5e-7 | 5e-7 | 5e-7 | 5e-7 | 1e-7 | 5e-7 | 5e-7 | 2e-6 |
|  | ICL (#examples) | 16 | 8 | 16 | 8 | 8 | 8 | 16 | 16 |

(e) **OPT-13B**

|  | Methods | SST-2 | RTE | CB | BoolQ | WSC | WiC | COPA | WinoG |
|---|---|---|---|---|---|---|---|---|---|
| Q, ZO, SGD | **SensZOQ** ($\epsilon$ =1e-3) | 2e-7 | 5e-7 | 2e-7 | 2e-7 | 2e-7 | 5e-7 | 2e-7 | 1e-6 |
| ZO, SGD | Full FT ($\epsilon$ =1e-3) | 2e-7 | 5e-7 | 2e-7 | 2e-7 | 1e-6 | 2e-7 | 2e-7 | 5e-7 |
|  | Sens. (C4, static) ($\epsilon$ =1e-3) | 2e-7 | 5e-7 | 2e-7 | 2e-7 | 2e-7 | 5e-7 | 2e-7 | 1e-6 |
|  | ICL (#examples) | 16 | 16 | 16 | 16 | 16 | 16 | 16 | 16 |

For the ZO-LoRA baseline in Table 1, we always add it to all linear layers with $r = 8$ and $\alpha = 16$.

Table 8: The chosen learning rates for experiments in Table 5 and 6. We repeat each learning rate for 3 random trials and report the average and standard deviation. Notice that in Table 6, the learning rates for RTE, WiC, COPA tasks for SensZOQ (OpenWebMath, ArXiv, and Wiki103) are the same as SensZOQ (C4), which are reported in Table 7.

| | Methods | Arc-E | Arc-C | HS | OBQA | PIQA | SIQA | AQuA | MMLU |
|---|---|---|---|---|---|---|---|---|---|
| Q, ZO, SGD | C4 ($\epsilon$ =1e-4) | 5e-9 | 1e-8 | 1e-7 | 2e-8 | 2e-7 | 2e-8 | 5e-8 | 1e-8 |
| | ArXiv ($\epsilon$ =1e-4) | 5e-9 | 1e-8 | 1e-7 | 2e-8 | 2e-7 | 2e-8 | 5e-8 | 1e-8 |
| | OpenWebMath ($\epsilon$ =1e-4) | 5e-9 | 1e-8 | 1e-7 | 2e-8 | 2e-7 | 2e-8 | 5e-8 | 1e-8 |
| | Wiki103 ($\epsilon$ =1e-4) | 5e-9 | 1e-8 | 1e-7 | 2e-8 | 2e-7 | 2e-8 | 5e-8 | 1e-8 |
| ZO, SGD | Full FT ($\epsilon$ =1e-4) | 5e-9 | 5e-9 | 5e-8 | 5e-9 | 5e-8 | 1e-8 | 1e-8 | 5e-9 |
| | ICL (#examples) | 16 | 16 | 16 | 8 | 16 | 16 | 16 | 16 |

Table 9: The chosen learning rates for experiments in Figure 6a and Figure 6b. We repeat each learning rate for 3 random trials and report the average to draw a line in Figure 6a and Figure 6b, and we use Llama2-7B for all experiments. For each subtable, we include the fraction to optimize on its header and report the chosen learning rate on each cell.

(a) **RTE**

| Methods | 1e-5 | 1e-4 | 1e-3 | 1e-2 | 1e-1 |
|---|---|---|---|---|---|
| **Sensitive (C4, static)** ($\epsilon$ =1e-3) | 1e-5 | 1e-6 | 1e-6 | 1e-6 | 1e-6 |
| Sensitive (task-specific, static) ($\epsilon$ =1e-3) | 1e-5 | 1e-6 | 1e-6 | 1e-6 | 1e-6 |
| Sensitive (task-specific, dynamic) ($\epsilon$ =1e-3) | 5e-6 | 1e-6 | 1e-6 | 1e-6 | 5e-7 |
| Random (static) ($\epsilon$ =1e-3) | 2e-2 | 5e-3 | 5e-4 | 5e-5 | 5e-5 |
| Random (dynamic) ($\epsilon$ =1e-3) | 2e-2 | 5e-3 | 2e-4 | 5e-5 | 5e-6 |
| Weights with largest magnitude (static) ($\epsilon$ =1e-3) | 2e-3 | 1e-3 | 2e-4 | 5e-5 | 1e-5 |
| Weights with smallest magnitude (static) ($\epsilon$ =1e-3) | 2e-3 | 5e-4 | 1e-4 | 1e-5 | 1e-6 |
| smallest GraSP scores (static) ($\epsilon$ =1e-3) | 2e-5 | 5e-6 | 2e-6 | 2e-6 | 1e-6 |

(b) **WiC**

| Methods | 1e-5 | 1e-4 | 1e-3 | 1e-2 | 1e-1 |
|---|---|---|---|---|---|
| **Sensitive (C4, static)** ($\epsilon$ =1e-3) | 1e-5 | 2e-6 | 1e-6 | 1e-6 | 1e-6 |
| Sensitive (task-specific, static) ($\epsilon$ =1e-3) | 1e-5 | 2e-6 | 1e-6 | 1e-6 | 1e-6 |
| Sensitive (task-specific, dynamic) ($\epsilon$ =1e-3) | 1e-5 | 2e-6 | 1e-6 | 1e-6 | 1e-6 |
| Random (static) ($\epsilon$ =1e-3) | 2e-2 | 5e-3 | 5e-4 | 5e-5 | 5e-6 |
| Random (dynamic) ($\epsilon$ =1e-3) | 2e-2 | 5e-3 | 5e-4 | 5e-5 | 5e-6 |
| Weights with largest magnitude (static) ($\epsilon$ =1e-3) | 1e-3 | 5e-4 | 2e-4 | 1e-4 | 2e-5 |
| Weights with smallest magnitude (static) ($\epsilon$ =1e-3) | 1e-3 | 5e-4 | 1e-4 | 1e-5 | 2e-6 |
| smallest GraSP scores (static) ($\epsilon$ =1e-3) | 2e-5 | 1e-5 | 5e-6 | 2e-6 | 2e-6 |

(c) **COPA**

| Methods | 1e-5 | 1e-4 | 1e-3 | 1e-2 | 1e-1 |
|---|---|---|---|---|---|
| **Sensitive (C4, static)** ($\epsilon$ =1e-3) | 5e-6 | 1e-6 | 5e-7 | 1e-6 | 1e-6 |
| Sensitive (task-specific, static) ($\epsilon$ =1e-3) | 1e-5 | 1e-6 | 2e-6 | 1e-6 | 5e-7 |
| Sensitive (task-specific, dynamic) ($\epsilon$ =1e-3) | 5e-6 | 1e-6 | 1e-6 | 1e-6 | 5e-7 |
| Random (static) ($\epsilon$ =1e-3) | 1e-2 | 2e-3 | 5e-4 | 5e-5 | 5e-6 |
| Random (dynamic) ($\epsilon$ =1e-3) | 2e-3 | 1e-3 | 2e-4 | 2e-5 | 2e-6 |
| Weights with largest magnitude (static) ($\epsilon$ =1e-3) | 1e-3 | 5e-4 | 5e-4 | 1e-4 | 1e-5 |
| Weights with smallest magnitude (static) ($\epsilon$ =1e-3) | 2e-3 | 5e-4 | 2e-5 | 2e-6 | 2e-6 |
| smallest GraSP scores (static) ($\epsilon$ =1e-3) | 5e-6 | 5e-6 | 1e-6 | 2e-6 | 1e-6 |

For the smaller FO-Adam experiment in Table 2, we report the used learning rates in Table 10. We use a batch size of 8 and train for 1000 steps. We use the Adam optimizer with linear learning rate decay (without warmup) to 0 and no weight decay. We evaluate the model's performance at the end of 1000-step training.

Table 10: The chosen learning rates for experiments in OPT-1.3B FO FT in Table 2. We repeat each learning rate for 3 random trials and report the average and standard deviation in Table 2.

|  | **Methods** | SST-2 | RTE | CB | BoolQ | WSC | WiC | COPA | WinoG |
|---|---|---|---|---|---|---|---|---|---|
| FO-Adam | Full FT | 1e-5 | 2e-5 | 1e-5 | 1e-5 | 2e-5 | 1e-5 | 1e-5 | 2e-5 |

## G.2 TASK-SPECIFIC PROMPTS IN EXPERIMENTS

We describe our task templates in Table 11 and 12.

Table 11: Task templates for all experiments (1/2). On the left column we include the task name and the model name, and on the right column we describe the exact prompt with answer candidates.

| Task | Prompts |
|------|---------|
| SST-2 (Llama2-7B, Llama2-13B) | ### Sentence: \<text> ### Sentiment: negative/positive |
| SST-2 (Mistral-7B, OPT-6.7B, OPT-13B) | \<text> It was terrible/great |
| RTE (Llama2-7B, Llama2-13B) | Suppose "\<premise>" Can we infer that "\<hypothesis>"? Yes or No? Yes/No |
| RTE (Mistral-7B, OPT-6.7B, OPT-13B) | \<premise> Does this mean that "\<hypothesis>" is true? Yes or No? Yes/No |
| CB (Llama2-7B, Mistral-7B, OPT-6.7B, Llama2-13B, OPT-13B) | Suppose \<premise> Can we infer that "\<hypothesis>"? Yes, No, or Maybe? Yes/No/Maybe |
| BoolQ (Llama2-7B, Llama2-13B) | \<passage> \<question>? Yes/No |
| BoolQ (Mistral-7B, OPT-6.7B, OPT-13B) | \<passage> \<question>? Yes/No |
| WSC (Llama2-7B, Mistral-7B, OPT-6.7B, Llama2-13B, OPT-13B) | \<text> In the previous sentence, does the pronoun "\<span2>" refer to \<span1>? Yes or No? Yes/No |
| WiC (Llama2-7B, Mistral-7B, OPT-6.7B, Llama2-13B, OPT-13B) | Does the word "\<word>" have the same meaning in these two sentences? Yes, No? \<sent1> \<sent2> Yes/No |
| COPA (Llama2-7B, Mistral-7B, OPT-6.7B, Llama2-13B, OPT-13B) | \<premise> so/because \<candidate> |
| WinoGrande (Llama2-7B, Mistral-7B, OPT-6.7B, Llama2-13B, OPT-13B) | \<context> \<option> |

Table 12: Task templates for all experiments (2/2).

| | |
|---|---|
| Arc-E, Arc-C, OBQA, MMLU (Mistral-7B) | Question: \<question\>
(A) \<choice A\>
(B) \<choice B\>
(C) \<choice C\>
(D) \<choice D\>
Answer: A/B/C/D |
| HS (HellaSwag) (Mistral-7B) | \<context\> \<candidate\> |
| PIQA (Mistral-7B) | Goal: \<goal\>?
Answer: \<candidate\> |
| SIQA (Mistral-7B) | Question: \<question\>
(A) \<choice A\>
(B) \<choice B\>
(C) \<choice C\>
Answer: A/B/C |
| AQuA (Mistral-7B) | Question: \<question\>
(A) \<choice A\>
(B) \<choice B\>
(C) \<choice C\>
(D) \<choice D\>
(E) \<choice E\>
Answer: A/B/C/D/E |

## G.3 ON-DEVICE MEMORY CONSTRAINTS

As illustrated in Table 13, a wide range of mobile or edge devices impose a memory constraint of **8 GB**, which is the our target when we develop our SensZOQ in Section 3.2.

Table 13: Device memory of some mobile devices or consumer-graded GPUs.

| Devices | Memory |
|---|---|
| NVidia GeForce GTX 1080 Ti | 11 GB |
| NVidia GeForce RTX 3060 Ti | 8 GB |
| NVidia Jetson TX2 | 8 GB |
| OPPO Find X7 Ultra (Li et al., 2024a) | 12 GB |
| Samsung Galaxy S10 with Mali-G76 GPU (Gim & Ko, 2022) | 8 GB |

## G.4 HARDWARE, PLATFORM, LIBRARIES, AND OTHER DETAILS FOR FINE-TUNING AND BENCHMARKING

Figure 15 (subfigure 1 and 3) is trained and evaluated on an single GPU node with 1 NVidia RTX A6000 GPU and 1 Intel Xeon Gold 6342 CPU, with PyTorch version 2.2, HuggingFace Transformer version 4.36, and CUDA 12.2. In subfigure 2 and 4 in Figure 15, we use NVidia A100-SXM4 (40 GB) and AMD EPYC 7543P 32-Core CPU with PyTorch version 2.1, HuggingFace version 4.38.2, and CUDA 12.2. We use Flash Attention 2 (Dao, 2023) in HuggingFace Transformers library throughout our experiments, and the base model for ZO full fine-tuning and benchmarking is always Llama2-7B with Float16 datatype (torch.float16). We also use the Float16 datatype (torch.float16) for all of our sparse parameters (sensitive sparse, random subsets, etc.) in ZO fine-tuning experiments. **In our preliminary experiments we found BrainFloat16 with 7 bits (ULP=$2^{-7}$ ≈8e-3) of mantissa cannot make ZO FT's training loss decreases, but ZO FT still works with Float16 that has 10 bits of mantissa (ULP=$2^{-10}$ ≈1e-3). We leave the investigation between ZO FT and the number of precision bits to future research.** We still use Float16 for FO-SGD Full FT while using BrainFloat16 for FO-Adam Full FT.

In Figure 15, we use sequence length of 512 and batch size 16 sampled from WikiText-2 dataset (Merity et al., 2016) as a representative computational intensity for ZO training.

