# OpenReview forum: "Zeroth-Order Fine-Tuning of LLMs with Transferable Static Sparsity"
_ICLR.cc/2025/Conference — ICLR 2025 Poster_

### Official Review · Reviewer_upxM · 2024-10-27

**Soundness:** 2
**Presentation:** 2
**Contribution:** 3
**Rating:** 6
**Confidence:** 3

**Summary:**

This paper introduces SensZOQ, a method that combines static sparse zeroth-order (ZO) fine-tuning with quantization to address the limitations of standard ZO. SensZOQ focuses on fine-tuning a small subset (0.1%) of "sensitive" parameters, while keeping the remaining parameters quantized to reduce memory usage and maintain competitive performance on various downstream tasks. This framework enables efficient fine-tuning of an LLaMA2-7B model on a GPU with less than 8GB of memory, outperforming traditional full-model ZO in both memory efficiency and learning performance.

**Strengths:**

**Memory Efficiency**: The method significantly reduces memory usage, allowing LLM fine-tuning on low-memory devices (less than 8GB).

**Quantization Integration**: Incorporating quantization further enhances memory efficiency without substantial performance loss.

**Resource-Constrained Applicability**: Designed for mobile and low-power devices, making it suitable for on-device personalization.

**Weaknesses:**

Some key experiment comparisons and discussions are missing; please refer to the questions for details.

Some abbreviations are not defined before their first use, making the paper somewhat challenging to follow. For example:
1. “FO” is not defined before it first appears.
2. Baseline methods listed on the x-axis of Fig. 1 are not explained in either the figure caption or the text referencing the figure, which hinders understanding of this key figure.

**Questions:**

For **Table 1**:
1. Although the proposed method surpasses the listed methods, it would be helpful to also report the fully fine-tuned FO results as a baseline.
2. Since this paper emphasizes training efficiency in both memory and wall-clock time, providing memory and time comparisons would help to further illustrate its effectiveness.
3. Could the authors explain why the proposed method achieves better performance compared to ZO Full FT (the fourth row), despite incorporating both sparse funetuning and quantization?
4. The introduction section discusses various advanced ZQ methods; comparing the proposed method with these would better highlight and validate its effectiveness.

For **Table 2**:
1. Similarly, it would be helpful to include both memory and runtime comparisons to better demonstrate the effectiveness of the proposed method.
2. Could the authors explain why the proposed method achieves better performance than 16-bit ZO-SGD (the second row), despite using both sparsity and 4-bit quantization?
3. While this paper focuses on training efficiency and notes that the memory consumption of 16-bit FO-Adam on OPT-1.3B is higher than that of 4-bit ZO-SGD on OPT-6.7B, I am curious about their memory and runtime consumption from an inference perspective.

---

> ### Author Response · Authors · 2024-11-26
> **Official Comments by Authors [1/4]**
>
> We sincerely appreciate your recognition of our work and thank you for your constructive suggestions. We believe that addressing your suggestions will substantially improve our work.
>
> ## [W1 & W2 - Clarity on this key figure - Yes we will add more clear explanation to the caption of Figure 1.]
>
> FO is the abbreviation for "First-order" (exact gradient computation with backpropagation) as compared to "Zeroth-order" (gradient estimated from random perturbation).
>
> X-ticks in Figure 1:
> - "SensZOQ" is our method (Figure 4 is the workflow).
>
> - "MeZO" refers to ZO-SGD Full FT with random seed trick. This refers to Malladi et al's works [1].
>
> - "FO-Adam LoRA" means fine-tuning LoRA on top of Llama2-7B with FO Adam optimizer.
>
> - "FO-Adam FT" means full fine-tuning (FT) with FO Adam optimizer.
>
> We agree with the reviewer that Figure 1 requires some context to understand upon the first time of reading. We will add more explanation to the caption of Figure 1 in our next revision.
>
>
> ## [Table 1 Q1 - results for FO - Sure! We have added to our rebuttal revision pdf]
>
> Thanks for this insightful comment! We have added the Adam BF16 Full FT results to Table 1 in our rebuttal revision. We also include the results for FO SGD FP16 results together (to ablate from the effect of optimizer choice). Meanwhile, we will list in the table below for our Table 1.
>
> Summary of 3 tables:
>
> - On average, **SensZOQ 4 bit** are ~1.5% accuracy higher than ZO Full FT while ~4.0% lower than FO regime.
> - **SensZOQ 4 bit** has notable performance on SST2, WSC, WiC, COPA, (and BoolQ for Llama2 and CB & WinoG for OPT-6.7B) that effectively closing the gap with FO.
>
>
> **Llama2-7B**
> | Method | SST2 $\uparrow$ | RTE $\uparrow$ | CB $\uparrow$ | BoolQ $\uparrow$ | WSC $\uparrow$ | WiC $\uparrow$ | COPA $\uparrow$ | WinoG $\uparrow$ | **Avg acc** $\uparrow$|
> | ------ | ----- |----- | ----- | ----- | ----- | ----- | ----- | ----- | ----- |
> | **SensZOQ, 4 bit** | 94.7 | 74.7 | 66.7 | 83.0 | 57.4 | 65.2 | 85.0 | 65.7 | 74.1 |
> | ZO-SGD, Full FT, FP16 | 94.6 | 73.3 | 66.7 | 81.9 | 58.0 | 61.9 | 82.7 | 63.1 | 72.8 |
> | ICL, FP16 | 94.8 | 71.5 | 72.6 | 77.5 | 53.2 | 61.1 | 87.0 | 67.5 | 73.2 |
> | FO-SGD, Full FT, FP16 | 95.4 | 84.1 | 73.2 | 85.1 | 62.8 | 72.0 | 85.3 | 71.1 | 78.6 |
> | FO-Adam, Full FT, BF16 | 96.0 | 85.1 | 86.9 | 85.5 | 57.7 | 71.8 | 87.7 | 79.2 | 81.2 |
>
>
> **Mistral-7B**
> | Method | SST2 $\uparrow$ | RTE $\uparrow$ | CB $\uparrow$ | BoolQ $\uparrow$ | WSC $\uparrow$ | WiC $\uparrow$ | COPA $\uparrow$ | WinoG $\uparrow$ | **Avg acc** $\uparrow$|
> | ------ | ----- |----- | ----- | ----- | ----- | ----- | ----- | ----- | ----- |
> | **SensZOQ, 4 bit** | 94.0 | 78.0 | 70.2 | 75.1 | 59.6 | 63.6 | 88.3 | 74.1 |  75.4 |
> | ZO-SGD, Full FT, FP16 | 94.6 | 74.6 | 68.8 | 76.6 | 54.8 | 62.6 | 88.3 | 72.2 | 74.1 |
> | ICL, FP16 | 60.7 | 55.2 | 33.3 | 46.8 | 50.4 | 63.8 | 88.7 | 74.0 | 59.1 |
> | FO-SGD, Full FT, FP16 | 94.9 | 87.6 | 85.7 | 86.1 | 62.5 | 70.8 | 88.3 | 82.1 | 82.3 |
> | FO-Adam, Full FT, BF16 | 95.1 | 86.4 | 88.1 | 83.1 | 64.7 | 72.7 | 82.7 | 85.9 | 82.3 |
>
> **OPT-6.7B**
> | Method | SST2 $\uparrow$ | RTE $\uparrow$ | CB $\uparrow$ | BoolQ $\uparrow$ | WSC $\uparrow$ | WiC $\uparrow$ | COPA $\uparrow$ | WinoG $\uparrow$ | **Avg acc** $\uparrow$|
> | ------ | ----- |----- | ----- | ----- | ----- | ----- | ----- | ----- | ----- |
> | **SensZOQ, 4 bit** | 94.9 | 72.8 | 83.3 | 73.9 | 59.3  | 62.0 | 84.0 | 65.0 | 74.4 |
> | ZO-SGD, Full FT, FP16 | 94.4 | 72.7 | 79.8 | 72.1 | 57.4 | 60.2 | 82.3 | 64.6 | 72.9 |
> | ICL, FP16 |  74.0 | 65.8 | 54.8 | 67.9 | 53.2 | 41.0 | 80.7 | 61.5 | 62.4 |
> | FO-SGD, Full FT, FP16 | 95.2 | 81.8 | 92.3 | 79.2 | 59.0 | 66.5 | 85.7 | 68.8 | 78.6 |
> | FO-Adam, Full FT, BF16 | 95.7 | 81.1 | 83.9 | 81.1 | 56.1 | 66.5 | 81.3 | 66.4 | 76.5 |
>
>
> [1] Malladi, Sadhika, et al. "Fine-tuning language models with just forward passes." Advances in Neural Information Processing Systems 36 (2023): 53038-53075.

---

> ### Author Response · Authors · 2024-11-29
> **Official Comment by Authors [2/4]**
>
> ## [Table 1 Q2 & Table 2 Q1 - memory & runtime comparisons - representative memory results are shown in Figure 1, runtime will be added to our camera-ready revision.]
>
> Table 1 Q2
>
> > Since this paper emphasizes training efficiency in both memory and wall-clock time, providing memory and time comparisons would help to further illustrate its effectiveness.
>
> Table 2 Q1
>
> > Similarly, it would be helpful to include both memory and runtime comparisons to better demonstrate the effectiveness of the proposed method.
>
>
> ### Memory
>
> - **We profile the memory consumption of 3 SuperGLUE tasks RTE, WiC, and COPA for Llama2-7B  in Figure 1** and we believe this represents the usual memory profile for a finetuning workflow (see our first reply to Reviewer KZSn).
>
> ### Runtime
>
> - The specific runtime of our SensZOQ is usually bottlenecked by the exact quantization implementation we use. The sparsity-induced overheads are minimum due to the extreme sparsity, and dequantization would contribute most to the runtime.
>
> - **We still provide an example runtime of sensitive sparse ZO-SGD optimizing with unquantized dense weights in Appendix H of our paper.** Meanwhile, we are still working on improving the runtime by enhancing the SqueezeLLM's quantization kernel codes, but this is not the main focus of our paper.

---

> ### Author Response · Authors · 2024-11-29
> **Official Comments by Authors [3/4]**
>
> ## [Table 1 Q3 & Table 2 Q2 - intuition behind the effectiveness of SensZOQ - due to 3 reasons as below ]
>
> Table 1 Q3
>
> > Could the authors explain why the proposed method achieves better performance compared to ZO Full FT (the fourth row), despite incorporating both sparse funetuning and quantization?
>
> Table 2 Q2
>
> > Could the authors explain why the proposed method achieves better performance than 16-bit ZO-SGD (the second row), despite using both sparsity and 4-bit quantization?
>
>
> This is an insightful comment on the high-level understanding behind SensZOQ! We believe the success is attributed to these 3 reasons:
>
> ## 1. Static sparsity works in general for ZO.
>
> ZO optimizes the parameters via random walks on the parameter space. The ambiguity in the exact descent direction in high dimensional space makes ZO hard. So people traditionally do not believe that ZO is not the right option for LLM finetuning. **The whole idea behind MeZO [1] is that the implicit search space of LLM FT is small** so ZO FT on LLM can somehow work, as shown in their local-r effective rank assumption.
>
> On the other hand, **static sparsity methods *explicitly* restricts the search space and makes convergence faster.** This is shown in our Figure 5a that different static sparse methods (even random) would all be higher than ZO full FT baseline at 10\% (0.1) ratio-to-optimize level.
>
> &nbsp;
>
> ## 2. Sensitive parameters are particularly effective in static sparsity.
>
> We choose sensitive parameters because it is **more effective** than other static sparse methods under the extreme sparsity regime (<1% tunable parameters). This is shown in our Figure 5a.
>
> **Figure 5a also suggests that if we choose the sparsity level 1e-2 (1%), SensZOQ's performance is likely to increase for another 1-2% average accuracy.** For inference latency and runtime purpose we choose 1e-3 level as it is already sufficient to outperform ZO full FT across tasks.
>
> &nbsp;
>
> ## 3. C4 sensitive parameters are transferable.
>
> Modern LLMs are usually well pre-trained before entering the FT phase. C4 is a common pretraining data mixtures and it is often used for calibration in both quantization [2, 3] and KV cache community [4]. C4 also covers a wide range of domains. So we choose C4 to produce a *generally good* transferable masks that will be used for *all tasks*.
>
> In our Figure 15 (page 34), we give an average overlap ratio between top gradient entries from C4 and the top gradient entries from downstream tasks for 3 commonsense reasoning, 1 math, and 3 superglue tasks. Top-0.1% gradients in C4 indeed covers ~20% of top-0.1% task gradient entries.
>
>
> ## How does 4-bit quantization affect here?
>
> 4-bit quantization makes sensitive sparse finetuning slightly worse, but only less than 1% average accuracy difference. We believe that the SOTA 4-bit quantization is effectively lossless now so the performance gap is small.
>
> For OPT-13B results in Table 4B (page 22), the C4 sensitive sparse finetuning (last row) without quantization achieves 72.7 average accuracy while with quantization (SensZOQ, first row) achieves 72.6. So the performance gap is small.
>
> **Thanks for this insightful comment and we will add such discussion to our camera-ready revision!**
>
> &nbsp;
>
> [1] Malladi, Sadhika, et al. "Fine-tuning language models with just forward passes." Advances in Neural Information Processing Systems 36 (2023): 53038-53075.
>
> [2] Frantar, Elias, et al. "Gptq: Accurate post-training quantization for generative pre-trained transformers." arXiv preprint arXiv:2210.17323 (2022).
>
> [3] Chee, Jerry, et al. "Quip: 2-bit quantization of large language models with guarantees." Advances in Neural Information Processing Systems 36 (2024).
>
> [4] Liu, Zichang, et al. "Deja vu: Contextual sparsity for efficient llms at inference time." International Conference on Machine Learning. PMLR, 2023.

---

> ### Author Response · Authors · 2024-11-29
> **Official Comment by Authors [4/4]**
>
> ## [Table 1 Q4 - advanced ZQ methods - advanced ZQ methods don't exist yet, advanced ZO methods exist but orthogonal to our work. ]
>
> Table 1 Q4
> > The introduction section discusses various advanced ZQ methods; comparing the proposed method with these would better highlight and validate its effectiveness.
>
> Thanks for this insightful comment!
>
> To the best of our knowledge, we are the first integration between ZO and quantization so advanced "ZQ" method doesn't exist yet. So we assume here it means advanced "ZO" method. Yes, there are multiple **ZO optimizers** that explore loss landscape better.
>
> **We note that the choice of ZO optimizer (SGD) is orthogonal to our focus on which parameters to optimize (C4 sensitive parameters)**. We choose SGD only because its simplicity, and in preliminary experiments we find that Adam does not accelerate ZO's convergence (we speculate that the high uncertainty in ZO's descent direction would eliminate the benefits brought from adaptive learning rate on the exact gradients.)
>
> We also note that **the nature of static sparse finetuning is *parameter-efficient*, making the adoption of SensZOQ to *any* optimizer choice fairly memory-efficient**. So such integration would deficit the purpose of our work. We will provide results on integrating with other ZO optimizers in the camera-ready revision.
>
>
>
> ## [Table 2 Q3 - inference latency & memory - inference latency is dependent on the quantization impl. For short prefills and decodes, inference memory will fit within 8 GiB. Long prefills and decodes are bottlenecked by KV-cache size.]
>
>
> Table 2 Q3
> > While this paper focuses on training efficiency and notes that the memory consumption of 16-bit FO-Adam on OPT-1.3B is higher than that of 4-bit ZO-SGD on OPT-6.7B, I am curious about their memory and runtime consumption from an inference perspective.
>
> Thanks for this insightful comment!
>
>
> ## Inference latency
>
> The inference latency of SensZOQ is dependent and still bottlenecked by the dequantization parts. **We profile an example inference throughput of sensitive sparse with unquantized FP16 dense weights in Appendix H.** We will similarly profile the time-to-first-token (TTFT) latency of unquantized FP16 dense weights in our camera-ready revision.
>
>
> ## Inference memory
>
> For Llama2-7B, SensZOQ's model weights are 4.2 GiB and the forward pass takes another 1-2 GiB. **So for the scenario of short prefills and short decodes, the memory consumption of SensZOQ will fit within 8 GiB memory.**
>
> The memory bottleneck for the scenario of long prefills or decodes is on KV-cache. This is not the focus of our paper.

---

> ### Author Response · Authors · 2024-11-30
> **Follow-up to Author Rebuttal**
>
> Dear Reviewer upxM,
>
> Thanks again for helping review our paper! Since we are approaching the end of the author-reviewer discussion period, would you please check our response and our revised paper and see if there is anything we could add to further strengthen our paper?
>
> We really appreciate the time and efforts you spend on reviewing our paper and the suggestions you have provided!
>
> Best, Authors

---

> ### Author Response · Authors · 2024-12-02
> **Discussion period ends soon, and we can still address remaining concerns**
>
> Dear Reviewer upxM,
>
> We sincerely thank your constructive comments on helping us improve our paper! As we are approaching the end of discussion period, we would greatly appreciate the opportunity to address any remaining concerns you may have regarding our response.
>
> Once again, we sincerely appreciate your time and effort in the review process.
>
> Best,
>
> Authors

---

> > ### Comment · Reviewer_upxM · 2024-12-02
> > **Thank you for your responses**
> >
> > Thank the authors for their responses, which have addressed some of my concerns. As a result, I will be raising my score.

---

> > > ### Author Response · Authors · 2024-12-02
> > > **Thanks for the reply!**
> > >
> > > Thanks for reading our response and raising the score! We are happy to address any further concerns.

---

### Official Review · Reviewer_dmR6 · 2024-11-03

**Soundness:** 2
**Presentation:** 3
**Contribution:** 3
**Rating:** 6
**Confidence:** 4

**Summary:**

The paper proposes a sparsity-aware ZO fine-tuning technique and the authors point out the method can be useful in on-device fine-tuning.

**Strengths:**

I believe the paper has the following strength:

1. ZO can help reduce memory consumption due to gradient and optimizer. When used in combination with quantization, dense weights can also be compressed, further reducing memory consumption.

2. Supporting the "sparse sensitive parameters" assumption with cumulative normalized sum of gradient norm square is intuitive, and apply it sparse ZO fine-tuning (SensZOQ) is elegant.

3. I appreciate detailed formulation on convergence rate of SensZOQ, along with proof for Eq.5 and Eq. 6.

4. Solid ablation studies on the efficacy of SensZOQ versus other weight importance metrics. Detailed analysis of gradient sparsity and sparse sensitive parameter dynamics in the appendix. Implementation and experiment details for reproducibility.

**Weaknesses:**

I have two major concerns about the paper.

1. The first is about its experiment section and its practicality. On device personalization is an important topic. I understand the authors might want to focus on edge applications and on some easier NLP benchmarks. However, all benchmarks used are can be easily handled with decoder-encoder transformers, yet at a much smaller scale. For example, RoBERTa (see page 8 in their paper) can outperform SensZOQ applied to LLMs (numbers shown in in Table 1 and Figure 5). RoBERTa-base only consists of 125M parameters, with RoBERTa-large at 355M scale. On the other hand, even with SOTA W3/W4 quantization, LLaMA2-7B takes > 3GiB of memory, just for forward pass. And this number is significantly higher than using smaller decoder-encoder transformers like RoBERTa, T5 etc. In this case, for , why should I use SensZOQ fine-tuning + LLMs instead of regular fine-tuning + smaller decoder-encoder transformer models?

2. The second concern is about the assumption: the transferable static sparsity argument is a strong argument. In Figure 3 and Figure 5, the authors have shown empirical evidences that the hypothesis holds for some tasks including RTE, WiC, COPA. However, how well this can be generalized to some other harder tasks (math, common sense reasoning, code, MMLU etc.)?

**Questions:**

I will be much more convinced if the authors could provide answers to the following questions (from high to low priority):
1. What is the justification of applying SensZOQ to LLaMA2-7B vs. directly fine-tuning decoder-encoder transformers like RoBERTa? It seems those smaller transformers consume less memory (great for edge devices), yet achieve better performance.
2. Can you provide some experiment results to show how well SensZOQ performs on harder common-sense reasoning, math/code reasoning and domain-knowledge benchmarks like (PIQA, gsm8k/humaneval, MMLU etc.)? I think those are the benchmarks that LLMs can excel in comparison with those smaller models. Even if SensZOQ fine-tuning doesn't perform well on those benchmarks and there is a big gap between SensZOQ and full-parameter fine-tuning, I think it would be fruitful to inform the community about its limitations.
3. Is the code going to be open-sourced in the future?
4. Since the loss landscapes of LLMs can be highly non-convex and complex, would assumption 2, 3 (page 17) always hold? If there are literature studies that discuss this issue, could you provide the pointers?
5. Could the authors elaborate on the concept of "fixed (gradient) features" and explain why it holds? Additionally, more evidence supporting the validity of transferable static sparsity across other tasks would be helpful.

---

> ### Author Response · Authors · 2024-11-26
> **Official Comment by Authors [1/4]**
>
> We sincerely appreciate your recognition of our work and thank you for your constructive suggestions. We believe that addressing your suggestions will substantially improve our work.
>
> ## [W1 & Q1, Q2 - Harder tasks that small-scaled RoBERTa or T5 cannot perform well - Sure! We add the results on commonsense reasoning, Math, and MMLU datasets]
> > W1: The first is about its experiment section and its practicality ... In this case, why should I use SensZOQ fine-tuning + LLMs instead of regular fine-tuning + smaller decoder-encoder transformer models?
>
> > Q1: What is the justification of applying SensZOQ to LLaMA2-7B vs. directly fine-tuning decoder-encoder transformers like RoBERTa? It seems those smaller transformers consume less memory (great for edge devices), yet achieve better performance.
>
> > Q2: Can you provide some experiment results to show how well SensZOQ performs on harder common-sense reasoning, math/code reasoning and domain-knowledge benchmarks like (PIQA, gsm8k/humaneval, MMLU etc.)? I think those are the benchmarks that LLMs can excel in comparison with those smaller models. Even if SensZOQ fine-tuning doesn't perform well on those benchmarks and there is a big gap between SensZOQ and full-parameter fine-tuning, I think it would be fruitful to inform the community about its limitations.
>
> We will merge these 3 points and discuss them together as below:
>
> We agree with the reviewer that harder tasks are more meaningful to the modern applications of LLM than SuperGLUE tasks. **In our main paper, we choose SuperGLUE tasks to follow standard ZO evaluation benchmarks [1] [2]**.
>
> **To the best of our knowledge, no ZO-LLM papers have yet evaluated reasoning or math tasks. We take a pioneering step in this direction and establish the following baselines.**
>
>
> **Mistral-7B** (test set accuracy over 3 runs, with the average reported in the last column). We tag `cs` for 8 commonsense reasoning tasks, and `math` for a math algebraic word problem dataset ([AQuA_Rat](https://huggingface.co/datasets/deepmind/aqua_rat))
> | Weight bitwidth | Method | Arc-E (`cs`) | Arc-C (`cs`) | HS (`cs`) | OBQA (`cs`) | PIQA (`cs`) | SIQA (`cs`) | BoolQ (`cs`) | WinoG (`cs`) | AQuA_Rat (`math`) | MMLU | **Avg Acc** |
> | ------ | ------ | ----- |----- | ----- | ----- | ----- | ----- | ----- | ----- | -----  | ----- | ----- |
> | 16 bit | Zeroshot | 86.8 | 75.9 | 77.9 | 71.0 | 82.1 | 59.9 | 43.4 |66.2 | 23.5 | 57.5 | 64.4 |
> | 16 bit | ICL | 90.5 | 80.0 | 80.3 | 79.8 | 84.5  | 69.9 | 46.8 | 74.0 | 26.6 | 59.2 | 69.2 |
> | 16 bit | ZO, Full FT (MeZO) | 89.2 | 78.6 | 80.5 | 76.4 | 84.1 | 67.6  | 76.6 | 72.2 | 24.1 | 59.2 | 70.9 |
> | 4 bit | **SensZOQ, C4 grad mask, 0.1% tunable params** | 88.6 | 77.9 | 82.1 | 76.5 | 84.5 | 68.1 | 75.1 | 74.1 | 27.7 | 58.2  | **71.3** |
>
> Small-scaled RoBERTa and T5 models cannot solve these hard tasks. For example, RoBERTa-large gets 56.7% accuracy [3] in Arc-C and RoBERTa-base gets 27.9% accuracy [4] in MMLU.
>
> We can see **SensZOQ** still achieves the best average performance among ZO Full FT and ICL. For the tasks (Arc-C, MMLU) that there exists a performance gap between SensZOQ and ZO Full FT, we speculate the data distribution mismatch might play a role here. If we change the source of our gradient mask from C4 to [OpenWebMath](https://huggingface.co/datasets/open-web-math/open-web-math) (a pile of Internet math proof) will close this performance gap (**see our discussion on the sparse mask source in our first response to Reviewer Agsv**).
>
> | Weight bitwidth | Method | Arc-C |  MMLU |
> |  ------ | ------ | ----- |----- |
> | 16 bit | ZO, Full FT (MeZO) | 78.6 | 59.2 |
> | 4 bit | SensZOQ, OpenWebMath grad mask, 0.1% tunable params | 78.5 | 58.8 |
> | 4 bit | SensZOQ, C4 grad mask, 0.1% tunable params | 77.9 | 58.2 |
>
>
>
> [1] Malladi, S., Gao, T., Nichani, E., Damian, A., Lee, J. D., Chen, D., & Arora, S. (2023). Fine-tuning language models with just forward passes. Advances in Neural Information Processing Systems, 36, 53038-53075.
>
> [2] Zhang, Yihua, et al. "Revisiting Zeroth-Order Optimization for Memory-Efficient LLM Fine-Tuning: A Benchmark." Forty-first International Conference on Machine Learning.
>
> [3] https://leaderboard.allenai.org/csqa2/submission/caddp8ok8ujcq3b5eiqg
>
> [4] https://paperswithcode.com/sota/multi-task-language-understanding-on-mmlu

---

> ### Author Response · Authors · 2024-11-26
> **Official Comment by Author [2/4]**
>
> ## [W2 & Q5 - transferable sparsity in harder tasks & understanding of fixed (gradient) features - Sure! We will add such discussion to our paper]
>
> > W2: The second concern is about the assumption: the transferable static sparsity argument is a strong argument. In Figure 3 and Figure 5, the authors have shown empirical evidences that the hypothesis holds for some tasks including RTE, WiC, COPA. However, how well this can be generalized to some other harder tasks (math, common sense reasoning, code, MMLU etc.)?
>
> > Q5: Could the authors elaborate on the concept of "fixed (gradient) features" and explain why it holds? Additionally, more evidence supporting the validity of transferable static sparsity across other tasks would be helpful.
>
> Thanks for this insightful comment! We will merge these 2 points and discuss them together below:
>
> &nbsp;
> ## Transferable static sparsity in commonsense reasoning tasks and math reasoning task
> &nbsp;
> ### [Sparsity of Mistral 7B's sensitive parameters in reasoning tasks](https://anonymous.4open.science/r/SensZOQRebuttal-9EE4/Figures/mistral-7b-sensitive-parameters-sparsity.png)
>
> We pick 3 commonsense reasoning tasks: `Arc-C`, `HellaSwag`, and `PIQA`, and 1 math reasoning task `AQuA` for Mistral-7B. We save 3 checkpoints: before FT, after 10% of FT steps, and after the FT. For each model checkpoint, we compute the cumulative sum of gradient squares during FO-Adam full FT for 1000 steps.
>
> We can see that sensitive parameters are still fairly sparse for these 4 tasks.
>
> - For 3 commonsense reasoning tasks, the argument that 0.1% of parameters contribute about 50% gradient norm still holds.
>
> - For the math task, the sensitive parameters tend to be slightly denser, but 1% of parameters can still cover 50% gradient norm (0.1% will cover ~30% gradient norm).
>
> &nbsp;
> ### [Transferability of C4 gradient mask in reasoning tasks for Mistral 7B](https://anonymous.4open.science/r/SensZOQRebuttal-9EE4/Figures/mistral-7b-C4-transferability.png)
>
> We compute the covered task gradient squares $[\mathcal{F}(w)]_i^2$ by C4 gradient mask.
>
>   - There are still no solid lines (for all top-(1e-2, 1e-3, 1e-4)) parameters with C4 mask) vanishes to 0, **we can see C4 gradient mask still demonstrates useful transferability to these 3 commonsense and 1 math task.** For top 1e-3 C4 gradient mask, the lowest covered gradient norm is ~0.2, while the maximum possible (*task grad, dyn.*) is ~0.6 across tasks.
>
>   - We still note that for AQuA_Rat (math algebraic word problem task), C4's transferability is weaker than the other 3 commonsense reasoning tasks. We speculate that this is due to the need to learn more math-related knowledge during FT as the covered task gradient squares by task gradient mask before FT mask (*task grad, static*) also declines more during FT.
>
> &nbsp;
> ## Understanding fixed gradient features.
> &nbsp;
>
> This is an interesting pheonomenon that is originally proposed by Malladi et al. [1], and has a connection to "skill neuron" that is also observed in LLM community. The fundadmental idea is that (1) certain parameters are critical to downstream task performance, and (2) finetuning these parameters **only** is sufficient for a satisfactory performance [3]. Similarly, Wang et al. studies "skill neurons" also observe that (1) "skill neurons" are developed in pretraining and (2) task-specific (3) crucial for task performance.
>
> In this work, **we assume that if we extract sensitive parameters from a diverse pretraining corpus (C4), we can cover these important parameters for a diverse set of tasks**. To validate the empirical success:
>
> 1. We give empirical evidence on the covered top gradient entries by C4 across SuperGLUE and reasoning tasks.
>     - We also add an [overlap ratio of top gradient entries from C4, OpenWebMath, ArXiv, and Wiki with 3 commonsense, 1 math, and 3 superglue tasks](https://anonymous.4open.science/r/SensZOQRebuttal-9EE4/Figures/mistral-7b-7tasks-top-overlap.png) to support this claim. For a quick overview, the task average is shown [here](https://anonymous.4open.science/r/SensZOQRebuttal-9EE4/Figures/mistral-7b-7tasks-top-overlap-avg.png).
> 2. We evaluate its performance and find it generally outperforms ZO full FT.
>
>
> [1] Malladi, Sadhika, et al. "A kernel-based view of language model fine-tuning." International Conference on Machine Learning. PMLR, 2023.
>
> [2] Wang, Xiaozhi, et al. "Finding Skill Neurons in Pre-trained Transformer-based Language Models." Proceedings of the 2022 Conference on Empirical Methods in Natural Language Processing. 2022.
>
> [3] Panigrahi, Abhishek, et al. "Task-specific skill localization in fine-tuned language models." International Conference on Machine Learning. PMLR, 2023.

---

> ### Author Response · Authors · 2024-11-26
> **Official Comment by Authors [3/4]**
>
> ## [Q3 - open-source the codebase - Yes ! We will release the code upon the acceptance. Meanwhile we will provide a preliminary demo as soon as possible.]
>
> As title. We will release the code upon the acceptance. Meanwhile we will provide a preliminary demo version as soon as possible!

---

> ### Author Response · Authors · 2024-11-26
> **Official Comment by Authors [4/4]**
>
> ## [Q4 - Assumptions on L-smoothness & P.L. condition in LLM - these are standard assumptions even in LLM community, and our theory could be extended to their relaxed versions]
>
> > Since the loss landscapes of LLMs can be highly non-convex and complex, would assumption 2, 3 (page 17) always hold? If there are literature studies that discuss this issue, could you provide the pointers?
>
> Thanks for this insightful comment!
>
> Assumption 2, 3 in Page 17 refer to L-smoothness and P.L. condition. **These are standard assumptions in nonconvex optimization community and also widely adopted in LLM community** [1, 2, 3, 4]. We give 2 convergences rate with adopting L-smoothness (Eqn 5) and L-smoothness + PL condition (Eqn 6).
>
> There are certain relaxation both L-smoothness and PL conditions. **The whole idea of L-smoothness is that gradients wouldn't vary too much in the parameter space (bounded gradient changes). Otherwise convergence analysis would become at least difficult if not infeasible.** There are certain relaxations to these 2 assumptions e.g. local smoothness ($(L_0, L_1)-\text{smoothness}$) [5]. We believe we can adopt a similar convergence rate with these relaxed assumptions, but as this is not a major focus of this paper (**as we focus on how to drive the success of memory-efficient LLM finetuning instead of designing a general ML optimization algorithm**). We adopt the standard assumptions that mostly everyone else use and give a convergence rate on top of it.
>
>
> [1] Malladi, S., Gao, T., Nichani, E., Damian, A., Lee, J. D., Chen, D., & Arora, S. (2023). Fine-tuning language models with just forward passes. Advances in Neural Information Processing Systems, 36, 53038-53075.
>
> [2] Dong, Yiming, Huan Li, and Zhouchen Lin. "Convergence Rate Analysis of LION." arXiv preprint arXiv:2411.07724 (2024).
>
> [3] Zhang, Liang, et al. "DPZero: dimension-independent and differentially private zeroth-order optimization." International Workshop on Federated Learning in the Age of Foundation Models in Conjunction with NeurIPS 2023. 2023.
>
> [4] Ling, Zhenqing, et al. "On the convergence of zeroth-order federated tuning for large language models." Proceedings of the 30th ACM SIGKDD Conference on Knowledge Discovery and Data Mining. 2024.
>
> [5] Zhang, Jingzhao, et al. "Why Gradient Clipping Accelerates Training: A Theoretical Justification for Adaptivity." International Conference on Learning Representations. 2019.

---

> ### Author Response · Authors · 2024-11-30
> **Follow-up to Author Rebuttal**
>
> Dear Reviewer dmR6,
>
> Thanks again for helping review our paper! Since we are approaching the end of the author-reviewer discussion period, would you please check our response and our revised paper and see if there is anything we could add to further strengthen our paper?
>
> We really appreciate the time and efforts you spend on reviewing our paper and the suggestions you have provided!
>
> Best, Authors

---

> ### Author Response · Authors · 2024-12-02
> **Discussion period ends soon, and we can still address remaining concerns**
>
> Dear Reviewer dmR6,
>
> We sincerely thank your constructive comments on helping us improve our paper! As we are approaching the end of discussion period, we would greatly appreciate the opportunity to address any remaining concerns you may have regarding our response.
>
> Once again, we sincerely appreciate your time and effort in the review process.
>
> Best,
>
> Authors

---

> > ### Author Response · Authors · 2024-12-03
> > **Few hours before the end of reviewer reply period**
> >
> > Dear Reviewer dmR6,
> >
> > The discussion period will end within a few hours, and we greatly appreciate the opportunity to hear from your final thoughts on our revision and rebuttal. We will respond as soon as possible for any last minute comments.
> >
> > Once again, we sincerely appreciate your time and effort in the review process.
> >
> > Best,
> >
> > Authors

---

> > > ### Comment · Reviewer_dmR6 · 2024-12-03
> > >
> > > Thank you for clarifying the theoretical premises and for adding additional experiments.
> > >
> > > However, while I acknowledge the sparsity of LLMs' sensitive parameters in reasoning tasks, I remain concerned about the claim of transferable sparsity, especially for tasks specifically tailored to LLMs. As a key novelty presented in the paper, the static sparsity derived from the pre-training corpus does not appear to transfer as effectively to tasks where LLMs excel (as seen in Figure 13, the gap is still non-negligible).
> > >
> > > That said, the results on commonsense QA, math, and MMLU demonstrate that applying SensZOQ to LLMs does draw a meaningful distinction compared to simply using smaller models like RoBERTa. This supports your claim of enabling lightweight on-device fine-tuning for LLMs. Therefore, I raise my rating to 6.

---

> > > > ### Author Response · Authors · 2024-12-04
> > > > **Thank you for your comment and raising the score.**
> > > >
> > > > Dear Reviewer dmR6
> > > >
> > > >
> > > > We sincerely thank the reviewer for recognizing the sparsity of LLMs' sensitive parameters in reasoning tasks and for raising the evaluation score. As shown in our experiments with commonsense QA, math, and MMLU, we are open to exploring additional tasks that can further motivate and validate SensZOQ. We believe such efforts align with the community goal of broadening the impact of customized LLMs.

---

### Official Review · Reviewer_Agsv · 2024-11-06

**Soundness:** 4
**Presentation:** 3
**Contribution:** 3
**Rating:** 6
**Confidence:** 4

**Summary:**

This paper proposes a memory-efficient zeroth-order (ZO) optimization approach for fine-tuning large language models (LLMs) in memory-limited environments, such as mobile devices. By tuning only a sparse subset of "sensitive" parameters derived from pre-training, the proposed method, SensZOQ, enables efficient on-device fine-tuning without backpropagation. SensZOQ demonstrates competitive performance compared to in-context learning and full-model ZO fine-tuning, even on GPUs with less than 8 GiB of memory. Key contributions include the identification of transferable sparse parameters and the integration of static sparsity for enhanced efficiency. While SensZOQ addresses memory limitations effectively, the ZO optimization’s slower convergence and static sparsity’s limited adaptability across tasks are noted as areas for potential improvement.

**Strengths:**

Intuitive and Effective Approach Supported by Theory: The paper presents an intuitive yet impactful approach by focusing on a sparse subset of sensitive parameters for zeroth-order (ZO) optimization. This idea is both straightforward and theoretically sound, and the authors support it well with theoretical analysis that adds credibility to their method.

Strong Experimental Validation: Through extensive experiments, the paper demonstrates the superiority of its proposed method. SensZOQ is shown to perform effectively across multiple tasks, validating the approach's potential for on-device fine-tuning in memory-constrained environments.

**Weaknesses:**

Sensitivity Assessment Limited to C4 Dataset: The authors measure sensitivity based solely on the C4 dataset, which predominantly covers natural language tasks. This raises concerns about the applicability of their sensitivity-based sparse tuning approach to tasks with different characteristics, such as code generation. It remains uncertain whether the proposed method would generalize well to tasks with substantially different data distributions and requirements.

Evaluation on Small Model Sizes Only (7B): The experiments focus on models with a maximum of 7 billion parameters, which is relatively small compared to leading models. This raises questions about whether the proposed approach would remain effective on larger models where more precise fine-tuning might be required. It would be valuable to see an assessment of SensZOQ’s applicability and performance on larger, state-of-the-art models where scaling issues might become more prominent.

Lack of Analysis on Slow Convergence Issue: The paper notes that ZO optimization can suffer from slow convergence, which is a well-documented drawback of ZO methods. However, it lacks an in-depth analysis of this issue. A discussion of potential mitigations or how this slow convergence might impact the method’s performance, especially in time-sensitive applications, would have added valuable context.

Unclear Insights from Theoretical Convergence Rate (Section 2.3): While the authors present a theoretical convergence rate, the practical implications or lessons from this analysis are unclear. It would benefit the reader to understand what actionable insights can be drawn from this theoretical result. Without this clarity, the theoretical component feels somewhat disconnected from the rest of the work.

Similarity to Existing Research on Activation Outliers: Although the paper asserts the effectiveness of using a static mask, this approach bears resemblance to previous work on activation outliers. If the sensitivity mask is closely related to known methods for identifying important model parameters, such as activation outliers, the novelty of this contribution may be limited. Further discussion comparing SensZOQ with prior work in this area could clarify its unique contributions.

Insufficient Details on Quantization Methodology: The paper does not provide adequate details regarding the specific quantization technique used. Given the potential impact of quantization on model performance, a clearer explanation is necessary to assess the reproducibility of the experiments. Without this information, readers may find it challenging to replicate the results, which is a significant drawback for scientific validation.

**Questions:**

Please refer to the weaknesses.

---

> ### Author Response · Authors · 2024-11-25
> **Official Comment by Author [1/5]**
>
> We sincerely appreciate your recognition of our work and thank you for your constructive suggestions. We believe that addressing your suggestions will substantially improve our work.
>
> ## [W1 - Sensitivity Assessment Limited to C4 Dataset - Sure! We add an additional ablation study.]
>
> In the paper, we choose C4 as a calibration dataset that (1) is commonly used in pretraining (2) has diverse texts that we hope could span a wide range of downstream tasks. C4 is a common choice for calibration for SOTA quantization method like GPTQ and QuIP. So using C4 would make our method naturally fits with the existing quantization pipeline.
>
> **If we have better knowledge on the exact downstream domain or task our method will be applied to, we can extract sparse masks with better specific task performance**. So we include an additional ablation study regarding 3 different masks from
>
> (1) [ArXiv](https://huggingface.co/datasets/armanc/scientific_papers), a pile of ArXiv papers. We use the ArXiv articles subset from this dataset.
>
> (2) [OpenWebMath](https://huggingface.co/datasets/open-web-math/open-web-math), a pile of Internet mathematical proofs.
>
> (3) [Wiki103](https://huggingface.co/datasets/Salesforce/wikitext), a pile of selected Wikipedia articles.
>
> Mistral does not disclose the exact pretraining dataset they use, but these datasets are commonly selected for pretraining data mixtures (e.g. OLMo).
>
> We also include the standard commonsense reasoning task results (ArcE, ArcC, HellaSwag, OBQA, PIQA, SIQA, BoolQ, WinoGrande), and math algebraic word problem dataset ([AQuA_Rat](https://huggingface.co/datasets/deepmind/aqua_rat)) [1], MMLU, and Superglue tasks (RTE, WiC, Copa) that we use in our paper as belows. For each task, we add a tag `CS` as commonsense reasoning, `math` as math problem, `sg` as Superglue tasks.
>
> **Mistral-7B** (test set accuracy over 3 runs, with the average reported in the last column)
> | Weight bitwidth | Method | ArcE (`cs`) $\uparrow$ | ArcC (`cs`) $\uparrow$ | HS (`cs`) $\uparrow$ | OBQA (`cs`) $\uparrow$ | PIQA (`cs`) $\uparrow$ | SIQA (`cs`) $\uparrow$ | BoolQ (`cs, sg`) $\uparrow$ | WinoG (`cs`) $\uparrow$ | AQuA_Rat (`math`) $\uparrow$ | MMLU $\uparrow$ | RTE (`sg`) $\uparrow$ | WiC (`sg`) $\uparrow$ | Copa (`sg`) $\uparrow$ | **Avg Acc** $\uparrow$ |
> | ------ | ------ | ----- |----- | ----- | ----- | ----- | ----- | ----- | ----- | ----- | ----- | ----- | ----- | ----- | ----- |
> | 16 bit | ICL | **90.5** | **80.0** | 80.3 | **79.8** | **84.5**  | **69.9** | 46.8 | 74.0 | 26.6 | **59.2** | 55.2 | **63.8** | 74.0 | 68.0 |
> | 16 bit | ZO, Full FT (MeZO) | 89.2 | 78.6 | 80.5 | 76.4 | 84.1 | 67.6  | 76.6 | 72.2 | 24.1 | **59.2** | 74.6 | 62.6  | 88.3 |  71.8 |
> | 4 bit | **SensZOQ, OpenWebMath, 0.1% tunable** | 87.8 | 78.5 | 81.8 | 74.1 | 83.7 | 67.2 | 71.5 | 72.2 | 25.2 | 58.8 | 66.7 | 60.8 | **89.0** |  70.6 |
> | 4 bit | **SensZOQ, ArXiv, 0.1% tunable** | 87.7 | 77.0 | **82.7** | 75.2 | 84.2 | 68.7 | 69.1 | 72.2 | 25.9 | 58.8 | 70.0 |  59.4  |  **89.0**  | 70.8 |
> | 4 bit | **SensZOQ, Wiki103, 0.1% tunable** | 87.8 | 77.9 | 82.0 | 73.0 | 83.9 | 68.6 | **79.7** | 73.2 | 26.4 | 57.6 | 69.2 | 60.9 |  88.7  | 71.5 |
> | 4 bit | **SensZOQ, C4, 0.1% tunable** | 88.6 | 77.9 | 82.1 | 76.5 | **84.5** | 68.1 | 75.1 | **74.1** | **27.7** | 58.2 | **78.0**  |  63.6  |  88.3  | **72.5** |
>
> **We can see that C4 still achieves the best average accuracy**, with notable performance on QA tasks like OBQA and PIQA, and NLU tasks like RTE and WIC. If we want better performance on education or hard reasoning tasks like MMLU or Arc-C, OpenWebMath and ArXiv is a better choice than C4. **In our paper, we do not claim C4 is the *only* choice but rather a *generally good* choice for downstream tasks**, and this is quite important as **we are using the *same* set of sparse parameters for different tasks and we want it to be suitable for different tasks** (otherwise we will have to create a separate sparse mask and quantized models for each task and this will make our method really unpractical).
>
> **We still note that the performance of other sensitivity masks from datasets like `ArXiv`, `OpenWebMath`, and `Wiki103` is *not* bad** -- they all at least have a comparable performance as ZO Full FT as shown in the last column. They might not perform as well as C4 on NLU or commonsense QA tasks, but the gap is usually not large. Meanwhile, they are more performant on MMLU and Arc-C so the average performance is kinda got balanced. We agree that there would be some performance degradation by data distribution mismatch, and we will leave how the find the best data distribution from the pretraining data mixtures as a future work.
>
> **We really appreciate this feedback and we will add this result and related discussion to the next revision!**

---

> ### Author Response · Authors · 2024-11-25
> **Official Comment by Author [2/5]**
>
> ## [W2 - Evaluation on Small Model Sizes Only (7B)  - Sure, We will add OPT-13B & Llama2-13B results.]
>
> It is common for different model families to have ~7B models (Llama2-7B, Llama3-8B, Gemma-7B, Mistral-7B, OLMo-7B, Falcon-7B). The next common scale beyond 7B model is usually 70B level, which we believe it is not of our paper's main interest as it will demand multiple high-end GPUs (80 GiB A100 or H100) even with SOTA model compression techniques. In addition, there are some related works indicate that the gap between finetuning and ICL in 70B regime is not large. **Given this crucial insight, we deliberately focus on the objective of promoting efficient fine-tuning within the 7B model regime, where we can achieve a great balance in performance and efficiency. The main interest of our paper is to enable efficient on-device finetuning, where 7B models are better positioned than their 70B counterparts to serve as the foundation for widespread deployment.**
>
> We still provide the results of OPT-13B as below, and we can see the performance of our SensZOQ is still maintained, although we sometimes need to surpass the 8 GiB memory budget (but 16 GiB is still generally sufficient).
>
> **OPT-13B** (test set accuracy over 3 runs, the average accuracy reported in the last column)
> | Method | SST2 $\uparrow$ | RTE $\uparrow$ | CB $\uparrow$ | BoolQ $\uparrow$ | WSC $\uparrow$ | WiC $\uparrow$ | COPA $\uparrow$ | WinoG $\uparrow$ | **Avg acc** $\uparrow$|
> | ------ | ----- |----- | ----- | ----- | ----- | ----- | ----- | ----- | ----- |
> | Zeroshot | 61.0 | 58.5 | 48.2 | 60.0 | 36.5 | 52.0 | 80.0 | 60.9 | 57.1 |
> | ICL | 83.0 | 59.8 | 72.0 | 71.6 | 38.1 | 53.6 | 84.0 | 63.2 | 65.6 |
> | ZO Full FT (MeZO) | **93.9** | 74.0 | **67.9** | 72.4 | **61.5** | 58.6 | 87.0 | 63.3 | 72.3 |
> | **SensZOQ** | 93.8 | **76.7** | 65.5 | **72.8** | 59.9 | **59.9** | **88.7** | **63.7** | **72.6** |
>
>
>
> --------------------------------------------------- Update -------------------------------------------------------
>
> We also finish an additional experiment on Llama2-13B and will add to the next revision of our paper.
>
> **Llama2-13B** (test set accuracy over 3 runs, with the average accuracy across tasks reported in the last column)
> | Method | SST2 $\uparrow$ | RTE $\uparrow$ | CB $\uparrow$ | BoolQ $\uparrow$ | WSC $\uparrow$ | WiC $\uparrow$ | COPA $\uparrow$ | WinoG $\uparrow$ | **Avg acc** $\uparrow$|
> | ------ | ----- |----- | ----- | ----- | ----- | ----- | ----- | ----- | ----- |
> | Zeroshot | 30.4 | 50.9 | 44.6 | 57.0 | 41.3  | 50.8  | 83.0  | 65.2 | 52.9 |
> | ICL | 38.8 | 51.4 | 22.6 | 59.9 | 36.5 | 52.1 | 88.7 | 71.0 | 52.6 |
> | ZO Full FT (MeZO) | **95.0** | 74.4 | 85.7 | 80.5 | 54.8 | **60.9** | **90.7** | 70.5 | 76.6 |
> | **SensZOQ** | 94.8 | **76.5** | **86.3** | **80.7** | **56.7** | 58.0 | 88.7 | **72.9** | **76.8** |

---

> ### Author Response · Authors · 2024-11-25
> **Official Comment by Authors [3/5]**
>
> ## [W3 - Lack of Analysis on Slow Convergence Issue & Unclear Insights from Theoretical Convergence Rate - Sure! we will discuss more in next revision!]
>
> > Lack of Analysis on Slow Convergence Issue: The paper notes that ZO optimization can suffer from slow convergence, which is a well-documented drawback of ZO methods. However, it lacks an in-depth analysis of this issue. A discussion of potential mitigations or how this slow convergence might impact the method’s performance, especially in time-sensitive applications, would have added valuable context.
>
> > Unclear Insights from Theoretical Convergence Rate (Section 2.3): While the authors present a theoretical convergence rate, the practical implications or lessons from this analysis are unclear. It would benefit the reader to understand what actionable insights can be drawn from this theoretical result. Without this clarity, the theoretical component feels somewhat disconnected from the rest of the work.
>
> Thanks for this insightful comments! We will merge these 2 points and discuss them together as belows:
>
> A key contribution of our paper is demonstrating how **pretraining-derived information (gradient mask) can effectively guide downstream zero-order (ZO) finetuning**.
>
> - ## Analysis on slow convergence of ZO
> ZO's slow convergence is rooted in the ambiguity of update direction which relies solely on random walks in the parameter space.
>
> - ## Our solution to this issue (important parameter selection)
> We tackle this problem by leveraging three key insights about sensitive parameters in pre-trained LLMs
>
> (1) Natural sparsity in parameter importance
>
> (2) Ability to guide ZO optimization toward crucial parameters rather than a full-scaled search
>
> (3) (approximately) identifiable during pre-training and on the cloud.
>
> - ## Other people's solutions (adaptively adjusting ZO perturbation/update directions)
>
> Classical algorithm like CMA-ES and recent ZO optimizer design such as HiZZO [1] follow such path as effectively restricting the isotropic search space $\mathcal{N}(0, I_d)$ to either a covariance matrix with adaptive variances $C$ (CMA-ES) or preconditioned by a loss-landscape-informed covariance matrix $\Sigma_t$ (HiZZO).
>
> **We thank reviewer for this insightful feedback and we will add these discussion as a paragraph to the Section 2 in our next revision!**
>
> [1] Zhao, Yanjun, et al. "Second-order fine-tuning without pain for llms: A hessian informed zeroth-order optimizer." arXiv preprint arXiv:2402.15173 (2024).

---

> ### Author Response · Authors · 2024-11-26
> **Official Comment by Authors [4/5]**
>
> ## [W4 - Similarity to Existing Research on Activation Outliers - No, sensitive parameters are different from activation outliers]
>
> The emergence of activation outliers in LLM is an interesting phenomenon to research in quantization & attention/KV cache community as it contributes to massive attention scores to certain dimensions [1] (crucial for KV cache reduction and fuzzy attention score computation) and make sub-4bit quantization of certain parameters hard [2].
>
> **We note that the research on sensitive parameters (fixed parameter entries with gradient outliers) are *different* from the research in activation outliers.** Activation outliers are mainly on forward passes and is caused by some special input structures (e.g. delimiter tokens or starting tokens)  [1]. **Sensitive parameters are featuring the maximum loss value changes upon random perturabtion and that is not what massive activations are describing** (see our Section 2.2). Although sensitive parameters are still data-dependent (leveraging pretraining text corpuses), **sensitive parameters do not take advantage of specific input structures.**
>
> **The only 2 sparse ZO works we have known is SparseMeZO [3] that uses smallest weight magnitude mask and random mask, which we have compared in our Figure 5a in the context of static transferability.**
>
>
> [1] Sun, Mingjie, et al. "Massive Activations in Large Language Models." ICLR 2024 Workshop on Mathematical and Empirical Understanding of Foundation Models.
>
> [2] Heo, Jung Hwan, et al. "Rethinking Channel Dimensions to Isolate Outliers for Low-bit Weight Quantization of Large Language Models." The Twelfth International Conference on Learning Representations.
>
> [3] Liu, Yong, et al. "Sparse mezo: Less parameters for better performance in zeroth-order llm fine-tuning." arXiv preprint arXiv:2402.15751 (2024).

---

> ### Author Response · Authors · 2024-11-26
> **Official Comment by Authors [5/5]**
>
> ## [W5 - Insufficient Details on Quantization Methodology - ]
>
> ## Quantization methodology
> We use SqueezeLLM [1] and their code is open-sourced: https://github.com/SqueezeAILab/SqueezeLLM. We first extract the sensitive parameters out and keep them as FP16, and for the rest of parameters we follow exactly as the SqueezeLLM 4-bit quantization recipe with C4 gradient masks.
>
>
> ## Relationship to our work
>
> **SensZOQ's workflow (Figure 4) is framework-independent with regards to different quantization methods, as long as the quantization approach is minimizing the layer-wise output differences in a lease-squares sense (line 317 in our paper).**
>
> SOTA quantization methods like QuIP GTPQ and SqueezeLLM are effectively lossless in 4-bit regime, and the mainstream methods are usually minimizing layerwise least square loss with certain variants (e.g. GPTQ, QuIP). **In this case, our method will maintain the performance as long as the quantization methodology is *reasonably good*.**  We have intentionally kept this separation to ensure our approach remains broadly applicable across different quantization implementations.
>
> [1] Kim, Sehoon, et al. "SqueezeLLM: Dense-and-Sparse Quantization." Forty-first International Conference on Machine Learning.
>
> [2] Frantar, Elias, et al. "Gptq: Accurate post-training quantization for generative pre-trained transformers." arXiv preprint arXiv:2210.17323 (2022).

---

> ### Author Response · Authors · 2024-11-30
> **Follow-up to Author Rebuttal**
>
> Dear Reviewer Agsv,
>
> Thanks again for helping review our paper! Since we are approaching the end of the author-reviewer discussion period, would you please check our response and our revised paper and see if there is anything we could add to further strengthen our paper?
>
> We really appreciate the time and efforts you spend on reviewing our paper and the suggestions you have provided!
>
> Best, Authors

---

> > ### Comment · Reviewer_Agsv · 2024-11-30
> > **Thank you for your effort**
> >
> > Thank you for your sincere effort in addressing my concerns. W1, W2, and W5 have been clearly addressed. However, I am still not fully convinced by your responses to W3 and W4. Based on this, I have adjusted my score to 6.

---

> > > ### Author Response · Authors · 2024-12-01
> > > **Concerns on W3 & W4**
> > >
> > > Thanks for reading our response and raising the score! Could you elaborate more on which part we are not fully addressing in W3 and W4? We are happy to address any of the concerns and include them in our next revision. Thanks again for your time!

---

### Official Review · Reviewer_KZSn · 2024-11-07

**Soundness:** 3
**Presentation:** 4
**Contribution:** 3
**Rating:** 6
**Confidence:** 4

**Summary:**

This paper proposes SensZOQ, a workflow framework sparse zeroth-order optimization with quantization for on-device fine-tuning of LLM. It identifies a small, static subset of "sensitive" parameters in pre-training, transfers them to downstream tasks, and quantizes the rest to reduce memory demands. Experiments on 7B-level LLMs show SensZOQ outperforms other methods in memory-constrained settings, achieving better performance than in-context learning and full ZO Full-FT.

**Strengths:**

The work enables fine-tuning of Llama2-7B model on <8GiB GPU memory by using only 0.1% static sensitive parameters and quantizing the rest, meeting memory constraints of edge/mobile devices and enabling more customization scenarios at the edge.

SensZOQ outperforms ICL and ZOFull-FT, achieving competitive results across various downstream tasks and models like SST-2, RTE, etc.

The work also conducts details theory analysis and overview of related ZO methods (Sec 2,3) and reveals that 0.1% contributes about 50% gradient norm scales.

**Weaknesses:**

The paper is well-presented with a clear structure. My main concerns revolve around the experiment setup and comparisons.

How does the fine-tuning fit within 8GB of memory? Does this include the memory for the model states themselves? For instance, the authors claim that LLaMA-7B fits within 8GB memory constraints. However, even when quantized to 4 bits, storing the model alone requires 3.5GB, not accounting for other intermediate buffers needed during training.

What is the transferability of the sparse mask (e.g., sensitive sparse masks), and how is it generated? Is this sparse mask generally applicable across different downstream tasks, or does SensZOQ require a new one for each task? Lines 236-240 reference a fixed work, but the authors don't provide an ablation study to support this claim.

The convergence curve or training curve is missing. As mentioned in the introduction and related work, zero-order (ZO) methods suffer from slow convergence. However, the authors only discuss the theoretical convergence rate without providing an empirical curve. Additionally, in all experiment tables, the results of full fine-tuning (Adam, BF16) are missing, and no actual memory usage or throughput data is provided.

**Questions:**

What is the rank meaning in Table 1, 2? Average all ranks?

---

> ### Author Response · Authors · 2024-11-25
> **Official Comment by Authors [1/5]**
>
> We sincerely appreciate your recognition of our work and thank you for your constructive suggestions. We believe that addressing your suggestions will substantially improve our work.
>
> ## [W1 - Memory Profile - Sure, let's dive the memory consumption deeper!]
>
> > Does this include the memory for the model states themselves?
>
> Yes, this is shown exactly on our Figure 1 (1st bar in each subfigure). We can give a breakdown for RTE (left subfigure in Figure 1) as an example:
>
>
> **Memory consumption (in GiB) for Llama2-7B finetune on RTE**
>
> (we report the memory in 1 GiB = $\dfrac{2^{30}}{1000^3} = 1.07$ GB, and Llama2 has 13.5 GB = 12.5 GiB, but this is a minor detail as the exact value in GiB or GB is roughly the same).
>
> | Method | Model weights | Optimization | ZO Forward | ZO Perturb | FO Forward & backward |
> | ------ | ----- |----- | ----- | ----- |  ----- |
> | FO-Adam, Full FT | 12.5 | 37.7 | NA | NA | 23.3 |
> | FO-Adam, LoRA | 12.7 | 0.7 | NA | NA | 24.4 |
> | ZO-SGD, Full FT | 12.5 | 0.3 | **1.1** | 0.3 | NA |
> | **SensZOQ** | **4.2** | **0.0003** | **1.1** | **0.0003** | NA |
>
> By using *stateless* SGD, SensZOQ doesn't have optimizer states. **We also use random seed trick to rematerialize the random Gaussian matrix for perturbation during optimization step.** The memory consumption in optimization steps is bottlenecked by initializing a random Gaussian matrix with same size as 0.1% $\times$ largest weight matrix (up/down projection weight in SwiGLU). This has a magnitude as $O(\dfrac{4096 \times 11008 \times 4 \times 0.001}{2^{30}}) = O(0.00016)$ GiB.
>
> The memory consumption of ZO perturbation is also bottlenecked by initializing the largest random Gaussian matrix as $O(0.00016)$ GiB.
>
> The memory consumption of SensZOQ's forward is identical to that of MeZO as 1.1 GiB. This is dependent on the batch size and sequence length but generally will not exceed 2 GiB (for RTE it is 1.1, for WiC it is 0.3, for COPA it is 0.9 GiB).
>
> And this includes *everything* (model weights/forward/optimization) during SensZOQ's finetuning stage.
>
> **This memory efficiency comes from the extreme sparsity what SensZOQ's taking advantages: optimizing 0.1% parameters in ZO finetuning, and quantize the dense parts into 4 bits.**

---

> ### Author Response · Authors · 2024-11-25
> **Official Comment by Authors [2/5]**
>
> ## [W2 -  Transferability of the sparse mask - We have included more results!]
> > What is the transferability of the sparse mask (e.g., sparse masks), and how is it generated?
>
> We take a random sample of 100 calibration sequences from C4 dataset and compute the gradient w.r.t. causal language modeling loss. We then extract out the parameters with highest gradient square magnitude as our tunable parameters.
>
>
> **Transferability means we can use the *same fixed* set of sparse parameters for every downstream task**, which is exactly what we did. We choose C4 as a calibration dataset that (1) is commonly used in pretraining (2) has diverse text corpuses that we hope could span a wide range of downstream tasks.
>
> **If we have better knowledge on the exact downstream domain or task our method will be applied to, we can extract sparse masks with better specific task performance**. So we include an additional ablation study regarding 3 different masks from
>
> (1) [ArXiv](https://huggingface.co/datasets/armanc/scientific_papers), a pile of ArXiv papers. We use the ArXiv articles subset from this dataset.
>
> (2) [OpenWebMath](https://huggingface.co/datasets/open-web-math/open-web-math), a pile of Internet mathematical proofs
>
> (3) [Wiki103](https://huggingface.co/datasets/Salesforce/wikitext), a pile of selected Wikipedia articles.
>
> We also note that Mistral does not disclose the exact pretraining dataset they use, but these datasets are commonly selected for pretraining data mixtures (e.g. Dolma).
>
> We also include the standard commonsense reasoning task results (Arc-E, Arc-C, HellaSwag, OBQA, PIQA, SIQA, BoolQ, WinoGrande), and math algebraic word problem dataset ([AQuA_Rat](https://huggingface.co/datasets/deepmind/aqua_rat)), MMLU, and Superglue tasks (RTE, WiC, Copa) that we use in our paper as belows. We denote `CS` as commonsense reasoning, `math` as math problem, `sg` as Superglue tasks.
>
> **Mistral-7B** (test set accuracy, with the average reported in the last column)
> | Weight bitwidth | Method | Arc-E (`cs`) | Arc-C (`cs`) | HS (`cs`) | OBQA (`cs`) | PIQA (`cs`) | SIQA (`cs`) | BoolQ (`cs, sg`) | Wino (`cs`) | AQuA_Rat (`math`) | MMLU | RTE (`sg`) | WiC (`sg`) | Copa (`sg`) | **Avg Acc** |
> | ------ | ------ | ----- |----- | ----- | ----- | ----- | ----- | ----- | ----- | ----- | ----- | ----- | ----- | ----- | ----- |
> | 16 bit | ICL | **90.5** | **80.0** | 80.3 | **79.8** | **84.5**  | **69.9** | 46.8 | 74.0 | 26.6 | **59.2** | 55.2 | **63.8** | 74.0 | 68.0 |
> | 16 bit | ZO, Full FT (MeZO) | 89.2 | 78.6 | 80.5 | 76.4 | 84.1 | 67.6  | 76.6 | 72.2 | 24.1 | **59.2** | 74.6 | 62.6  | 88.3 |  71.8 |
> | 4 bit | **SensZOQ, OpenWebMath, 0.1% tunable** | 87.8 | 78.5 | 81.8 | 74.1 | 83.7 | 67.2 | 71.5 | 72.2 | 25.2 | 58.8 | 66.7 | 60.8 | **89.0** |  70.6 |
> | 4 bit | **SensZOQ, ArXiv, 0.1% tunable** | 87.7 | 77.0 | **82.7** | 75.2 | 84.2 | 68.7 | 69.1 | 72.2 | 25.9 | 58.8 | 70.0 |  59.4  |  **89.0**  | 70.8 |
> | 4 bit | **SensZOQ, Wiki103, 0.1% tunable** | 87.8 | 77.9 | 82.0 | 73.0 | 83.9 | 68.6 | **79.7** | 73.2 | 26.4 | 57.6 | 69.2 | 60.9 |  88.7  | 71.5 |
> | 4 bit | **SensZOQ, C4, 0.1% tunable** | 88.6 | 77.9 | 82.1 | 76.5 | **84.5** | 68.1 | 75.1 | **74.1** | **27.7** | 58.2 | **78.0**  |  63.6  |  88.3  | **72.5** |
>
> **We can see that C4 still achieves the best average accuracy**, with notable performance on QA tasks like OBQA and PIQA, and NLU tasks like RTE and WIC. If we want better performance on education or hard reasoning tasks like MMLU or Arc-C, OpenWebMath and ArXiv will become a better choice. In our paper, we do not claim C4 is the only choice but rather a *generally good* choice for downstream tasks, and this is supported by the extensive tasks above.
>
> > Is this sparse mask generally applicable across different downstream tasks, or does SensZOQ require a new one for each task?
>
> No, we are using the *same* set of sparse parameters for different tasks (otherwise we will have to create a separate sparse mask and quantized models for each task and this will make our method unpractical).

---

> ### Author Response · Authors · 2024-11-25
> **Official Comment by Author [3/5]**
>
> ## [W3 -  Convergence curve or training curve - Sure! ]
> We have included the training loss of SensZOQ vs. ZO Full FT (MeZO) in this anonymous github link: [OPT-13B training loss, same learning rate](https://anonymous.4open.science/r/SensZOQRebuttal-9EE4/Figures/opt-13b-loss-same-learning-rate.png), and [OPT-13B training loss, best learning rate of both parties](https://anonymous.4open.science/r/SensZOQRebuttal-9EE4/Figures/opt-13b-loss-best-learning-rate-of-both-side.png), and we will include in our next revision!
>
> Note that the `Sensitive ZO (C4 mask, 4 bit)` is our `SensZOQ` method, and the `Sensitive ZO (C4 mask, 16 bit)` will leave the dense weights unquantized (still in FP16) as it more aligns with our theory.
>
> 1. [OPT-13B training loss, same learning rate](https://anonymous.4open.science/r/SensZOQRebuttal-9EE4/Figures/opt-13b-loss-same-learning-rate.png): best learning rate from ZO Full FT (MeZO), also used for SensZOQ and the unquantized version
>
> * Summary of the figure: For RTE, SensZOQ has much faster convergence while for WiC and COPA, the convergence is about the same.
>
> For these 3 figures, we first search the best learning rate for ZO Full FT (MeZO) that reaches the lowest training loss in [1e-7, 2e-7, 5e-7, 1e-6] grid. We then change to Sensitive ZO with C4 gradient masks (4 bit as our SensZOQ method, and 16 bit as an ablation study) with **an identical learning rate** (as well as other hyperparameters as batch size $B$, ZO perturbation constant $\epsilon$). Our theory is more applicable to the unquantized version (Sensitive ZO with 16 bit weights), but as we are focusing on the on-device finetuning workflow, we also report such results for the 4-bit version (as our SensZOQ method).
>
> 2. [OPT-13B training loss, best learning rate of both parties](https://anonymous.4open.science/r/SensZOQRebuttal-9EE4/Figures/opt-13b-loss-best-learning-rate-of-both-side.png): best learning rate for both ZO Full FT and SensZOQ
>
> * Summary of the figure: For all of these 3 tasks (RTE, WiC, COPA), SensZOQ has much faster convergence.
>
> We conduct a learning rate grid search for both ZO Full FT and SensZOQ separately (the other hyperparams $B$ and $\epsilon$ are still the same), and make these 3 figures. We identify SensZOQ can tolerate higher learning rate without divergence. For example, the best learning rates for ZO Full FT in WiC and COPA are 2e-7 (5e-7 would result in a divergence) while *SensZOQ* can tolerate 5e-7 and produce much faster convergence results.
>
> **In conclusion, SensZOQ would have at least similar convergence as ZO Full FT with the same learning rate (so a direct learning rate transfer would still yield satisfactory results), and with the same learning rate grid search, SensZOQ can achieve much faster convergence in practice.**

---

> ### Author Response · Authors · 2024-11-25
> **Official Comment by Authors [4/5]**
>
> ## [W4 - Results for Adam BF16 Full FT - Sure! Will add in next revision]
>
> Thanks for this insightful comment! We will add the Adam BF16 Full FT results in our next revision. We also include the results for FO SGD FP16 results together (to ablate from the effect of optimizer choice). Meanwhile, we will list in the table below for our Table 1.
>
> Summary of 3 tables:
>
> - On average, **SensZOQ 4 bit** are ~1.5% accuracy higher than ZO Full FT while ~4.0% lower than FO regime.
> - **SensZOQ 4 bit** has notable performance on SST2, WSC, WiC, COPA, (and BoolQ for Llama2 and CB & WinoG for OPT-6.7B) that effectively closing the gap with FO.
>
>
> **Llama2-7B**
> | Method | SST2 $\uparrow$ | RTE $\uparrow$ | CB $\uparrow$ | BoolQ $\uparrow$ | WSC $\uparrow$ | WiC $\uparrow$ | COPA $\uparrow$ | WinoG $\uparrow$ | **Avg acc** $\uparrow$|
> | ------ | ----- |----- | ----- | ----- | ----- | ----- | ----- | ----- | ----- |
> | **SensZOQ, 4 bit** | 94.7 | 74.7 | 66.7 | 83.0 | 57.4 | 65.2 | 85.0 | 65.7 | 74.1 |
> | ZO-SGD, Full FT, FP16 | 94.6 | 73.3 | 66.7 | 81.9 | 58.0 | 61.9 | 82.7 | 63.1 | 72.8 |
> | ICL, FP16 | 94.8 | 71.5 | 72.6 | 77.5 | 53.2 | 61.1 | 87.0 | 67.5 | 73.2 |
> | FO-SGD, Full FT, FP16 | 95.4 | 84.1 | 73.2 | 85.1 | 62.8 | 72.0 | 85.3 | 71.1 | 78.6 |
> | FO-Adam, Full FT, BF16 | 96.0 | 85.1 | 86.9 | 85.5 | 57.7 | 71.8 | 87.7 | 79.2 | 81.2 |
>
>
> **Mistral-7B**
> | Method | SST2 $\uparrow$ | RTE $\uparrow$ | CB $\uparrow$ | BoolQ $\uparrow$ | WSC $\uparrow$ | WiC $\uparrow$ | COPA $\uparrow$ | WinoG $\uparrow$ | **Avg acc** $\uparrow$|
> | ------ | ----- |----- | ----- | ----- | ----- | ----- | ----- | ----- | ----- |
> | **SensZOQ, 4 bit** | 94.0 | 78.0 | 70.2 | 75.1 | 59.6 | 63.6 | 88.3 | 74.1 |  75.4 |
> | ZO-SGD, Full FT, FP16 | 94.6 | 74.6 | 68.8 | 76.6 | 54.8 | 62.6 | 88.3 | 72.2 | 74.1 |
> | ICL, FP16 | 60.7 | 55.2 | 33.3 | 46.8 | 50.4 | 63.8 | 88.7 | 74.0 | 59.1 |
> | FO-SGD, Full FT, FP16 | 94.9 | 87.6 | 85.7 | 86.1 | 62.5 | 70.8 | 88.3 | 82.1 | 82.3 |
> | FO-Adam, Full FT, BF16 | 95.1 | 86.4 | 88.1 | 83.1 | 64.7 | 72.7 | 82.7 | 85.9 | 82.3 |
>
> **OPT-6.7B**
> | Method | SST2 $\uparrow$ | RTE $\uparrow$ | CB $\uparrow$ | BoolQ $\uparrow$ | WSC $\uparrow$ | WiC $\uparrow$ | COPA $\uparrow$ | WinoG $\uparrow$ | **Avg acc** $\uparrow$|
> | ------ | ----- |----- | ----- | ----- | ----- | ----- | ----- | ----- | ----- |
> | **SensZOQ, 4 bit** | 94.9 | 72.8 | 83.3 | 73.9 | 59.3  | 62.0 | 84.0 | 65.0 | 74.4 |
> | ZO-SGD, Full FT, FP16 | 94.4 | 72.7 | 79.8 | 72.1 | 57.4 | 60.2 | 82.3 | 64.6 | 72.9 |
> | ICL, FP16 |  74.0 | 65.8 | 54.8 | 67.9 | 53.2 | 41.0 | 80.7 | 61.5 | 62.4 |
> | FO-SGD, Full FT, FP16 | 95.2 | 81.8 | 92.3 | 79.2 | 59.0 | 66.5 | 85.7 | 68.8 | 78.6 |
> | FO-Adam, Full FT, BF16 | 95.7 | 81.1 | 83.9 | 81.1 | 56.1 | 66.5 | 81.3 | 66.4 | 76.5 |

---

> ### Author Response · Authors · 2024-11-25
> **Official Comment by Author [5/5]**
>
> ## [W5 - Actual memory and throughput data - Memory reported in Figure 1, Throughput will be reported in next revision]
>
> We report the results for Llama2-7B's actual memory usage in RTE, WiC, and COPA task in Figure 1. We will benchmark and add more memory usuage results in our next revision, but we believe that generally SensZOQ on 7B models would not exceed memory consumption of 8 GiB (see W1's discussion).
>
> We are still benchmarking the throughput data and will add to our next revision.
>
>
> ## [Q1 - Rank meaning in Table 1, 2? -  Answer: avg. rank of each method across all of the tasks listed in the table]
>
> The rank in Table 1, 2 refers to the average rank of each method across all of the tasks listed in the table.
>
> We first rank all methods for each task (each column) in Table 1 and 2, and then compute the average of such rankings across all tasks.

---

> ### Author Response · Authors · 2024-11-30
> **Follow-up to Author Rebuttal**
>
> Dear Reviewer KZSn,
>
> Thanks again for helping review our paper! Since we are approaching the end of the author-reviewer discussion period, would you please check our response and our revised paper and see if there is anything we could add to further strengthen our paper?
>
> We really appreciate the time and efforts you spend on reviewing our paper and the suggestions you have provided!
>
> Best,
> Authors

---

> ### Author Response · Authors · 2024-12-02
> **Discussion period ends soon, and we can still address remaining concerns**
>
> Dear Reviewer KZSn,
>
> We sincerely thank your constructive comments on helping us improve our paper! As we are approaching the end of discussion period, we would greatly appreciate the opportunity to address any remaining concerns you may have regarding our response.
>
> Once again, we sincerely appreciate your time and effort in the review process.
>
> Best,
>
> Authors

---

> > ### Author Response · Authors · 2024-12-03
> > **Few hours before the end of reviewer reply period**
> >
> > Dear Reviewer KZSn,
> >
> > The discussion period will end within a few hours, and we greatly appreciate the opportunity to hear from your final thoughts on our revision and rebuttal. We will respond as soon as possible for any last minute comments.
> >
> > Once again, we sincerely appreciate your time and effort in the review process.
> >
> > Best,
> >
> > Authors

---

### Official Review · Reviewer_eqp9 · 2024-11-07

**Soundness:** 3
**Presentation:** 3
**Contribution:** 2
**Rating:** 5
**Confidence:** 4

**Summary:**

This paper proposes a transferable static sparse ZO LLM fine-tuning strategy. It observes an extreme sparsity pattern in LLM parameters: a subset, determined by selecting the top k magnitude entries from the diagonal of empirical Fisher information matrix, is effective for ZO fine-tuning. Moreover, it finds this sparsity pattern can be obtained through LLM’s pre-training process and transferred to various downstream tasks without modification. Building on these insights,  it proposes SensZOQ, an on-device LLM personalization workflow
via integrating Sensitive ZO optimization with Quantization to further improve the memory efficiency of ZO fine-tuning.

**Strengths:**

It identifies that only an extremely small portion (0.1%) of LLM parameters should be updated during ZO LLM fine-tuning. Moreover, it observes that this sparsity pattern can be derived in LLM pre-training process and transferred across different downstream tasks while still maintaining good ZO performance without any modification.

It conducts extensive experiments across various LLMs and demonstrate that the method achieves competitive performance across various downstream tasks.

**Weaknesses:**

The novelty may be limited. The proposed method mainly use masks to select a small set of parameters and update these parameters with ZO method while quantizing the rest parameters. The quantization and ZO methods are very typical, following previous works. The mask construction by selecting the top k magnitude entries from the diagonal of empirical Fisher information matrix has been proposed in (Sung et al., 2021, Training Neural Networks with Fixed Sparse Masks).  The idea of updating the weights with fixed sparse masks for language models has been investigated in (Sung et al., 2021) and SparseMeZO  (Liu et al., 2024a). The technical components are not novel and their combination seems to be straightforward. The technical contribution may be limited.

The comparison on memory usage may not be fair. The method quantize the model and it compares the memory of quantized model with other unquantized models from other methods. We are not sure whether simply quantizing models can lead to such memory reduction. Furthermore, during the mask construction, it needs to use backpropagation to obtain the gradients of all parameters. This part costs massive memory. But in the memory comparison, it does not count the memory cost for mask construction and only include the memory in ZO finetuning after the mask is obtained. And it also not count the memory cost for quantization part. Although the authors use a on-device flow to count on-device memory,  it may not be fair to compare part of the memory usage of the proposed method with the full memory usage of other methods.


It only experiments with 7B  or smaller models. It is better to have some results on larger models such as 13B and 30B to demonstrate the general performance on larger models. It claims to be efficient with less memory usage and faster training speed. It should be easy to apply to larger models.

It only experiments with one single finetuning dataset C4. It may be better to experiment with multiple finetuning datasets to demonstrate the general performance.

It claims to be faster in inference with higher Token generation speed. But after finetuning, it is still a dense model, and we are not sure how the faster token generation is achieved. Though there are some discussions in appendix for the sparse operations,  there do not seem to be much details and we only see some function calls without detailed implementation for the functions to understand why it is faster.  It may be due to the quantization. But quantization in the work just follows SqueezeLLM (Kim et al., 2024). It may be better to provide some high level discussions for why the inference can be faster.

**Questions:**

see the weakness.

It is better to highlight the contributions and discuss the fairness of memory comparison. It is better to have some results on larger models such as 13B and 30B to demonstrate the general performance on larger models. It may be better to provide some high level discussions for why the inference can be faster.

---

> ### Author Response · Authors · 2024-11-24
> **Official Comment by Authors [1/5]**
>
> We sincerely appreciate your recognition of our work and thank you for your constructive suggestions. We believe that addressing your suggestions will substantially improve our work.
>
> ##  [W1 - Novelty - We clarify our contribution in both idea and methodology design]
>
> ## 1. Important problem and novel high-level idea
>
> The central problem our paper is targeting to solve is memory-efficient on-device finetuning, and we propose a realistic on-device finetuning workflow and perform extensive experiments and solid theoretical supports for the effectiveness. **The high level idea of our paper is that we can leverage transferable information (fixed Fisher masks) obtained from the quantization pipeline to enable efficient ZO finetuning, and such idea is absent from *any* of the existing ZO optimization works.** Such design would eliminate the memory consumption during the finetuning to *a minimum* as shown in Figure 1. i.e. we are able to fine-tune a 7B model under 8GiB GPU memory *without any offloading*.
>
> ## 2. Careful methodology design and extensive ablations with theoretical supports
>
> We also argue that finding a transferable sparse masks suitable for *ZO* is a nontrivial task. A strong prior on determining the exact update direction or important parameters (which is our paper) is essential to maintain finetuning performance in face of fuzzy ZO gradient estimation in billion parameter scale. **We give strong empirical supports for the natural sparsity of Fisher masks in LLMs in Figure 6, 9, 10. In Section 4.2 and Figure 5 we perform extensive ablation studies on other sparse masks to support the effectiveness of transferable Fisher masks.**

---

> ### Author Response · Authors · 2024-11-24
> **Official Comment by Authors [2/5]**
>
> ## [W2 - Memory usage comparison - We have included in the Figure 1.]
>
> We would like to clarify that our work's primary innovation and contribution lies in enabling efficient on-device finetuning. Our method is specifically designed to **integrate with existing quantization workflows, not to compete with them**. To elaborate on this:
>
> - The memory comparison in Figure 1 focuses on fine-tuning memory requirements because this is bottleneck we are addressing. Existing quantization pipelines running in the cloud (e.g., GPTQ, QuIP, SqueezeLLM) all compute gradients on calibration sets such as C4 as part of their standard workflow. Our method leverages these existing computations without introducing additional memory overhead.
> - From a user perspective, which is this paper's key focus, the most important metric is the memory required for on-device fine-tuning and personalization. Our method makes this possible under resource-constrained settings where it was previously infeasible.
> - Our approach follows a practical cloud-device workflow (Figure 4) that aligns with real-world deployment scenarios. The cloud provider handles the computationally intensive steps of transferrable sparse mask computation, while the end users benefits from efficient on-device fine-tuning capabilities.
>
> **We hope this clarifies that our memory comparisons are intentionally focused on the finetuning phase**, as this represents the core technical challenge we are solving and the metric most relevant to end-user applications. That's why we are reporting the fine-tuning memory usage in Figure 1.

---

> ### Author Response · Authors · 2024-11-24
> **Official Comment by Authors [3/5]**
>
> ## [W3 - Experiment on larger models - Sure!]
>
> It is common for different model families to have ~7B models (Llama2-7B, Llama3-8B, Gemma-7B, Mistral-7B, OLMo-7B, Falcon-7B) than 13B (OPT-13B, Llama2-13B, Falcon-11B) than 30B (OPT-30B, Falcon-40B). The next common scale beyond 7B model is usually 70B level, which we believe it is not of our paper's main interest as it will demand multiple high-end GPUs (80 GiB A100 or H100) even with SOTA model compression techniques. We still provide the results of OPT-13B as belows, and we can see the performance of our SensZOQ is still maintained, although we sometimes need to surpass the 8 GiB memory budget (but 16 GiB is still generally sufficient).
>
>
> **OPT-13B** (test set accuracy over 3 runs, the average accuracy reported in the last column)
> | Method | SST2 $\uparrow$ | RTE $\uparrow$ | CB $\uparrow$ | BoolQ $\uparrow$ | WSC $\uparrow$ | WiC $\uparrow$ | COPA $\uparrow$ | WinoG $\uparrow$ | **Avg acc** $\uparrow$|
> | ------ | ----- |----- | ----- | ----- | ----- | ----- | ----- | ----- | ----- |
> | Zeroshot | 61.0 | 58.5 | 48.2 | 60.0 | 36.5 | 52.0 | 80.0 | 60.9 | 57.1 |
> | ICL | 83.0 | 59.8 | 72.0 | 71.6 | 38.1 | 53.6 | 84.0 | 63.2 | 65.6 |
> | ZO Full FT (MeZO) | **93.9** | 74.0 | **67.9** | 72.4 | **61.5** | 58.6 | 87.0 | 63.3 | 72.3 |
> | **SensZOQ** | 93.8 | **76.7** | 65.5 | **72.8** | 59.9 | **59.9** | **88.7** | **63.7** | **72.6** |
>
>
> We are currently working on the Llama2-13B results and will also provide to the appendix of our paper.
>
> We believe the scenario where we have to finetune 30B or 70B models is not a main focus of our paper as it entails multiple high-end GPUs even with SOTA weight quantization/compression methods.
>
> --------------------------------------------- Update --------------------------------------------
>
> We have included the Llama2-13B results as below and will add it to the next revision of our paper.
>
> **Llama2-13B** (test set accuracy over 3 runs, with the average accuracy across tasks reported in the last column)
> | Method | SST2 $\uparrow$ | RTE $\uparrow$ | CB $\uparrow$ | BoolQ $\uparrow$ | WSC $\uparrow$ | WiC $\uparrow$ | COPA $\uparrow$ | WinoG $\uparrow$ | **Avg acc** $\uparrow$|
> | ------ | ----- |----- | ----- | ----- | ----- | ----- | ----- | ----- | ----- |
> | Zeroshot | 30.4 | 50.9 | 44.6 | 57.0 | 41.3  | 50.8  | 83.0  | 65.2 | 52.9 |
> | ICL | 38.8 | 51.4 | 22.6 | 59.9 | 36.5 | 52.1 | 88.7 | 71.0 | 52.6 |
> | ZO Full FT (MeZO) | **95.0** | 74.4 | 85.7 | 80.5 | 54.8 | **60.9** | **90.7** | 70.5 | 76.6 |
> | **SensZOQ** | 94.8 | **76.5** | **86.3** | **80.7** | **56.7** | 58.0 | 88.7 | **72.9** | **76.8** |

---

> ### Author Response · Authors · 2024-11-24
> **Official Comment by Authors [4/5]**
>
> ## [W4 - Dataset beyond C4 - Sure! C4 still performs better on average.]
>
> In the paper, we choose C4 as a calibration dataset that (1) is commonly used in pretraining (2) has diverse texts that we hope could span a wide range of downstream tasks. *We do not finetune the sparse parameters on C4* -- we only use it for extracting the sparse masks. C4 is a common choice for calibration for SOTA quantization method like GPTQ and QuIP. So using C4 would make our method naturally fits with the existing quantization pipeline.
>
> We also note that **if we have better knowledge on the exact downstream domain or task our method will be applied to, we can extract sparse masks with better specific task performance**. So we include an additional ablation study regarding 3 different masks from
>
> (1) [ArXiv](https://huggingface.co/datasets/armanc/scientific_papers), a pile of ArXiv papers. We use the ArXiv articles subset from this dataset.
>
> (2) [OpenWebMath](https://huggingface.co/datasets/open-web-math/open-web-math), a pile of Internet mathematical proofs.
>
> (3) [Wiki103](https://huggingface.co/datasets/Salesforce/wikitext), a pile of selected Wikipedia articles.
>
> We also note that Mistral does not disclose the exact pretraining dataset they use, but these datasets are commonly selected for pretraining data mixtures (e.g. OLMo).
>
> We also include the standard commonsense reasoning task results (ArcE, ArcC, HellaSwag, OBQA, PIQA, SIQA, BoolQ, WinoGrande), and math algebraic word problem dataset ([AQuA_Rat](https://huggingface.co/datasets/deepmind/aqua_rat)) [1], MMLU, and Superglue tasks (RTE, WiC, Copa) that we use in our paper as belows. We denote `CS` as commonsense reasoning, `math` as math problem, `sg` as Superglue tasks. We also want to note that we have included *much more diverse and difficult tasks* than the MeZO paper [2] and even ZO LLM benchmark paper [3].
>
> **Mistral-7B** (test set accuracy over 3 runs, with the average reported in the last column)
> | Weight bitwidth | Method | ArcE (`cs`) $\uparrow$ | ArcC (`cs`) $\uparrow$ | HS (`cs`) $\uparrow$ | OBQA (`cs`) $\uparrow$ | PIQA (`cs`) $\uparrow$ | SIQA (`cs`) $\uparrow$ | BoolQ (`cs, sg`) $\uparrow$ | WinoG (`cs`) $\uparrow$ | AQuA_Rat (`math`) $\uparrow$ | MMLU $\uparrow$ | RTE (`sg`) $\uparrow$ | WiC (`sg`) $\uparrow$ | Copa (`sg`) $\uparrow$ | **Avg Acc** $\uparrow$ |
> | ------ | ------ | ----- |----- | ----- | ----- | ----- | ----- | ----- | ----- | ----- | ----- | ----- | ----- | ----- | ----- |
> | 16 bit | ICL | **90.5** | **80.0** | 80.3 | **79.8** | **84.5**  | **69.9** | 46.8 | 74.0 | 26.6 | **59.2** | 55.2 | **63.8** | 74.0 | 68.0 |
> | 16 bit | ZO, Full FT (MeZO) | 89.2 | 78.6 | 80.5 | 76.4 | 84.1 | 67.6  | 76.6 | 72.2 | 24.1 | **59.2** | 74.6 | 62.6  | 88.3 |  71.8 |
> | 4 bit | **SensZOQ, OpenWebMath, 0.1% tunable** | 87.8 | 78.5 | 81.8 | 74.1 | 83.7 | 67.2 | 71.5 | 72.2 | 25.2 | 58.8 | 66.7 | 60.8 | **89.0** |  70.6 |
> | 4 bit | **SensZOQ, ArXiv, 0.1% tunable** | 87.7 | 77.0 | **82.7** | 75.2 | 84.2 | 68.7 | 69.1 | 72.2 | 25.9 | 58.8 | 70.0 |  59.4  |  **89.0**  | 70.8 |
> | 4 bit | **SensZOQ, Wiki103, 0.1% tunable** | 87.8 | 77.9 | 82.0 | 73.0 | 83.9 | 68.6 | **79.7** | 73.2 | 26.4 | 57.6 | 69.2 | 60.9 |  88.7  | 71.5 |
> | 4 bit | **SensZOQ, C4, 0.1% tunable** | 88.6 | 77.9 | 82.1 | 76.5 | **84.5** | 68.1 | 75.1 | **74.1** | **27.7** | 58.2 | **78.0**  |  63.6  |  88.3  | **72.5** |
>
>
> **We can see that C4 still achieves the best average accuracy**, with notable performance on QA tasks like OBQA and PIQA, and NLU tasks like RTE and WIC. If we want better performance on education or hard reasoning tasks like MMLU or Arc-C, OpenWebMath and ArXiv will become a better choice. **In our paper, we do not claim C4 is the *only* choice but rather a *generally good* choice for downstream tasks**, and this is quite important as **we are using the *same* set of sparse parameters for different tasks and we want it to be transferable and suitable for different tasks** (otherwise we will have to create a separate sparse mask and quantized models for each task and this will make our method unpractical).
>
> **We really appreciate reviewer's feedback on this ablation study and we will add it to the next revision!**
>
>
>
> Reference:
>
> [1] Ling, Wang, et al. "Program induction by rationale generation: Learning to solve and explain algebraic word problems." arXiv preprint arXiv:1705.04146 (2017). link: https://huggingface.co/datasets/deepmind/aqua_rat
>
> [2] Malladi, S., Gao, T., Nichani, E., Damian, A., Lee, J. D., Chen, D., & Arora, S. (2023). Fine-tuning language models with just forward passes. Advances in Neural Information Processing Systems, 36, 53038-53075.
>
> [3] Zhang, Yihua, et al. "Revisiting Zeroth-Order Optimization for Memory-Efficient LLM Fine-Tuning: A Benchmark." Forty-first International Conference on Machine Learning.

---

> ### Author Response · Authors · 2024-11-24
> **Official Comment by Authors [5/5]**
>
> ## [W5 - Inference speed - We have presented in Figure 13]
>
> We are sorry for the confusion. The statement on higher token generation refers to the fact that Sensitive ZO would need to optimize far less parameters (0.1%) to reach the same performance as **other sparsity methods** in Section 3.2. The comparison is *not* made with the (quantized) dense model which would be impossible. We give a Figure 13 (1st bar) in Appendix E on how much throughput degradation would be induced from 0.1% sparse operations. We will make such claim more clear in our next revision.

---

> ### Author Response · Authors · 2024-11-30
> **Follow-up to Author Rebuttal**
>
> Dear Reviewer eqp9,
>
> Thanks again for helping review our paper! Since we are approaching the end of the author-reviewer discussion period, would you please check our response and our revised paper and see if there is anything we could add to further strengthen our paper?
>
> We really appreciate the time and efforts you spend on reviewing our paper and the suggestions you have provided!
>
> Best,
> Authors

---

> ### Author Response · Authors · 2024-12-02
> **Discussion period ends soon, and we can still address remaining concerns**
>
> Dear Reviewer eqp9,
>
> We sincerely thank your constructive comments on helping us improve our paper! As we are approaching the end of discussion period, we would greatly appreciate the opportunity to address any remaining concerns you may have regarding our response.
>
> Once again, we sincerely appreciate your time and effort in the review process.
>
> Best,
>
> Authors

---

> ### Comment · Reviewer_eqp9 · 2024-12-03
>
> Thanks for the rebuttal. After going through the rebuttal, I still have some concerns for the novelty. I am satisfied with other issues and updated my score.
>
>  As I mentioned in my review, the mask construction by selecting the top k magnitude entries from the diagonal of empirical Fisher information matrix has been proposed in (Sung et al., 2021, Training Neural Networks with Fixed Sparse Masks). The idea of updating the weights with fixed sparse masks for language models has been investigated in (Sung et al., 2021) and SparseMeZO (Liu et al., 2024a). The technical contribution may be limited. The rebuttal seems to highlight the combination of fixed Fisher masks and ZO finetuning is not explored before. But  SparseMeZO already investigates the sparse weight updating in ZO finetuning, and [Sung et al., 2021] explores how to select masks with Fisher information, it does not seem to be a significant contribution to combine them.

---

> ### Author Response · Authors · 2024-12-04
> **Thank you for your reconsideration!**
>
> We sincerely thank the reviewer for recognizing that our responses have effectively addressed concerns regarding memory usage comparison, experiments on larger models, datasets beyond C4, and inference speed. We deeply appreciate the reviewer’s acknowledgment of these efforts and for raising the score. Additionally, we provide the following clarification regarding the novelty of our work.
>
> ### Clarification on Novelty
>
> While novelty is a multifaceted concept in academic research, we believe it can be roughly viewed from two fronts: **empirical novelty**, which involves uncovering previously unknown properties and behaviors, and **technical novelty**, which focuses on the development of new methodologies or solutions.
>
> We appreciate the reviewer’s acknowledgment of the uniqueness of our work, particularly the integration of Fisher-informed sparsity and ZO. We further highlight the empirical novelty of this synergy through the following contributions:
>
> 1. **Transferable Sparse Patterns for Extremely Parameter-Efficient ZO:**
>    Our analysis and empirical results demonstrate that the proposed combination of Fisher-informed sparse masking and ZO optimization enables the discovery of a transferable, learnable parameter set with extreme sparsity (0.1%). This density is **5x** smaller than the 0.5% achieved by Sung et al. (2021) and up to **300x** smaller than Sparse MeZO, showcasing the efficiency and scalability of our approach. We also ablate on a smaller weight magnitude based mask as SparseMeZO in our Figure 5a and it doesn't perform well under extreme sparsity regime as our approach.
>
> 2. **A New Pathway for On-Device Fine-Tuning Under Compression:**
>    We introduce a novel approach to combining compression and fine-tuning during LLM deployment. By preserving extremely sparse patterns in original precision, our sparse ZO method enables effective fine-tuning of LLMs even when other parameters are under compression. This approach is compatible with most popular compression techniques, opening a new avenue for improving the performance of compressed LLMs on-device.
>
> We hope this clarification provides additional insight into the contributions of our work and its potential impact.

---

### Author Response · Authors · 2024-12-04

We sincerely thank the reviewers and area chairs for their thoughtful and insightful discussions regarding our submission. Below, we summarize the positive feedback received and the concerns addressed during the discussion.


### Reviewer's positive feedback

* **Discovery of transferable, extremely sparse parameter for ZO**
    * R **eqp9**: It identifies that only an extremely small portion (0.1%) of LLM parameters should be updated during ZO LLM fine-tuning. Moreover, it observes that this sparsity pattern can be derived in LLM pre-training process and transferred across different downstream tasks while still maintaining good ZO performance without any modification.
    * R **KZSn**: The work enables fine-tuning of Llama2-7B model on <8GiB GPU memory by using only 0.1% static sensitive parameters and quantizing the rest, meeting memory constraints of edge/mobile devices and enabling more customization scenarios at the edge.
    * R **dmR6**: Supporting the "sparse sensitive parameters" assumption with cumulative normalized sum of gradient norm square is intuitive, and apply it sparse ZO fine-tuning (SensZOQ) is elegant.




* **Extensive evaluation with good performance**
    * R **eqp9**: It conducts extensive experiments across various LLMs and demonstrate that the method achieves competitive performance across various downstream tasks.
    * R **KZSn**: SensZOQ outperforms ICL and ZO Full-FT, achieving competitive results across various downstream tasks and models like SST-2, RTE, etc.
    * R **AgsV**: Strong Experimental Validation
    * R **dmR6**: Solid ablation studies on the efficacy of SensZOQ versus other weight importance metrics. Detailed analysis of gradient sparsity and sparse sensitive parameter dynamics in the appendix. Implementation and experiment details for reproducibility.


* **Supportive theoretical analysis**
    * R **KZSn**: The work also conducts details theory analysis and overview of related ZO methods (Sec 2,3) and reveals that 0.1% contributes about 50% gradient norm scales.
    * R **AgsV**: Intuitive and Effective Approach Supported by Theory
    * R **dmR6**: I appreciate detailed formulation on convergence rate of SensZOQ, along with proof for Eq.5 and Eq. 6.


* **Practical insight in memory consumption reduction**
    * R **dmR6**: ZO can help reduce memory consumption due to gradient and optimizer. When used in combination with quantization, dense weights can also be compressed, further reducing memory consumption.
    * R **upxM**: Memory Efficiency,  Quantization Integration, and resource-Constrained Applicability.



### Addressed concerns acknowledged by the reviewers
* Evaluation on larger models - Addressed concern acknolowdged by R **eqp9** and R **Agsv**
* Evaluation on calibration datasets beyond C4 - Addressed concern acknolowdged by R **eqp9** and R **Agsv**
* Details of quantization methodology - Addressed concern acknolowdged by R **Agsv**
* Results on commonsense QA, math, and MMLU demonstrate that applying SensZOQ - Addressed concern acknolowdged by R **dmR6**
* Comparison with FO - Addressed concern acknolowdged by R **upxM**
* Efficiency evaluation including memory, time - Addressed concern acknolowdged R **upxM**
* Intuition behind the effectiveness of SensZOQ - Addressed concern acknolowdged R **upxM**


We hope the above overview will provide our AC and reviewers with a concise way to navigate through the mass information on this page.

Further, we sincerely hope our appreciation of simple but effective design, as well as our novel observations and insights into transferable and extremely sparse ZO for LLMs, can be shared with you and our fellow scholars in this important field of research.

Sincerely,
Paper8905 Authors

---

### Meta-Review · Area_Chair_UifF · 2024-12-18

**Metareview:**

The authors provided a detailed and thoughtful response to address the reviewers' questions. Post-rebuttal, the majority of reviewers felt their concerns were mostly or partially resolved, resulting in raised ratings. However, Reviewer eqp9 maintained reservations about the novelty of the method, which I believe is a valid concern. I encourage the authors to further clarify and emphasize the paper's contributions to make its novelty more prominent.

That said, many reviewers recognized the paper as a solid piece of work with insightful results and substantial contributions. Given the overall positive shift in reviewers' evaluations and the consensus that the paper's strengths significantly outweigh its weaknesses, I recommend acceptance.

**Additional Comments On Reviewer Discussion:**

Reviewer eqp9 raised valid concerns about the method's novelty, and I encourage the authors to further clarify and highlight their contributions. Nonetheless, the majority of reviewers acknowledged the paper's insightful results and substantial contributions. With the overall positive evaluations, I recommend acceptance.

---

### Decision · Program_Chairs · 2025-01-22

Accept (Poster)